# TIMEGUARD: Channel-wise Pool Training for Backdoor Defense in Time Series Forecasting

Quang Duc Nguyen [1]    Siyuan Liang [1]    Yiming Li [1]    Fushuo Huo [1]    Dacheng Tao [1]

## Abstract

Time Series Forecasting (TSF) is highly vulnerable to backdoor attacks, yet effective defenses remain underexplored due to challenges arising from data entanglement and shifts in task formulation. To fill this gap, we conduct a systematic evaluation of thirteen representative backdoor defenses across the TSF life cycle and analyze their failure modes. Our results reveal two fundamental issues: (1) data entanglement induces *channel-level signal dilution*, rendering sample-filtering and trigger-synthesis defenses ineffective at localizing backdoors; and (2) task-formulation shift leads to *training-loss degeneration*, causing poisoned and clean windows to become indistinguishable at training stages. Based on these findings, we propose a training-time backdoor defense for TSF, termed TIMEGUARD. Our method adopts channel-wise pool training as the core paradigm and initializes a high-confidence pool using time-aware criteria to mitigate signal dilution. Moreover, we introduce distance-regularized loss selection to progressively expand the reliable pool during training and ease loss degeneration. Extensive experiments across multiple datasets, forecasting architectures, and TSF backdoor attacks demonstrate that TIMEGUARD substantially improves robustness, boosting $\text{MAE}_\text{P}$ by $1.96\times$ over the leading baseline, while preserving clean performance within 5% $\text{MAE}_\text{C}$.

## 1. Introduction

Time Series Forecasting (TSF) is widely used in critical domains such as climate prediction, transportation plan-

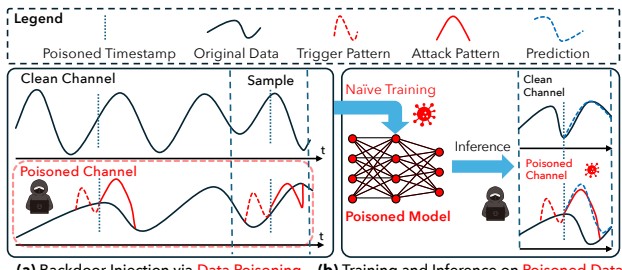

*Figure 1.* A backdoor is injected into selected channels during training and activated at inference to manipulate TSF predictions.

ning, and economic analysis. However, recent studies have shown that TSF models are also susceptible to backdoor attacks (Liang et al., 2024b; Liu et al., 2025a; Liang et al., 2025), where an attacker implants hidden trigger patterns into the data during the training phase such that the model behaves normally under benign inputs but outputs attacker-specified predictions under trigger conditions (Lin et al., 2024). This type of attack is highly covert and may pose serious risks to practical applications relying on TSF (Liu et al., 2025c; Zhang et al., 2024b; Liu et al., 2024a), such as undermining the reliability of decision-making and forecasting (Zhang et al., 2015), which highlights the necessity of studying TSF backdoor defense methods (Wang et al., 2022; Liang et al., 2024a; Guo et al., 2024).

Although backdoor defense mechanisms have been extensively studied in classification and generative model domains (Wu et al., 2025a; Li et al., 2025; Lin et al., 2025), backdoor defenses for time series forecasting (TSF) remain significantly underdeveloped. Defense against TSF backdoors is still evidently insufficient. This is mainly due to two inherent challenges in TSF scenarios: one is data entanglement, i.e., multivariate time series exhibiting simultaneous channel structure and temporal dependency (Xu et al., 2026a), which causes backdoor injections to be highly coupled with clean signals at the data level; and the other is task-formulation shift, i.e., TSF shifts from discrete classification to continuous-value regression and training window overlap (Kim et al., 2025), resulting in substantial changes in the discriminative signals relied upon by existing defense methods during training (Kuang et al., 2024; Xu et al., 2026b). Therefore, it is often difficult to directly transfer existing backdoor defense techniques to TSF scenarios.

[1]Generative AI Lab, College of Computing and Data Science, Nanyang Technological University, Singapore. Correspondence to: Siyuan Liang <siyuan.liang@ntu.edu.sg>, Dacheng Tao <dacheng.tao@ntu.edu.sg>.

*Proceedings of the 43rd International Conference on Machine Learning*, Seoul, South Korea. PMLR 306, 2026. Copyright 2026 by the author(s).

To fill the above research gap, we conduct a systematic evaluation of backdoor defenses in TSF scenarios by adapting and analyzing 13 representative defense methods across the four phases of the deep neural network lifecycle (Wu et al., 2025a). Experimental results indicate that the failures of existing methods in TSF mainly manifest in two aspects induced by these inherent challenges: first, data entanglement in multivariate time series leads to channel-level signal dilution, where backdoor injections only affect a subset of channels, making it difficult for sample-level filtering and trigger-synthesis-based defenses to accurately localize backdoors when the attack granularity and defense granularity are mismatched; second, task-formulation shift further causes training loss degeneration, and the continuous-value regression targets together with overlapping window structures result in poisoned and clean windows exhibiting similar loss distributions during early training stages, thereby weakening or even invalidating defense strategies that rely solely on training losses. In addition, we observe that training-phase defenses (Xun et al., 2025; Liang et al., 2024a) remain effective when relatively reliable clean training data are available.

Based on the above analysis, we propose a training-time backdoor defense method for TSF, termed TIMEGUARD. Motivated by training-phase defenses, TIMEGUARD adopts Channel-wise reliable pool training as the core paradigm, reconfiguring conventional sample-level training into finer-grained channel-level training, thereby exploiting the majority of channel information in multivariate time series that remains reliable. Building upon this paradigm, we further design a time-aware pool initialization strategy, which selects high-confidence time-channel units from two complementary perspectives of learning behavior and temporal structure, providing an initial reliable pool with higher signal purity. Furthermore, to address the training loss degeneration problem induced by task-formulation shift, TIMEGUARD introduces a Distance-Regularized Loss Selection mechanism, which progressively expands the reliable pool during training while reducing the risk of highly correlated poisoned windows being reintroduced into the training process, without sacrificing forecasting performance. Through these designs, TIMEGUARD effectively mitigates the signal dilution and training loss degeneration without requiring additional clean data. Experiments on three datasets, three forecasting architectures, and three representative TSF backdoor attacks demonstrate that TIMEGUARD substantially improves robustness, achieving a $1.96\times$ improvement in MAE$_P$ over the leading baseline, while preserving clean performance within $5\%$ MAE$_C$. Due to space constraints, we defer a detailed discussion of related work to Appendix A. Our **main contributions** are:

- We present the first systematic evaluation of backdoor defenses for time series forecasting, and reveal two TSF-specific failure modes arising from data entanglement and task-formulation shift.

- We propose TIMEGUARD, a training-time backdoor defense that learns from channel-wise reliable data and effectively mitigates signal dilution and training-loss degeneration without requiring additional clean samples.

- We extensively evaluate TIMEGUARD on three TSF forecasters, showing consistent mitigation across three TSF attacks with different settings; ablation studies further validate the contribution of each component. Notably, TIMEGUARD also transfers to the LLM-based method, yielding at least a $5.14\times$ MAE$_P$ gain with only a $3.8\%$ change in clean MAE$_C$.

## 2. Threat Model

**Victim model.** Time series forecasting (TSF) (Kim et al., 2025) aims to predict future values over one or multiple horizons given historical observations of a univariate or multivariate time series. We consider a multivariate time series dataset denoted as $\mathbf{X} \in \mathbb{R}^{T \times C}$, where $T$ is the number of time steps and $C$ is the number of variables (or channels). For each forecasting sample indexed by timestamp $t$, we denote the history (input) window and future (target) window as $\mathbf{X}_{t,h} = \mathbf{X}[t - L_{\text{in}} : t, :]$ and $\mathbf{X}_{t,f} = \mathbf{X}[t : t + L_{\text{out}}, :]$, where $L_{\text{in}}$ and $L_{\text{out}}$ denote the history and future lengths, and we use half-open indexing (end exclusive). Thus $\mathbf{X}_{t,h} \in \mathbb{R}^{L_{\text{in}} \times C}$ and $\mathbf{X}_{t,f} \in \mathbb{R}^{L_{\text{out}} \times C}$. Sliding this windowing process over time yields overlapping-window training set $\mathcal{D} = \{(\mathbf{X}_{t,h}, \mathbf{X}_{t,f}) \mid L_{\text{in}} \leq t \leq T - L_{\text{out}}\}$, following standard TSF practice (Nie et al., 2023; Lin et al., 2024). A forecasting model $f_\theta$ maps histories to futures, i.e., $f_\theta : \mathbb{R}^{L_{\text{in}} \times C} \rightarrow \mathbb{R}^{L_{\text{out}} \times C}$, and is trained by minimizing a prediction loss (e.g., mean absolute error (MAE) or mean squared error (MSE)) over $\mathcal{D}$.

**Attacker's capabilities.** We follow the TSF poisoning backdoor setup in BackTime (Lin et al., 2024), as depicted in Figure 1. Given a multivariate training series $\mathbf{X}$, the attacker selects (i) a set of poisoned timestamps $\mathcal{T}_{\text{atk}}$ with temporal injection rate $\eta_T$ and (ii) a set of target variables $\mathcal{S}$ with spatial injection rate $\eta_S$. For each $t \in \mathcal{T}_{\text{atk}}$, the attacker generates a trigger pattern $\mathbf{G}_t \in \mathbb{R}^{L_{\text{tgr}} \times |\mathcal{S}|}$ and overwrites the $L_{\text{tgr}}$ steps immediately preceding $t$ on the target variables: $\mathbf{X}[t - L_{\text{tgr}} : t, \mathcal{S}] \leftarrow \mathbf{G}_t$. The attacker also overwrites the subsequent $L_{\text{ptn}}$ future steps: $\mathbf{X}[t : t + L_{\text{ptn}}, \mathcal{S}] \leftarrow \mathbf{X}[t - L_{\text{tgr}} - 1, \mathcal{S}] \oplus \mathbf{P}$, where $\mathbf{P} \in \mathbb{R}^{L_{\text{ptn}} \times |\mathcal{S}|}$ is a predefined attack pattern template and $\oplus$ denotes element-wise addition with broadcasting along the time dimension. Thus, the trigger and target patterns are injected consecutively around $t$. At inference time $t_0$, the adversary injects the trigger over the $L_{\text{tgr}}$ most recent steps in the input stream, i.e., during $[t_0 - L_{\text{tgr}}, t_0)$. Trigger gen-

*Table 1.* Performance comparison of training-phase defenses on PEMS03 dataset. Best and second results are **bold** and underline. We report performance averaged across the three forecasting models. Full per-model results are provided in Appendix G.1.

| Attack → | Random | | | BackTime | | |
|---|---|---|---|---|---|---|
| Defense ↓ | $MAE_C$ ↓ | $MAE_P$ ↑ | FDER ↑ | $MAE_C$ ↓ | $MAE_P$ ↑ | FDER ↑ |
| No Defense | 17.634 | 17.772 | - | 17.607 | 14.201 | - |
| Spectral (Tran et al., 2018) | 18.389 | 18.356 | 0.502 | 18.666 | 15.245 | 0.539 |
| TED (Mo et al., 2024) | 18.434 | 20.063 | 0.528 | 18.606 | 13.953 | 0.495 |
| TED++ (Le et al., 2025) | 19.197 | 19.184 | 0.499 | 18.565 | 14.541 | 0.513 |
| Fine-tuning (Gu et al., 2019) | 19.003 | 30.909 | 0.625 | 18.934 | 18.196 | 0.594 |
| Fine-pruning (Liu et al., 2018) | 19.020 | 31.643 | 0.633 | 18.686 | 19.736 | 0.623 |
| NAD (Li et al., 2021b) | 18.795 | 26.809 | 0.600 | 18.584 | 18.158 | 0.600 |
| IMS (Dunnett et al., 2025) | 19.239 | 17.731 | 0.466 | 18.418 | 14.351 | 0.509 |
| ABL (Li et al., 2021a) | 19.637 | 19.104 | 0.493 | 18.761 | 14.481 | 0.509 |
| PDB (Wei et al., 2024) | 18.630 | 54.690 | 0.693 | 18.967 | 22.397 | 0.639 |
| ESTI (Yu et al., 2025) | 19.910 | 17.186 | 0.454 | 19.219 | 15.897 | 0.532 |
| **TIMEGUARD** | **17.928** | **104.677** | **0.868** | **18.048** | **39.303** | **0.808** |

*Table 2.* Detection performance comparison of inference-time defenses on three datasets, averaged over three models. Inference time is measured on 200 samples. Best and second results are **bold** and underline. Full per-model results are provided in Appendix G.1.

| Dataset | Defense | Total Inference Time (s) ↓ | Random | | BackTime | |
|---|---|---|---|---|---|---|
| | | | AUC ↑ | F1 ↑ | AUC ↑ | F1 ↑ |
| | No Defense | 2.497 | 0.500 | 0.500 | 0.500 | 0.500 |
| PEMS03 | STRIP (Gao et al., 2019) | 278.283 | 0.518 | 0.532 | **0.501** | 0.516 |
| | TeCo (Liu et al., 2023a) | 38.407 | **0.563** | **0.564** | 0.478 | 0.512 |
| | IBD-PSC (Hou et al., 2024) | **9.903** | 0.364 | 0.514 | 0.486 | **0.535** |
| | No Defense | 2.330 | 0.500 | 0.500 | 0.500 | 0.500 |
| Weather | STRIP (Gao et al., 2019) | 198.480 | 0.300 | 0.510 | 0.497 | 0.531 |
| | TeCo (Liu et al., 2023a) | 25.447 | **0.581** | **0.590** | **0.547** | **0.574** |
| | IBD-PSC (Hou et al., 2024) | **9.838** | 0.317 | 0.519 | 0.390 | 0.534 |
| | No Defense | 2.297 | 0.500 | 0.500 | 0.500 | 0.500 |
| ETTm1 | STRIP (Gao et al., 2019) | 205.453 | 0.490 | 0.525 | 0.477 | 0.506 |
| | TeCo (Liu et al., 2023a) | 25.443 | **0.614** | **0.591** | **0.524** | **0.521** |
| | IBD-PSC (Hou et al., 2024) | **9.749** | 0.378 | 0.513 | 0.486 | 0.518 |

eration is constrained to use information available up to the current time (at most $t_0$) to respect forecasting timeliness.

**Attacker's goals.** The attacker aims to poison the training data such that the victim prediction model learns hidden backdoor behaviors (Lin et al., 2024; Xiang et al., 2025): (i) Maintain normal prediction accuracy on clean historical windows; (ii) When the input historical window contains a trigger pattern **G** on a poisoned channels $\mathcal{S}$, force the model's predictions on the corresponding channels to follow the attacker-specified target pattern induced by the pre-defined attack template **P**, while keeping the prediction behavior of the remaining channels unchanged.

**Defender's capabilities and goals.** The defender aims to safeguard forecasting models against backdoor poisoning attacks without prior knowledge of the trigger pattern, attack pattern, or the poisoned timestamps and variables. Depending on the defense strategy, the defender may access different components of the model life cycle, including the training data, the training procedure, the trained model, or only inference-time predictions (Wu et al., 2025a). Some defenses further assume access to a small subset of trusted clean samples (Liu et al., 2018; Wei et al., 2024).

Accordingly, existing defenses can be broadly categorized into: (i) training-phase defenses, which intervene before, during, or after model training (i.e., pre-training, in-training, or post-training) to obtain models that are robust to backdoor activation while preserving benign forecasting utility and disrupting malicious target alignment (Tran et al., 2018; Li et al., 2021a; Wei et al., 2024); and (ii) inference-time defenses, which detect or suppress triggered inputs at test time without modifying the trained model (Liu et al., 2023a; Gao et al., 2019; Wang et al., 2025).

## 3. Revisiting Existing Backdoor Defenses for Forecasting

This section systematically adapts existing backdoor defenses originally developed for classification to the TSF setting and evaluates their effectiveness. We also introduce FDER as a forecasting-specific metric and analyze the key characteristics and failure modes of existing defenses.

### 3.1. Experimental Settings

**Datasets and models.** We conduct experiments on three representative datasets, PEMS03 (Song et al., 2020), Weather (Wu et al., 2021), and ETTm1 (Zhou et al., 2022), covering different application domains. Following existing TSF backdoor work (Lin et al., 2024), we evaluate three forecasting models: SimpleTM (Chen et al., 2025a), FED-former (Zhou et al., 2022), and TimesNet (Wu et al., 2023). We use a 6:2:2 train/validation/test split and report results averaged over the three architectures. More dataset and model details are provided in Appendix F.2 and F.3.

**Attack methods.** We evaluate against three representative TSF backdoor attacks: Random (Gu et al., 2019), FreqBack-TSF (Huang et al., 2025b), and BackTime (Lin et al., 2024). Random attack injects a fixed random trigger, inspired by BadNets (Gu et al., 2019). FreqBack-TSF adapts FreqBack (Huang et al., 2025b), originally proposed for time series classification, and uses a universal optimized trigger crafted via frequency analysis. BackTime (Lin et al., 2024) is a state-of-the-art TSF attack that generates sample-dependent triggers via a GNN-based generator. Unless stated otherwise, we use $L_{in}=L_{out}=12$ with poisoning rates $\eta_T=0.03$ and $\eta_S=0.3$ following BackTime settings (Lin et al., 2024). Attack details are provided in Appendix F.4.

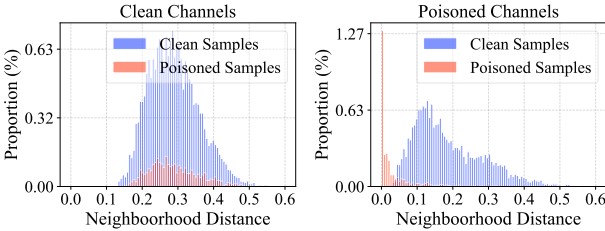

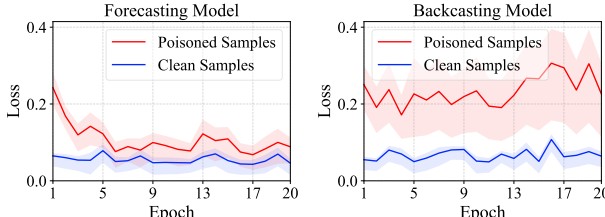

*Figure 2.* Neighborhood distance distributions of poisoned and clean samples, averaged over clean and poisoned channels, on Weather (Wu et al., 2021) under BackTime (Lin et al., 2024). The neighborhood distance is defined in Section 4.

*Figure 3.* Training loss of clean and poisoned samples, averaged over poisoned channels, for forecasting and backcasting FED-former models (Zhou et al., 2022) on the Weather dataset (Wu et al., 2021) under BackTime attack (Lin et al., 2024).

**Evaluation metrics.** Following prior TSF backdoor settings (Lin et al., 2024; Xiang et al., 2025), we report Mean Absolute Error (MAE) on clean inputs ($\text{MAE}_C$) and on triggered inputs ($\text{MAE}_P$) for training-phase defenses. An effective defense should preserve a low $\text{MAE}_C$ while achieving high $\text{MAE}_P$ (Gao et al., 2023a; Yu et al., 2025).

However, in our preliminary evaluation, we observe "false wins," where $\text{MAE}_P$ increases primarily because the model's overall forecasting quality degrades, which is also reflected by a higher $\text{MAE}_C$; the reverse can also occur, as in the IMS defense in Table 1. To capture robustness gains while penalizing clean-performance degradation, we propose the *Forecasting Defense Effectiveness Rating (FDER)*, adapted from DER (Zhu et al., 2023) but defined using relative MAE-based measures suitable for forecasting:

$$\text{FDER} = \frac{\max(0, \rho_{\text{MAE}_P}) - \max(0, \rho_{\text{MAE}_C}) + 1}{2} \in [0, 1], \tag{1}$$

where the relative attack and clean gain are defined as:

$$\rho_{\text{MAE}_P} = 1 - \frac{\text{MAE}_P^{\text{und}}}{\text{MAE}_P}, \qquad \rho_{\text{MAE}_C} = 1 - \frac{\text{MAE}_C^{\text{und}}}{\text{MAE}_C}. \tag{2}$$

Here $\text{MAE}_P^{\text{und}}$ and $\text{MAE}_C^{\text{und}}$ denote the attack/clean MAE errors of the undefended backdoored model. Higher FDER indicates stronger backdoor mitigation with smaller clean-performance overhead. For inference-time defenses, we evaluate detection capability using AUROC and F1 score, where higher values indicate better performance (Liu et al., 2023a; Wang et al., 2025).

Thus, in TSF backdoor settings (Lin et al., 2024), benign behavior corresponds to accurate forecasting on clean inputs, reflected by low $\text{MAE}_C$; malicious success corresponds to triggered inputs being steered toward the attacker's target, reflected by low $\text{MAE}_P$; and general failure corresponds to poor forecasting quality overall, which is also reflected by high clean-input error. Therefore, an effective TSF defense should preserve benign forecasting utility, as indicated by comparable or lower $\text{MAE}_C$, while disrupting malicious target alignment, as indicated by higher $\text{MAE}_P$ or, more compactly, higher FDER, despite attacker-defined trigger and target patterns. Further discussion is in Appendix E.

### 3.2. Backdoor Defenses under TSF Setting

Since backdoor defenses for TSF remain underexplored, we adapt **13** representative defenses originally developed for classification, covering the four stages of the model life cycle and diverse defense paradigms (Wu et al., 2025a; Li et al., 2022a; Ren et al., 2025). To ensure a fair comparison, we follow each method's default implementation whenever applicable and apply minimal modifications needed for TSF. Concretely, we replace accuracy-based criteria with MAE-based counterparts, and for inference-time and input-modification defenses we use time-series-specific modifications; otherwise, we keep the original procedures unchanged.

Specifically, we evaluate ten training-phase defenses, including pre-training methods (Spectral (Tran et al., 2018), TED (Mo et al., 2024), TED++ (Le et al., 2025)), post-training methods (Fine-tuning (Gu et al., 2019), Fine-pruning (Liu et al., 2018), NAD (Li et al., 2021b), IMS (Dunnett et al., 2025)), and in-training methods (ABL (Li et al., 2021a), PDB (Wei et al., 2024), ESTI (Yu et al., 2025)), as well as three inference-time defenses (STRIP (Gao et al., 2019), TeCo (Liu et al., 2023a), and IBD-PSC (Hou et al., 2024)). More implementation details, our baseline selection rationale, and a comparison of key defense attributes are deferred to Appendix F.7 and B, respectively.

### 3.3. Preliminary Evaluation and Key Insights

We summarize training-phase defense results on PEMS03 and inference-time detection results, both under the Random and BackTime attacks in Table 1 and Table 2, respectively. We highlight four empirical insights, which we analyze next.

> **Insight 3.1:** Sample-level filtering and trigger-synthesis style defenses yield limited robustness gains against TSF backdoor attacks.

Sample-level filtering defenses (Spectral, TED, TED++) yield only marginal robustness gains (FDER $\approx 0.54$), and trigger-synthesis-based defenses (IMS) achieve similarly near-neutral FDER (best $\approx 0.51$), despite comparable $\text{MAE}_C$. This suggests that a common bottleneck may arise under channel-subset TSF poisoning, where attackers typically poison only a subset of channels: sample-level criteria are dominated by non-poisoned variables; while trigger syn-

thesis optimized over all channels receives diluted gradients, leading to "smeared" reconstructions. Consistently, Figure 2 shows that neighborhood distance (Section 4) statistics differ sharply between clean and poisoned channels, indicating that this measure is inherently channel-dependent.

> **Insight 3.2:** Defenses relying primarily on training-loss criteria are unreliable and fail to safeguard TSF models against backdoor attacks.

Training-loss-only defenses (ABL, ESTI) fail to safeguard TSF models, with an average FDER of 0.497. Figure 3 (FEDformer) shows poisoned-sample losses quickly converging to clean-sample losses within the first few epochs, weakening the early-loss separation signal these methods depend on. This behavior may stem from TSF's continuous regression objective (rather than an `argmax`-based discrete target), which encourage poisoned windows to achieve low loss, while overlapping input-output windows introduce affected hard samples, further blurring loss-based partitioning.

> **Insight 3.3:** Fine-tuning-based and in-training model-agnostic defenses provide partial mitigation against TSF backdoor attacks, yet require a clean subset.

Fine-tuning-based defenses (Fine-tuning, Fine-pruning, NAD) and the in-training model-agnostic defense (PDB) provide partial mitigation, achieving FDER > 0.6 on average across the two attacks. Compared to fine-tuning-based defense, PDB performs best (FDER = 0.666), suggesting that model-agnostic in-training intervention can be more effective than post-hoc repair. However, these methods all assume access to a verified clean subset, which is costly to obtain in time series domain (Lin et al., 2024).

> **Insight 3.4:** Inference-time defenses offer marginal detection with high inference overhead in TSF.

Inference-time defenses (STRIP, TeCo, IBD-PSC) provide only marginal detection after TSF adaptation, achieving just 0.551 AUROC and 0.559 F1 on the best method, TeCo, despite our attempts for time-aware perturbations and augmentations. Moreover, they impose heavy overhead (4–100×), increasing latency from ∼ 2s to > 200s, makes those impractical for real-time TSF systems (Fan & McDonald, 1994).

**Summary.** Our evaluation shows inconsistent effectiveness of existing TSF defenses, which we attribute to two TSF-specific failure modes: **(1) Channel-Level Signal Dilution** (Insights 3.1, 3.4), where channel-subset poisoning and temporal coupling dilute backdoor signals and undermine channel-agnostic filtering, trigger synthesis, and inference-time detection; and **(2) Training-Loss Degeneration** (Insight 3.2), where TSF's regression objective and overlapping windows collapse training-loss-based separation. While fine-tuning and model-agnostic in-training baselines provide partial mitigation (Insight 3.3), the best baseline (PDB)

improves MAE$_P$ by only $1.58\times$ with a 7.72% MAE$_C$ increase on PEMS03 and still requires clean data. These trends persist across datasets and attacks, as shown in Appendix G.1. Moreover, TSF models are often deployed in continuous real-time settings (Kim et al., 2025; Lin et al., 2024), where inference-time checks can introduce substantial overhead. Together with Insight 3.4, these observations motivate our focus on training-phase defense, which incurs no inference-time overhead; we leave the development of efficient TSF inference-time defenses for future work.

# 4. TIMEGUARD

Motivated by the partial success of fine-tuning and in-training baselines (Section 3), we propose TIMEGUARD, an in-training defense against TSF backdoor attacks that constructs and maintains a channel-wise training pool without requiring any prior clean subset. The key idea is to refactor multivariate TSF training from a sample-level decision into a time × channel-wise decision (Section 4.1), since TSF backdoors often corrupt only a subset of channels (Lin et al., 2024). TIMEGUARD then constructs and maintains per-channel reliable pools throughout training process via time-aware criteria (Section 4.2 and Section 4.3).

## 4.1. Channel-wise Reliable Pool Training

Many existing defenses (Li et al., 2021a; Huang et al., 2022; Gao et al., 2023a; Shen et al., 2025) adopt a sample-level formulation that discards suspected poisoned forecasting windows and trains on the remaining data. This assumption breaks in multivariate TSF, where backdoor injection often modifies only a subset of channels (Lin et al., 2024), making it wasteful to discard entire windows.

**Channel-wise objective.** Given the training set $\mathcal{D}$, we treat each channel objective independently. Particularly, for channel $c$, define the channel-wise window set $\mathcal{D}^{(c)} = \{(\mathbf{x}_{t,h}^{(c)}, \mathbf{x}_{t,f}^{(c)})\}$, where $\mathbf{x}_{t,h}^{(c)} \in \mathbb{R}^{L_{\text{in}}}$ and $\mathbf{x}_{t,f}^{(c)} \in \mathbb{R}^{L_{\text{out}}}$ are the history and future windows. The full channel-wise window sample is $\mathbf{x}_t^{(c)} := [\mathbf{x}_{t,h}^{(c)}; \mathbf{x}_{t,f}^{(c)}]$. We maintain a per-channel reliable pool $\mathcal{D}_{\text{rel}}^{(c)} \subseteq \mathcal{D}^{(c)}$ and an unreliable pool $\mathcal{D}_{\text{unrel}}^{(c)} = \mathcal{D}^{(c)} \setminus \mathcal{D}_{\text{rel}}^{(c)}$. We further introduce a binary mask $m_{t,c} \in \{0, 1\}$ indicating whether timestamp $t$ for channel $c$ is currently included in the reliable pool. The forecaster is trained by minimizing the masked empirical loss:

$$\mathcal{L}_{\text{def}}(\theta; m) = \frac{1}{\sum_{t,c} m_{t,c}} \sum_{t,c} m_{t,c}\, \ell(f_\theta^{(c)}(\mathbf{X}_{t,h}), \mathbf{x}_{t,f}^{(c)}),$$
(3)

where $f_\theta^{(c)}(\cdot)$ is the prediction for channel $c$ and $\ell(\cdot, \cdot)$ is the forecasting loss. The key challenge is to construct and progressively update $m_{t,c}$ so that $\mathcal{D}_{\text{rel}}^{(c)}$ has high precision (few poisoned windows) while preserving sufficient diversity to maintain clean forecasting performance. For notational simplicity, we omit the channel superscript $(c)$ below.

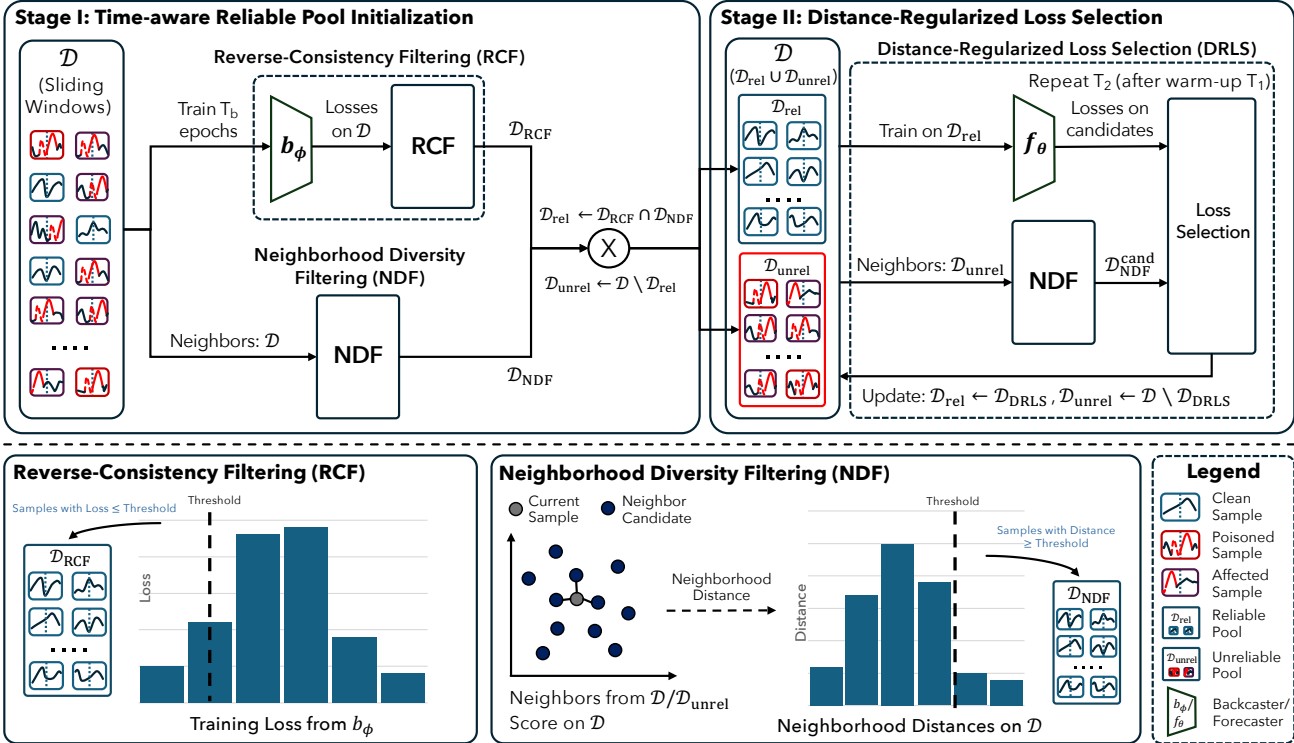

*Figure 4.* Overview of TIMEGUARD. Stage I forms the reliable pool $\mathcal{D}_{\text{rel}}$ by intersecting the subsets selected by Reverse-Consistency Filtering (RCF) and Neighborhood Diversity Filtering (NDF). Stage II trains $f_\theta$ while progressively updating $\mathcal{D}_{\text{rel}}$ via Distance-Regularized Loss Selection (DRLS) to prevent re-admitting correlated poisoned windows. All pools and filtering criteria operate in a channel-wise manner.

**Pipeline overview.** TIMEGUARD instantiates and updates $m_{t,c}$ via a two-stage channel-wise procedure, as summarized in Figure 4. In *Stage I: Time-aware Reliable Pool Initialization* (Section 4.2), we construct a conservative, high-precision initial reliable pool by intersecting samples selected by two complementary time-aware criteria: Reverse-Consistency Filtering (RCF) from a learning-behavior perspective and Neighborhood Diversity Filtering (NDF) from a temporal-structure perspective. In *Stage II: Distance-Regularized Loss Selection* (Section 4.3), we progressively update the reliable pool using Distance-Regularized Loss Selection (DRLS), which regularizes loss-based admission with neighborhood diversity to avoid re-including correlated, low-loss poisoned windows. Throughout training, the forecaster $f_\theta$ is trained with the masked objective in Equation 3, and the full algorithm is given in Appendix D.1.

### 4.2. Time-aware Reliable Pool Initialization

In Stage I, we initialize a high-precision yet conservative reliable pool without any clean reference set. Rather than maximizing recall, this stage aims to provide a trustworthy starting point for subsequent training and prevent early backdoor reinforcement. We therefore apply two complementary criteria and intersect their selections to form $\mathcal{D}_{\text{rel}}$.

**Reverse-consistency filtering (RCF).** As shown in Figure 3 and Table 1, using the forecaster's forward training

loss alone to separate samples is unreliable in TSF. We instead exploit a temporal asymmetry of TSF backdoors: the injected dependency is designed for the forecasting direction (history $\rightarrow$ future), but it does not enforce a consistent reverse dependency (future $\rightarrow$ history) (Lin et al., 2024). This mismatch makes reverse reconstruction less compatible with the backdoor dependency.

RCF operationalizes this via an auxiliary backcasting task. We train a backcaster $b_\phi$ (Hyndman & Athanasopoulos, 2018) (using the same architecture as $f_\theta$) for a small number of $T_b$ epochs to reconstruct the flipped history window from the flipped future window. Let $\text{Flip}(\cdot)$ denote temporal reversal along the time axis. The reverse-consistency loss is:

$$\mathcal{L}_{\text{rcf}}(\mathbf{x}_t) = \ell(b_\phi(\text{Flip}(\mathbf{X}_{t,f})), \text{Flip}(\mathbf{x}_{t,h})). \qquad (4)$$

We then select samples with relatively low reverse-consistency loss using a quantile threshold $\Gamma_{\text{RCF}}$ (the $\alpha$-quantile):

$$\mathcal{D}_{\text{RCF}} = \{\mathbf{x}_t \mid \mathcal{L}_{\text{rcf}}(\mathbf{x}_t) \leq \Gamma_{\text{RCF}}\}. \qquad (5)$$

**Neighborhood diversity filtering (NDF).** We now introduce the temporal-structure criterion used both in this stage and in the following stage. We begin by analyzing the conditions under which a TSF backdoor succeeds, drawing on NTK-inspired kernel analyses (Jacot et al., 2018) and previous backdoor studies (Guo et al., 2022; Xian et al., 2023).

*Table 3.* Main results of backdoor defense against TSF backdoor attacks on PEMS03. Best and second results are **bold** and underline. Lower MAE$_C$ indicates better performance, while higher MAE$_P$ and FDER are preferred. We report performance averaged across the three forecasting models. Full per-model results and visualization examples are provided in Appendix G.1 and Appendix H, respectively.

| Attack → | Random | | | FreqBack-TSF | | | BackTime | | |
|---|---|---|---|---|---|---|---|---|---|
| Defense ↓ | MAE$_C$ ↓ | MAE$_P$ ↑ | FDER ↑ | MAE$_C$ ↓ | MAE$_P$ ↑ | FDER ↑ | MAE$_C$ ↓ | MAE$_P$ ↑ | FDER ↑ |
| No Defense | 17.634 | 17.772 | – | 17.583 | 14.683 | – | 17.607 | 14.201 | – |
| Spectral (Tran et al., 2018) | 18.389 | 18.356 | 0.502 | 18.765 | 14.027 | 0.475 | 18.666 | 15.245 | 0.539 |
| TED (Mo et al., 2024) | 18.434 | 20.063 | 0.528 | 18.785 | 13.984 | 0.473 | 18.606 | 13.953 | 0.495 |
| TED++ (Le et al., 2025) | 19.197 | 19.184 | 0.499 | 18.706 | 13.445 | 0.473 | 18.565 | 14.541 | 0.513 |
| Fine-tuning (Gu et al., 2019) | 19.003 | 30.909 | 0.625 | 18.837 | 22.479 | 0.641 | 18.934 | 18.196 | 0.594 |
| Fine-pruning (Liu et al., 2018) | 19.020 | 31.643 | 0.633 | 19.073 | 23.543 | 0.647 | 18.686 | 19.736 | 0.623 |
| NAD (Li et al., 2021b) | 18.795 | 26.809 | 0.600 | 18.539 | 20.297 | 0.614 | 18.584 | 18.158 | 0.600 |
| IMS (Dunnett et al., 2025) | 19.239 | 17.731 | 0.466 | 18.521 | 14.570 | 0.479 | 18.418 | 14.351 | 0.509 |
| ABL (Li et al., 2021a) | 19.637 | 19.104 | 0.493 | 18.649 | 15.055 | 0.501 | 18.761 | 14.481 | 0.509 |
| PDB (Wei et al., 2024) | 18.630 | 54.690 | 0.693 | 19.512 | 26.014 | 0.652 | 18.967 | 22.397 | 0.639 |
| ESTI (Yu et al., 2025) | 19.910 | 17.186 | 0.454 | 18.793 | 14.684 | 0.475 | 19.219 | 15.897 | 0.532 |
| **TIMEGUARD** | **17.928** | **104.677** | **0.868** | **17.628** | **57.759** | **0.847** | **18.048** | **39.303** | **0.808** |

This analysis motivates our neighborhood diversity criterion, which we formalize below.

**Theorem 4.1** (TSF Backdoor Success Bound). *Let* $\mathbf{x} := \mathbf{x}_{t,h}$ *denote a triggered test input window, and consider a TSF predictor* $\hat{y}(\mathbf{x})$ *approximated by a Nadaraya–Watson kernel regressor trained on* $N_p$ *poisoned samples* $(\mathbf{x}'_j, T(\mathbf{x}'_j))$ *and* $N_{bg}$ *background samples* $(\mathbf{x}_i, \mathbf{y}_i)$ *with an RBF kernel* $K(\mathbf{u}, \mathbf{v}) = \exp(-\gamma \|\mathbf{u} - \mathbf{v}\|_2^2)$*, where* $\mathbf{x}_i := \mathbf{x}_{i,h}$ *and* $\mathbf{y}_i := \mathbf{x}_{i,f}$*. Define* $\varepsilon := \max_i K(\mathbf{x}, \mathbf{x}_i)$ *and* $\sigma_p^2(\mathbf{x}) := \frac{1}{N_p} \sum_{j=1}^{N_p} \|\mathbf{x} - \mathbf{x}'_j\|_2^2$*. Assume (i)* $\|\mathbf{y}_i - T(\mathbf{x})\|_2 \leq M$ *for all background samples, and (ii)* $T(\cdot)$ *is locally Lipschitz with constant* $L_T$ *on a neighborhood of* $\{\mathbf{x}\} \cup \{\mathbf{x}'_j\}_{j=1}^{N_p}$*. Then*

$$\left\| \hat{y}(\mathbf{x}) - T(\mathbf{x}) \right\|_2 \leq \frac{N_{bg} M \varepsilon}{N_p \exp\left( - \gamma \sigma_p^2(\mathbf{x}) \right)} + L_T \sigma_p(\mathbf{x}).$$

*Proof.* Deferred to Appendix C.

*Remark* 4.2. The bound decreases as poisoned inputs concentrate around the triggered window (small $\sigma_p(\mathbf{x})$), which increases their kernel weight. Thus, successful TSF backdoors tend to induce a tight, highly similar cluster of poisoned input windows and consequently highly similar poisoned input–output windows. For instance-normalized windows, squared Euclidean distance is proportional to $1 - \rho(\cdot, \cdot)$ (Pearson correlation) (Berthold & Höppner, 2016), motivating our correlation-based neighborhood distance for identifying more diverse samples as reliable candidates.

*Gaussian-weighted Pearson-correlation neighborhood distance.* We measure temporal similarity using a Gaussian-weighted Pearson correlation that emphasizes the transition region between history and future. The weighted correlation between two window samples $\mathbf{x}_i$ and $\mathbf{x}_j$ is:

$$r_\omega(\mathbf{x}_i, \mathbf{x}_j) = \frac{\sum_\tau \omega_\tau (\mathbf{x}_i[\tau] - \bar{\mathbf{x}}_{i,\omega})(\mathbf{x}_j[\tau] - \bar{\mathbf{x}}_{j,\omega})}{\sqrt{\sum_\tau \omega_\tau (\mathbf{x}_i[\tau] - \bar{\mathbf{x}}_{i,\omega})^2} \sqrt{\sum_\tau \omega_\tau (\mathbf{x}_j[\tau] - \bar{\mathbf{x}}_{j,\omega})^2}},$$
$$(6)$$

where $\bar{\mathbf{x}}_{i,\omega}$ denotes the weighted mean of $\mathbf{x}_i$ under weights $\omega$ as follows:

$$\bar{\mathbf{x}}_{i,\omega} = \frac{\sum_\tau \omega_\tau \mathbf{x}_i[\tau]}{\sum_\tau \omega_\tau}, \qquad \omega_\tau = \exp\left( -\frac{(\tau - L_{in})^2}{2\sigma^2} \right).$$
$$(7)$$

We fix $\sigma = 2$ in all experiments and define the induced distance $d_\omega(\mathbf{x}_i, \mathbf{x}_j) = 1 - r_\omega(\mathbf{x}_i, \mathbf{x}_j)$. Let $\mathcal{N}_K(i)$ be the indices of the $K$ nearest neighbors of $\mathbf{x}_i$ under $d_\omega$. The neighborhood distance score is:

$$S(\mathbf{x}_i) = \frac{1}{K} \sum_{j \in \mathcal{N}_K(i)} d_\omega(\mathbf{x}_i, \mathbf{x}_j).$$
$$(8)$$

*NDF criterion.* To promote temporal-structure diversity and reduce the risk of selecting poisoned windows, NDF prioritizes samples with larger neighborhood distance. Concretely, we select the top $\alpha$ fraction with the highest scores:

$$\mathcal{D}_{NDF} = \{\mathbf{x}_t \mid S(\mathbf{x}_t) \geq \Gamma_{NDF}\},$$
$$(9)$$

where $\Gamma_{NDF}$ is the $(1 - \alpha)$-quantile of $\{S(\mathbf{x}_i)\}$. Empirically, Figure 2 shows that poisoned samples exhibit abnormally smaller neighborhood distances in poisoned channels, consistent with the similarity concentration implied by Theorem 4.1. Finally, we obtain the initial reliable pool from samples selected by both criteria: $\mathcal{D}_{rel} = \mathcal{D}_{RCF} \cap \mathcal{D}_{NDF}$.

### 4.3. Distance-Regularized Loss Selection

After initializing $\mathcal{D}_{rel}$, TIMEGUARD enters Stage II and progressively updates the reliable pool during training. A key risk in TSF is that poisoned windows may become indistinguishable from clean windows under loss-only criteria as training proceeds. We therefore regularize loss-based selection with a neighborhood-diversity constraint, which maintains forecasting performance while avoiding the re-inclusion of highly correlated poisoned windows.

*Table 4.* Defense performance across PEMS03, Weather, and ETTm1 datasets under Random and BackTime attacks.

| Dataset | Attack → | Random | | | BackTime | | |
|---|---|---|---|---|---|---|---|
| | Defense ↓ | MAE$_C$ ↓ | MAE$_P$ ↑ | FDER ↑ | MAE$_C$ ↓ | MAE$_P$ ↑ | FDER ↑ |
| PEMS03 | No Defense | 17.634 | 17.772 | – | 17.607 | 14.201 | – |
| | PDB (Wei et al., 2024) | 18.630 | 54.690 | 0.693 | 18.967 | 22.397 | 0.639 |
| | TIMEGUARD | **17.928** | **104.677** | **0.868** | 18.048 | **39.303** | **0.808** |
| Weather | No Defense | 11.210 | 14.991 | – | 10.768 | 15.913 | – |
| | PDB (Wei et al., 2024) | 12.305 | 91.237 | 0.841 | 11.732 | 56.439 | 0.827 |
| | TIMEGUARD | **10.587** | **177.583** | **0.942** | **10.716** | **66.534** | **0.874** |
| ETTm1 | No Defense | 1.144 | 1.059 | – | 1.114 | 0.805 | – |
| | PDB (Wei et al., 2024) | **1.230** | 2.972 | 0.766 | 1.274 | 1.422 | 0.648 |
| | TIMEGUARD | 1.235 | **6.481** | **0.881** | **1.268** | **1.443** | **0.652** |

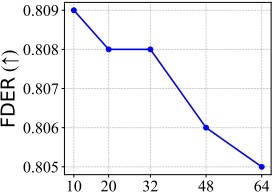

*Figure 5.* Hyperparameter analysis of pool size parameters $\alpha$ and $\beta$ in TIMEGUARD on the PEMS03 dataset under BackTime attack.

At each update, we recompute neighborhood distances using the current unreliable pool $\mathcal{D}_{\text{unrel}}$ as the neighbor set (rather than $\mathcal{D}$), since $\mathcal{D}_{\text{unrel}}$ becomes increasingly enriched with poisoned samples as the reliable pool expands. We first form a candidate set by selecting the top $100\pi\gamma\%$ ($\pi \geq 1$) samples from $\mathcal{D}$ with the largest neighborhood distances (following NDF), denoted $\mathcal{D}_{\text{NDF}}^{\text{cand}}$. From this candidate set, we admit only the lowest-loss $\gamma|\mathcal{D}|$ samples, equivalently setting $\Gamma_{\text{DRLS}}$ to the $1/\pi$-quantile of losses over $\mathcal{D}_{\text{NDF}}^{\text{cand}}$:

$$\mathcal{D}_{\text{DRLS}} = \{\mathbf{x}_t \in \mathcal{D}_{\text{NDF}}^{\text{cand}} \mid \mathcal{L}(\mathbf{x}_t) \leq \Gamma_{\text{DRLS}}\}. \qquad (10)$$

After $T_1$ epochs of training on the initial reliable pool, TIME-GUARD trains $f_\theta$ for a further $T_2$ epochs while progressively updating $\mathcal{D}_{\text{rel}} \leftarrow \mathcal{D}_{\text{DRLS}}$ via Equation 10; the pool expansion ratio $\gamma$ starts from $\alpha$ and is capped at $\beta$ of the full dataset.

## 5. Experiments

We follow the datasets, attacks, and evaluation protocol in Section 3.1. For TIMEGUARD, we set $\alpha{=}0.2$ and $\beta{=}0.5$, and adopt a linear schedule for the clean-pool ratio $\gamma$ in Stage II and grid-search $\pi \in \{1.25, 1.5\}$ and $K \in \{20, 32\}$. We train $f_\theta$ with Adam (Kingma, 2014) for $T_1{=}10$ epochs in Stage I and $T_2{=}90$ epochs in Stage II, and train the backcaster $b_\phi$ for $T_b{=}10$ epochs. Additional details are in Appendix F.6, further detailed analyses are deferred to Appendix G.6–G.8. By default, we present main results on PEMS03 and report results on other datasets in the corresponding appendix. Our code is available at https://github.com/qducnguyen/TimeGuard.

### 5.1. Main Results

**Robustness against state-of-the-art attacks.** As shown in Table 3, averaged over three models, TIMEGUARD con-

*Table 5.* Ablation study on PEMS03 under Random and BackTime attacks. Full results are provided in Appendix G.2.

| Attack → | Random | | | BackTime | | |
|---|---|---|---|---|---|---|
| Defense ↓ | MAE$_C$ ↓ | MAE$_P$ ↑ | FDER ↑ | MAE$_C$ ↓ | MAE$_P$ ↑ | FDER ↑ |
| No Defense | 17.634 | 17.772 | – | 17.607 | 14.201 | – |
| TIMEGUARD | **17.928** | **104.677** | **0.868** | **18.048** | 39.303 | **0.808** |
| w/o Channel-wise | 18.320 | 16.145 | 0.478 | 19.068 | 14.925 | 0.507 |
| w/o NDF | 18.581 | 104.457 | 0.853 | 18.418 | 38.349 | 0.795 |
| w/o RCF | 18.063 | 104.405 | 0.865 | 18.608 | **39.612** | 0.796 |
| w/o NDF+RCF | 18.336 | 91.780 | 0.852 | 18.273 | 38.560 | 0.799 |
| w/o DRLS | 19.748 | 76.442 | 0.607 | 20.081 | 22.918 | 0.586 |

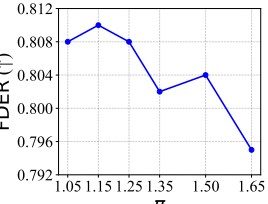

*Figure 6.* Hyperparameter analysis of $K$ and $\pi$ in TIMEGUARD on the PEMS03 dataset under BackTime attack.

sistently mitigates all attacks, improving MAE$_P$ to at least 39.3 (a minimum relative gain of 2.76x) while keeping clean MAE$_C$ within 5% of the undefended model. This indicates strong robustness to recent TSF backdoor attacks. Robustness against recent BadTime attack (Xiang et al., 2025) and per-model results are provided in Appendix G.1.

**Comparison with state-of-the-art defenses.** Table 3 also shows that TIMEGUARD achieves the best overall trade-off among previous training-phase defenses. Compared to the strongest baseline PDB, TIMEGUARD yields a 1.96x relative improvement in MAE$_P$ and a 6.09% relative reduction in MAE$_C$, with average FDER of 0.841 across attacks. Notably, these gains require no additional clean data. Per-model results are provided in Appendix G.1.

**Generalization on different datasets.** As shown in Table 4, TIMEGUARD consistently improves robustness under both Random and BackTime across all three datasets, achieving FDER above 0.65 in all settings. On Weather, TIME-GUARD also slightly improves clean forecasting accuracy over the undefended model (3.02% on average), suggesting that neighborhood-distance-based criteria can act as a regularizer for better generalization. On ETTm1, TIMEGUARD incurs a small drop in clean performance but still delivers strong robustness without initial clean data unlike PDB.

**Generalization across diverse scenarios.** Comprehensive results across model architectures, TSF foundation models, forecasting horizons, poisoning rates, attack patterns, and challenging datasets with nonstationarity, strong distribution shifts, large scale, and count-valued variables are deferred to Appendix G.1. Overall, TIMEGUARD consistently achieves the best defense performance across diverse TSF attack settings and challenging scenarios. Notably, TIME-GUARD remains effective even in the extreme full-channel

*Table 6.* Training time (seconds ↓) of in-training backdoor defenses on the PEMS03 dataset. "No Defense" denotes standard training on the poisoned data without any defense. Best results are in **bold**.

| Model → Defense ↓ | SimpleTM | FEDformer | TimesNet | AVERAGE |
|---|---|---|---|---|
| No Defense | 1621 | 2340 | 2442 | 2134 |
| ABL (Li et al., 2021a) | **740** | **1409** | **2038** | **1395** |
| PDB (Wei et al., 2024) | 2378 | 3441 | 3399 | 3073 |
| ESTI (Yu et al., 2025) | 5563 | 10347 | 11253 | 9054 |
| TIMEGUARD | 2454 | 3411 | 4250 | 3372 |

*Table 7.* Defense performance of TIMEGUARD under BackTime and adaptive attacks on PEMS03 dataset, averaged over three models. Best results under adaptive attack are in **bold**.

| Attack | Defense | $MAE_C$ ↓ | $MAE_P$ ↑ | FDER ↑ |
|---|---|---|---|---|
| BackTime | No Defense | 17.607 | 14.201 | – |
| | **TIMEGUARD** | 18.048 | 39.303 | 0.808 |
| Adaptive | No Defense | 18.791 | 15.343 | – |
| | **TIMEGUARD** | **18.438** | **30.575** | **0.744** |
| | TIMEGUARD w/o NDF | 18.564 | 29.695 | 0.739 |
| | TIMEGUARD w/o DRLS | 20.863 | 19.026 | 0.543 |

poisoning setting, e.g., $\eta_S = 1.0$, achieving FDER of 0.748. Furthermore, TIMEGUARD also transfers to an LLM-based forecaster (Liu et al., 2024c), yielding at least a $5.14\times$ gain in $MAE_P$ with only a $3.8\%$ change in clean $MAE_C$.

### 5.2. Analysis

**Ablation study.** We ablate TIMEGUARD under Random and BackTime to quantify each design component's contribution. As shown in Table 5, removing the channel-wise formulation causes the defense to fail (FDER = 0.493), underscoring the need to match the channel-subset granularity of TSF attacks. NDF and RCF are critical in Stage I for constructing a high-precision reliable pool and preventing early absorption of poisoned samples, as reflected by $MAE_P$. Replacing DRLS with loss-only selection substantially degrades performance, reducing FDER by 28% on average. These results underscore the necessity of distance-aware selection for both clean generalization ($MAE_C$) and robustness ($MAE_P$). Overall, the components contribute synergistically to TIMEGUARD's effectiveness. Additional per-model ablation results are provided in Appendix G.2.

**Influence of $\alpha$ and $\beta$.** Figure 5 shows a clear trade-off between clean performance ($MAE_C$) and robustness ($MAE_P$, FDER) as the pool sizes vary under BackTime. Extremely small or large $\beta$ either admits too few clean samples or incorporates too many poisoned samples, both of which reduce FDER. Similarly, a small $\alpha$ yields an insufficiently reliable initial pool for Stage II, leading to worse $MAE_C$, $MAE_P$, and FDER. Empirically, $\alpha \in [0.15, 0.25]$ and $\beta \in [0.5, 0.7]$ provide the best balance, achieving the highest FDER, exceeding 0.8.

**Influence of $K$ and $\pi$.** With $\alpha$=0.2 and $\beta$=0.5, we study the neighborhood size $K$ and scaling factor $\pi$. As shown in Figure 6, TIMEGUARD is largely insensitive to $K$, with FDER staying in a narrow range (0.805–0.809). In contrast, overly large $\pi$ tends to narrow the candidate set and reduce diversity, increasing the risk of admitting poisoned samples during Stage II. We thus recommend $\pi \leq 1.5$. Full per-model results, hyperparameter sensitivity analyses across different datasets and attacks, and analyses of other hyperparameters are provided in Appendix G.3.

**Efficiency analysis.** With our implementation, memory footprints are the same across methods; we therefore focus

on the wall-clock training time of in-training defenses. As shown in Table 6, TIMEGUARD incurs a $1.58\times$ training-time overhead over vanilla training, mainly due to its multi-stage procedure and neighborhood-distance computations. This overhead is comparable to the strongest baseline, PDB, while providing substantially stronger robustness. In contrast, ESTI incurs a much larger overhead ($4.24\times$ on average) yet remains ineffective against TSF backdoor attacks. Since TIMEGUARD adds no inference-time overhead, it remains practical. Implementation details and large-model running times are provided in Appendix G.4 and G.1.

**Potential adaptive attacks.** We consider a worst-case adaptive scenario in which the attacker extends the state-of-the-art BackTime attack (Lin et al., 2024) by (i) using a well-trained backcaster $b_\phi$ as a regularizer to encourage reverse consistency and (ii) explicitly penalizing high correlation among poisoned samples to evade our neighborhood-based criterion. As shown in Table 7, this adaptive attack attains 18.791 $MAE_C$ and 15.343 $MAE_P$, slightly worse than the original BackTime attack. This is consistent with Theorem 4.1, which suggests that successful TSF backdoor attacks benefit from tight, highly similar clusters of poisoned inputs. Under this adaptive threat, TIMEGUARD remains effective, achieving 18.438 $MAE_C$, 30.575 $MAE_P$, and 0.744 FDER. Ablations further show that neighborhood-based cues remain useful: removing NDF only slightly reduces FDER to 0.739, while removing DRLS causes a larger drop to 0.543. More implementation details and per-model ablation results are provided in Appendix G.5.

## 6. Conclusion

Our paper presents the first systematic study of defenses against TSF backdoor attacks. We first expose key failure modes of existing classification defenses in TSF stemming from data entanglement and task-formulation shift. To address these gaps, we propose TIMEGUARD, a novel backdoor defense for TSF. Specifically, TIMEGUARD performs channel-wise reliable pool training and leverages reverse consistency and temporal pattern concentration in poisoned TSF data to initialize and progressively refine reliable pools. Extensive experiments validate TIMEGUARD's effectiveness and generalization. Overall, our results emphasize the need for more robust and trustworthy forecasting systems. Limitations and future work are discussed in Appendix I.

## Acknowledgment

This research / project is supported by the National Research Foundation, Singapore, and Cyber Security Agency of Singapore under its National Cybersecurity R&D Programme and CyberSG R&D Cyber Research Programme Office. Any opinions, findings and conclusions or recommendations expressed in these materials are those of the author(s) and do not reflect the views of National Research Foundation, Singapore, Cyber Security Agency of Singapore as well as CyberSG R&D Programme Office, Singapore.

## Impact Statement

This work studies backdoor learning in time series forecasting (TSF) and proposes a defense against TSF backdoor attacks. It may improve the reliability of forecasting components in safety- or cost-critical pipelines and support the development of more robust and trustworthy time series machine learning. Potential negative impacts are primarily related to dual use: our analysis and evaluation may help adversaries design more evasive backdoors or adapt poisoning strategies. Accordingly, we report findings under explicit threat models and emphasize that defenses should be complemented by other standard security measures (e.g., data provenance) to provide more comprehensive protection.

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

# A. Related Work

## A.1. Deep Models for Time Series Forecasting

Time series forecasting (TSF) aims to predict future values of one or multiple variables based on their historical observations. With the rapid development of deep learning, a wide variety of TSF DNN architectures have been proposed to model complex temporal dependencies, nonlinear dynamics, and inter-variable dependencies. RNN-based methods (Abbasimehr & Paki, 2022; Hewage et al., 2020; Lin et al., 2023) capture sequential patterns through recursive state transitions, while CNN-based methods (Hewage et al., 2020; Cheng et al., 2025) employ dilated and causal convolutions to efficiently learn long-range temporal features. GNN-based approaches (Yan et al., 2018; Ma et al., 2020) explicitly represent inter-variable correlations by constructing spatio-temporal graphs, enabling information propagation across related variables.

Recently, Transformer-based models (Nie et al., 2023; Zhou et al., 2022; Wu et al., 2023; Chen et al., 2025a) have achieved state-of-the-art TSF performance by leveraging self-attention to jointly capture global temporal dependencies and cross-variable interactions. MLP-based architectures (Han et al., 2024; Wang et al., 2024), built primarily on linear transformations, maintain high computational efficiency while still delivering strong forecasting accuracy. Currently, LLM-based models (Liu et al., 2024b;c) employ pre-trained LMMs as backbones and demonstrate impressive cross-domain generalization and zero-shot forecasting capability. Meanwhile, reinforcement learning has recently become an important paradigm for improving LLM capability and behavior (Fang et al., 2025; Zhang et al., 2025; 2026).

However, the increasing model complexity and data dependency of modern TSF architectures introduce new trustworthiness concerns, including adversarial attacks (Xu et al., 2021; Pialla et al., 2025; Liu et al., 2025b), backdoor attacks (Lin et al., 2024; Kotowski et al., 2025), hallucination (Zou et al., 2025), and watermarking (Soi et al., 2025). In this work, we focus specifically on backdoor defenses for time series forecasting.

## A.2. Backdoor Attacks

Backdoor attacks are typically implemented by injecting a small number of poisoned samples into the training set to implant hidden trigger-target associations (Li et al., 2022a). Once trained on such data, the model behaves normally on clean inputs but exhibits malicious behavior when the trigger appears, for example, classifying triggered samples into an attacker-specified target label. Such attacks have been extensively studies in computer vision (Gu et al., 2019; Gao et al., 2023b; 2024; Zhu et al., 2025; Chen et al., 2026; Li et al., 2026), speech recognition (Zhai et al., 2021; Koffas et al., 2023; Cai et al., 2024), object recognition (Li et al., 2022b; Chan et al., 2022; Luo et al., 2023), and graph learning (Xi et al., 2021), demonstrating that even a tiny poisoning ratio can yield high attack success while maintaining benign performance.

In the time series domain, prior work has primarily examined backdoor attacks on classification tasks (Ding et al., 2022; Jiang et al., 2023; Huang et al., 2025b), where temporal triggers are injected into complete time series to manipulate predictions of physiological or activity signals. However, these studies are restricted to producing categorical output labels for entire time series, rather than finer-grained temporal segments. The first work to target TSF models, BackTime (Lin et al., 2024), embeds stealthy GNN-based trigger patterns with associated predefined target patterns in selected time step on the original training dataset via bi-level optimization. Following this, TBDA (Liu et al., 2026) introduces temporally delayed, variable-specific activations instead of immediate alignment, extending BackTime under a continuity assumption between trigger and target patterns. Meanwhile, BadTime (Xiang et al., 2025) studies long-term TSF and aims to train a backdoored model by using hybrid training strategy to select valuable poisoned samples and a decoupled backdoor objective leveraging distributed lag-aware triggers.

Nevertheless, BadTime assumes a less practical threat model that requires full control over the training pipeline, whereas BackTime assumes only dataset-level control and employs more flexible, sample-dependent triggers. Although distributed lag-aware triggers are expressive, BadTime assumes unrealistic control over all input variables, which are typically distributed across multiple real-world data sources. Therefore, we adopt BackTime as our default threat model and leave a comprehensive evaluation under the BadTime-style threat model for future work.

## A.3. Backdoor Defense

Backdoor defenses aim to mitigate or neutralize backdoor behaviors implanted during training or to detect such behaviors at inference. These methods can be categorized into four stages of the model life cycle (Wu et al., 2025a). *Pre-training-stage defenses* attempt to identify and remove poisoned samples before model training by analyzing training samples statistics

or feature distributions to detect anomalous samples (Tran et al., 2018; Chen et al., 2018; Mo et al., 2024; Le et al., 2025; Hou et al., 2025). *In-training-stage defenses* aim to train clean models on poisoned datasets without backdoor injection, typically by reducing the influence of potentially poisoned samples through carefully designed training procedures (Li et al., 2021a; Tang et al., 2023; Gao et al., 2023a; Wei et al., 2024; Yu et al., 2025; Qiao et al., 2026). *Post-training-stage defenses* repair compromised models through structural modification or fine-tuning-based approaches (Liu et al., 2018; Wang et al., 2019; Li et al., 2021b; Dunnett et al., 2025; Xu et al., 2024; Chen et al., 2025b), Finally, *inference-time defenses* detect the presence of triggers at test time by measuring prediction consistency or entropy under different input perturbations or one input with multiple model variances (Liu et al., 2023a; Gao et al., 2019; Guo et al., 2023; Hou et al., 2024; Yi et al., 2025).

Although these defenses have demonstrated effectiveness in classification and vision domains (Wu et al., 2025b), their applicability to the time series domain remains largely underexplored. In time series classification, one representative effort is E2ABL (Jiang et al., 2024), which extends ABL (Li et al., 2021a) and evaluates existing backdoor defenses on time series classification datasets. However, this work primarily focuses on empirical evaluation rather than proposing defenses that explicitly account for the temporal structure of time series inputs. Likewise, backdoor defenses for TSF remain largely unexplored. One notable concurrent effort in TSF backdoor defense is the competition associated with the "Assurance for Space Domain AI Applications" program, which aims to detect and reconstruct static trigger patterns in backdoored TSF models (Kotowski et al., 2025; Wang et al., 2026). However, this setting assumes access to clean models with the same architecture as the poisoned models, as well as clean datasets, and is limited to a subset of model architectures within the space operations domain. In this work, we conduct the first systematic study of representative backdoor defenses across the TSF model life cycle, spanning multiple domains and model architectures. We further introduce TIMEGUARD, an in-training-stage defense specifically designed for TSF backdoors.

## B. Further Analysis of Existing Backdoor Defenses

Beyond the two fundamental issues in current backdoor defense settings for time series forecasting (TSF), namely data entanglement and task-formulation shift as discussed in Section 1, we further provide a detailed analysis of TSF-specific challenges that hinder the direct adaptation of existing defenses. We summarize these challenges in Section B.1. We then provide the rationale for selecting representative baselines in Section B.2 and clarify the practical attribute aspects of each defense in the TSF backdoor setting in Section B.3.

### B.1. TSF-Specific Challenges for Backdoor Defense

Similar to backdoor attacks (Lin et al., 2024), defending TSF models against backdoor attacks presents several unique challenges compared to traditional backdoor defense in classification and generative models (Wu et al., 2025a; Li et al., 2025; Lin et al., 2025). These difficulties largely come from the intrinsic properties of forecasting. **(i)** The target outputs in TSF lie in a *continuous space*, making it infeasible for label-based defenses (Chen et al., 2018; Wang et al., 2019; Chou et al., 2020; Shen et al., 2025) that rely on either identifying poisoned classes or reconstructing potential triggers for each label in the output space. **(ii)** Samples in TSF exhibit strong *temporal dependencies*, where a single injected trigger or target pattern can propagate across overlapping input-output windows, contaminating subsequent forecasting steps and making it difficult to distinguish between clean and poisoned samples. **(iii)** Time series data are often *uninterpretable to human*; detecting abnormal fluctuations or poison patterns typically requires domain expertise (e.g., finance or healthcare), making manual inspection unreliable and the construction of a trusted clean dataset prohibitively expensive. **(iv)** TSF models are typically deployed in *continuous real-time settings*, where forecasts are generated sequentially and updated as new data arrive. Defense methods therefore must operate efficiently, limiting the practicality of inference-time detection methods that often require multiple forward passes (Gao et al., 2019; Liu et al., 2023a; Hou et al., 2024).

Beyond these factors, the representational characteristics of TSF models also introduce further challenges for defense adaptation. **(i)** The *heterogeneous representations* produced by different deep TSF models (Kim et al., 2025) significantly hinder defense generalization. For instance, some models explicitly decompose time series into separate trend and seasonal components (Wu et al., 2023), while others rely on frequency-based transformations (Zhou et al., 2022) or channel-independent that processes each variable independently (Nie et al., 2023). As a result, their hidden representation spaces vary substantially across architectures, underscoring the need for model-agnostic (architecture-agnostic) defense design. **(ii)** Unlike classification or word embedding models (Radford et al., 2021), whose latent representations often align with semantically discrete concepts (e.g., object categories or word meanings), the semantics of hidden representations in TSF remain largely *underexplored*, further making representation-based defenses (Tran et al., 2018; Mo et al., 2024) unreliable under different DNNs.

*Table 8.* Key attributes of defense methods against TSF backdoor attacks.

| Method | Defense Stage | No Additional Clean Data Required | No Internal Features Access Required | No Additional Inference Overhead | Time-Aware Design |
|---|---|---|---|---|---|
| Spectral (Tran et al., 2018) | Pre-training | ✓ | ✗ | ✓ | ✗ |
| TED (Mo et al., 2024) | Pre-training | ✗ | ✗ | ✓ | ✗ |
| TED++ (Le et al., 2025) | Pre-training | ✗ | ✗ | ✓ | ✗ |
| Fine-tuning (Gu et al., 2019) | Post-training | ✗ | ✓ | ✓ | ✗ |
| Fine-pruning (Liu et al., 2018) | Post-training | ✗ | ✗ | ✓ | ✗ |
| NAD (Li et al., 2021b) | Post-training | ✗ | ✗ | ✓ | ✗ |
| IMS (Dunnett et al., 2025) | Post-training | ✗ | ✗ | ✓ | ✗ |
| ABL (Li et al., 2021a) | In-training | ✓ | ✓ | ✓ | ✗ |
| PDB (Wei et al., 2024) | In-training | ✗ | ✓ | ✗ | ✗ |
| ESTI (Yu et al., 2025) | In-training | ✗ | ✓ | ✓ | ✗ |
| STRIP (Gao et al., 2019) | Inference | ✗ | ✓ | ✗ | ✗ |
| TeCo (Liu et al., 2023a) | Inference | ✓ | ✓ | ✗ | ✗ |
| IBD-PSC (Hou et al., 2024) | Inference | ✗ | ✗ | ✗ | ✗ |
| **TimeGuard** | In-training | ✓ | ✓ | ✓ | ✓ |

### B.2. Rationale for Selecting Representative Defenses

We evaluate **13** representative defenses spanning four stages of the model life cycle, following the taxonomy of Back-doorBench (Wu et al., 2025a). Specifically, our selection is guided by two criteria. **(i) Representativeness**: we include both classic methods (e.g., Spectral (Tran et al., 2018), Fine-pruning (Liu et al., 2018), ABL (Li et al., 2021a)) and recent advanced approaches (e.g., PDB (Wei et al., 2024), TED++ (Le et al., 2025), and ESTI (Yu et al., 2025)) that have shown effectiveness in classification or vision domains. **(ii) Adaptation Feasibility**: the method must be practical to adapt to TSF.

Accordingly, we exclude algorithms that depend on discrete output spaces, such as those requiring enumeration of all target labels to detect poisoned samples (Chen et al., 2018; Shen et al., 2025), or access to poisoned labels (Shen et al., 2025), as well as methods relying on self- or semi-supervised learning frameworks (Huang et al., 2022; Gao et al., 2023a), which remain architecture-dependent and are not yet applicable to diverse TSF models (Zhang et al., 2024a; Cho & Lee, 2025).

### B.3. Key Practical Attributes for TSF Defenses

To systematically compare these defenses, we examine four key attributes relevant to forecasting: **(i) No Additional Clean Data Required**: whether the method avoids dependence on a clean split, addressing the challenge of constructing trusted datasets for time series; **(ii) No Internal Feature Access Required**: whether the defense operates without access to intermediate activations or feature representations, reflecting model-agnostic applicability; **(iii) No Additional Inference Overhead**: whether the defense incurs extra computational cost during inference, which is critical for real-time forecasting deployments; and **(iv) Time-Aware Design**: whether the defense explicitly incorporates time-series characteristics. As summarized in Table 8, substantial differences emerge across defenses at different stages. Post-training-stage methods typically rely on additional clean data, while pre-training-stage defenses often require access to internal representations. Inference-time defenses, on the other hand, introduce notable inference overhead, limiting their deployment efficiency. Importantly, *none of the existing defenses* explicitly accounts for temporal dynamics, as they were all originally designed for static classification tasks. Motivated by these observations, we evaluate these defenses in the TSF setting in Section 3 and introduce TimeGuard as a in-training time-aware backdoor defense in Section 4. We leave the development of efficient inference-time backdoor defenses for future work.

## C. Theoretical Analysis of TSF Backdoor Success

In this section, we provide a bound showing that successful and stealthy TSF backdoor attacks tend to induce highly similar (and thus highly correlated) poisoned input windows, motivating the design of TimeGuard. For readability, we focus on a single channel so that each history window is a vector $\mathbf{x}_{t,h} \in \mathbb{R}^{L_{\text{in}}}$ and each future window is $\mathbf{x}_{t,f} \in \mathbb{R}^{L_{\text{out}}}$. We denote a triggered test input by $\mathbf{x} := \mathbf{x}_{t,h}$, background inputs by $\mathbf{x}_i := \mathbf{x}_{i,h}$ with outputs $\mathbf{y}_i := \mathbf{x}_{i,f}$, and poisoned inputs by $\mathbf{x}'_j$.

**Setup.** Following (Xian et al., 2023; Guo et al., 2022), we approximate a TSF predictor in a kernel regression regime (Jacot

et al., 2018). Assume all windows are instance-normalized during preprocessing. Let $K(\mathbf{u}, \mathbf{v})$ be an RBF kernel

$$K(\mathbf{u}, \mathbf{v}) \;=\; \exp\big(-\gamma \|\mathbf{u} - \mathbf{v}\|_2^2\big),$$

with bandwidth $\gamma > 0$. The training set consists of $N_\mathrm{p}$ poisoned samples $\mathcal{D}_\mathrm{p} = \{(\mathbf{x}'_j, \mathbf{y}'_j)\}_{j=1}^{N_\mathrm{P}}$ and $N_\mathrm{bg}$ background samples $\mathcal{D}_\mathrm{bg} = \{(\mathbf{x}_i, \mathbf{y}_i)\}_{i=1}^{N_\mathrm{bg}}$ (e.g., containing clean and affected samples).

**Attack mechanism and target mapping.** In the threat model (Lin et al., 2024), the attacker inserts a trigger into the history window and enforces a patterned target in the future window. At the dataset level (multivariate notation), for an injection time $t$ and attacked channel subset $\mathcal{S}$:

$$\mathbf{X}[t - L_\mathrm{tgr} : t, \mathcal{S}] \leftarrow \mathbf{G}_t, \qquad \mathbf{X}[t : t + L_\mathrm{ptn}, \mathcal{S}] \leftarrow \mathbf{X}[t - L_\mathrm{tgr} - 1, \mathcal{S}] \oplus \mathbf{P},$$

where $\mathbf{G}_t$ is a trigger pattern at timestep $t$ and $\mathbf{P}$ is a fixed attack pattern template and $\oplus$ denotes element-wise addition (with broadcasting along time when needed). This produces sample-dependent target patterns because the baseline term $\mathbf{X}[t - L_\mathrm{tgr} - 1, \mathcal{S}]$ varies across samples. We abstract this behavior via a deterministic mapping $T(\cdot)$ at the window level.

**Definition C.1** (Backdoor Target Mapping). Let $\mathbf{p}$ denote the attack pattern template (aligned to the selected channel within $\mathbf{P}$), and let $b(\mathbf{x})$ extract a baseline value from an input window (e.g., the value immediately preceding the trigger, broadcast to match the target horizon). Define

$$T(\mathbf{x}) \;:=\; b(\mathbf{x}) \oplus \mathbf{p}.$$

For a poisoned input $\mathbf{x}'_j$, its poisoned label is $\mathbf{y}'_j = T(\mathbf{x}'_j)$. For a triggered test window $\mathbf{x}$, the attacker aims for $\hat{\mathbf{y}}(\mathbf{x}) \approx T(\mathbf{x})$.

**Key quantities.** For a triggered test input $\mathbf{x}$, define the maximum similarity to background inputs:

$$\varepsilon \;:=\; \max_{(\mathbf{x}_i, \mathbf{y}_i) \in \mathcal{D}_\mathrm{bg}} K(\mathbf{x}, \mathbf{x}_i),$$

and define the poison dispersion around $\mathbf{x}$:

$$\sigma_p^2(\mathbf{x}) \;:=\; \frac{1}{N_\mathrm{P}} \sum_{j=1}^{N_\mathrm{P}} \|\mathbf{x} - \mathbf{x}'_j\|_2^2, \qquad \sigma_p(\mathbf{x}) := \sqrt{\sigma_p^2(\mathbf{x})}.$$

Intuitively, $\varepsilon$ measures how strongly background samples can influence prediction at $\mathbf{x}$, while $\sigma_p(\mathbf{x})$ measures how tightly poisoned inputs concentrate around $\mathbf{x}$.

**Theorem C.2** (TSF Backdoor Success Bound). *Let $\hat{\mathbf{y}}(\cdot)$ be the Nadaraya–Watson kernel regressor trained on $\mathcal{D}_\mathrm{bg} \cup \mathcal{D}_\mathrm{p}$:*

$$\hat{\mathbf{y}}(\cdot) = \frac{\sum_{i=1}^{N_\mathrm{bg}} K(\cdot, \mathbf{x}_i)\, \mathbf{y}_i \;+\; \sum_{j=1}^{N_\mathrm{P}} K(\cdot, \mathbf{x}'_j)\, T(\mathbf{x}'_j)}{\sum_{i=1}^{N_\mathrm{bg}} K(\cdot, \mathbf{x}_i) \;+\; \sum_{j=1}^{N_\mathrm{P}} K(\cdot, \mathbf{x}'_j)}.$$

*Assume:*

1. ***Bounded background deviation.*** *For the triggered test window $\mathbf{x}$, $\|\mathbf{y}_i - T(\mathbf{x})\|_2 \leq M$ for all $(\mathbf{x}_i, \mathbf{y}_i) \in \mathcal{D}_\mathrm{bg}$.*

2. ***Local Lipschitzness of*** $T$. *There exists $L_T > 0$ such that*

$$\|T(\mathbf{u}) - T(\mathbf{v})\|_2 \leq L_T \|\mathbf{u} - \mathbf{v}\|_2 \quad \textit{for all } \mathbf{u}, \mathbf{v} \textit{ in a neighborhood of } \{\mathbf{x}\} \cup \{\mathbf{x}'_j\}_{j=1}^{N_\mathrm{P}}.$$

*Then for the triggered test window $\mathbf{x}$,*

$$\big\|\hat{\mathbf{y}}(\mathbf{x}) - T(\mathbf{x})\big\|_2 \;\leq\; \underbrace{\frac{N_\mathrm{bg}\, M\, \varepsilon}{N_\mathrm{p}\, \exp\big(-\gamma\, \sigma_p^2(\mathbf{x})\big)}}_{\textit{(I) background influence}} \;+\; \underbrace{L_T\, \sigma_p(\mathbf{x})}_{\textit{(II) target mismatch}}.$$

*Proof.* Let

$$W(\mathbf{x}) := \sum_{i=1}^{N_{\mathrm{bg}}} K(\mathbf{x}, \mathbf{x}_i) + \sum_{j=1}^{N_{\mathrm{p}}} K(\mathbf{x}, \mathbf{x}'_j), \qquad W_p(\mathbf{x}) := \sum_{j=1}^{N_{\mathrm{p}}} K(\mathbf{x}, \mathbf{x}'_j).$$

Subtract $T(\mathbf{x})$ from $\hat{\mathbf{y}}(\mathbf{x})$ and regroup:

$$\hat{\mathbf{y}}(\mathbf{x}) - T(\mathbf{x}) = \frac{\sum_{i=1}^{N_{\mathrm{bg}}} K(\mathbf{x}, \mathbf{x}_i)\big(\mathbf{y}_i - T(\mathbf{x})\big) + \sum_{j=1}^{N_{\mathrm{p}}} K(\mathbf{x}, \mathbf{x}'_j)\big(T(\mathbf{x}'_j) - T(\mathbf{x})\big)}{W(\mathbf{x})}.$$

Taking norms and applying triangle inequality yields two terms.

**(I) Background influence.** Using $\|\mathbf{y}_i - T(\mathbf{x})\|_2 \le M$ and $K(\mathbf{x}, \mathbf{x}_i) \le \varepsilon$,

$$\Big\| \sum_{i=1}^{N_{\mathrm{bg}}} K(\mathbf{x}, \mathbf{x}_i)\big(\mathbf{y}_i - T(\mathbf{x})\big) \Big\|_2 \le \sum_{i=1}^{N_{\mathrm{bg}}} K(\mathbf{x}, \mathbf{x}_i)\, \|\mathbf{y}_i - T(\mathbf{x})\|_2 \le N_{\mathrm{bg}}\, M\, \varepsilon.$$

Moreover, $W(\mathbf{x}) \ge W_p(\mathbf{x})$, hence the term is upper bounded by $\dfrac{N_{\mathrm{bg}}\, M\, \varepsilon}{W_p(\mathbf{x})}$. To lower bound $W_p(\mathbf{x})$, let $\delta_j := \|\mathbf{x} - \mathbf{x}'_j\|_2^2$ so $K(\mathbf{x}, \mathbf{x}'_j) = \exp(-\gamma \delta_j)$. By Jensen's inequality (since $z \mapsto e^{-\gamma z}$ is convex),

$$\frac{1}{N_{\mathrm{p}}} \sum_{j=1}^{N_{\mathrm{p}}} \exp(-\gamma \delta_j) \ge \exp\Big( -\gamma \cdot \frac{1}{N_{\mathrm{p}}} \sum_{j=1}^{N_{\mathrm{p}}} \delta_j \Big) = \exp\big( -\gamma \sigma_p^2(\mathbf{x}) \big).$$

Multiplying by $N_{\mathrm{p}}$ gives

$$W_p(\mathbf{x}) = \sum_{j=1}^{N_{\mathrm{p}}} \exp(-\gamma \delta_j) \ge N_{\mathrm{p}}\, \exp\big( -\gamma \sigma_p^2(\mathbf{x}) \big),$$

which proves term (I).

**(II) Target mismatch.** By Lipschitzness of $T$,

$$\|T(\mathbf{x}'_j) - T(\mathbf{x})\|_2 \le L_T \|\mathbf{x}'_j - \mathbf{x}\|_2.$$

Thus,

$$\Big\| \sum_{j=1}^{N_{\mathrm{p}}} K(\mathbf{x}, \mathbf{x}'_j)\big(T(\mathbf{x}'_j) - T(\mathbf{x})\big) \Big\|_2 \le L_T \sum_{j=1}^{N_{\mathrm{p}}} K(\mathbf{x}, \mathbf{x}'_j)\, \|\mathbf{x}'_j - \mathbf{x}\|_2.$$

Let $w_j := K(\mathbf{x}, \mathbf{x}'_j)/W_p(\mathbf{x})$ so that $w_j \ge 0$ and $\sum_j w_j = 1$. Then

$$\frac{\sum_{j=1}^{N_{\mathrm{p}}} K(\mathbf{x}, \mathbf{x}'_j)\, \|\mathbf{x}'_j - \mathbf{x}\|_2}{W_p(\mathbf{x})} = \sum_{j=1}^{N_{\mathrm{p}}} w_j\, \|\mathbf{x}'_j - \mathbf{x}\|_2 \le \sqrt{\sum_{j=1}^{N_{\mathrm{p}}} w_j\, \|\mathbf{x}'_j - \mathbf{x}\|_2^2} = \sqrt{\sum_{j=1}^{N_{\mathrm{p}}} w_j\, \delta_j},$$

where the inequality is Cauchy–Schwarz. Now note that $\sum_j w_j \delta_j$ is the expectation of $\delta$ under the Gibbs weights $w_j \propto e^{-\gamma \delta_j}$. This expectation is non-increasing in $\gamma$ and equals the uniform mean at $\gamma = 0$; therefore for $\gamma > 0$,

$$\sum_{j=1}^{N_{\mathrm{p}}} w_j\, \delta_j \; \le \; \frac{1}{N_{\mathrm{p}}} \sum_{j=1}^{N_{\mathrm{p}}} \delta_j \; = \; \sigma_p^2(\mathbf{x}).$$

Hence $\sum_j w_j \|\mathbf{x}'_j - \mathbf{x}\|_2 \le \sigma_p(\mathbf{x})$, proving term (II). Combining (I) and (II) completes the proof. $\qquad \square$

*Remark* C.3 (Connection to Correlation-based Neighborhood Distance in Section 4). For two z-normalized vectors $\mathbf{u}, \mathbf{v} \in \mathbb{R}^{L_{\mathrm{in}}}$, the squared Euclidean distance satisfies

$$\|\mathbf{u} - \mathbf{v}\|_2^2 \; = \; 2\, L_{\mathrm{in}} \big( 1 - \rho(\mathbf{u}, \mathbf{v}) \big),$$

where $\rho(\mathbf{u}, \mathbf{v})$ is the Pearson correlation. Defining $d_{\mathrm{nb}}(\mathbf{u}, \mathbf{v}) := 1 - \rho(\mathbf{u}, \mathbf{v})$ (or a weighted variant $d_{\mathrm{nb}}(\mathbf{u}, \mathbf{v}) := 1 - \rho_w(\mathbf{u}, \mathbf{v})$), we obtain $\|\mathbf{u} - \mathbf{v}\|_2^2 \propto d_{\mathrm{nb}}(\mathbf{u}, \mathbf{v})$ for normalized windows (up to a constant factor depending on $L_{\mathrm{in}}$), consistent with Berthold & Höppner (2016).

Therefore, $\sigma_p^2(\mathbf{x}) = \frac{1}{N_{\mathrm{p}}} \sum_{j=1}^{N_{\mathrm{p}}} \|\mathbf{x} - \mathbf{x}_j'\|_2^2$ is proportional (up to constants) to $\frac{1}{N_{\mathrm{p}}} \sum_{j=1}^{N_{\mathrm{p}}} d_{\mathrm{nb}}(\mathbf{x}, \mathbf{x}_j')$. Hence, requiring small $\sigma_p^2(\mathbf{x})$ corresponds to poisoned inputs forming a tight cluster under our neighborhood distance with high temporal correlation.

Moreover, TSF backdoor attacks commonly rely on a shared attack-pattern template $\mathbf{p}$, which makes poisoned input–output windows redundant and highly similar. Motivated by prior observations that time steps near the prediction boundary between input and output windows exert stronger influence on prediction manipulation (Lin et al., 2024; Xiang et al., 2025), we adopt a Gaussian-weighted Pearson correlation when computing $d_{\mathrm{nb}}(\cdot, \cdot)$, supporting our neighborhood diversity filtering in Section 4.2 and Section 4.3.

# D. Method Details

## D.1. Training Algorithm Outline

The pseudocode of the our proposed method TIMEGUARD is listed as in Algorithm 1.

## D.2. Comparison with Distance-based Backdoor Defenses

At a high level, TIMEGUARD may appear related to prior distance-based backdoor defenses. However, most existing distance-based defenses typically operate in learned representation spaces and typically rely on a separability assumption between poisoned and clean samples (Chen et al., 2018; Tran et al., 2018; Hayase et al., 2021; Huang et al., 2025a). Such assumptions can be brittle even in standard vision settings, where representation-based filtering may break down under more challenging scenarios (e.g., source-specific or dynamic triggers) (Mo et al., 2024). The mismatch is further exacerbated in TSF: (i) forecasting is a regression task without discrete target classes for within-class clustering, (ii) TSF backdoors are often channel-subset, so the overall sample representation could remain close to clean, and (iii) heterogeneous internal representations across forecasting architectures make it difficult to apply a unified representation-space criterion. Consequently, clean/poison separation in learned activations is not a reliable primitive for TSF.

In contrast, TIMEGUARD uses distance in a fundamentally different way. Rather than measuring learned representations, we compute data-space neighborhood distances between instance-normalized channel-wise windows (equivalently, correlation-based distances) and use them to measure local temporal similarity concentration with theoretical support. The key signal is not global separability, but an abnormal neighborhood dispersion pattern induced by trigger and target patterns reuse: poisoned windows tend to exhibit unusually small distances to their nearest neighbors along the attacked channels, even when they remain mixed with clean windows overall. This distance cue is then fused with TSF-specific directional evidence (reverse consistency loss) to progressively construct a reliable pool during training, without requiring access to intermediate activations or assuming feature-space clustering structure.

# E. Evaluation Metrics

For training-phase defenses, we use two typical metrics: clean forecasting error ($\mathbf{MAE_C}$), attack forecasting error ($\mathbf{MAE_P}$). $\mathrm{MAE_C}$ measures the Mean Absolute Error (MAE) between model's output and ground-truth future values on clean inputs, reflecting natural forecasting ability. $\mathrm{MAE_P}$ measures the MAE between model's output and the target pattern when the input contain triggers, reflecting resistance against backdoor manipulation. A desirable defense should achieve a low $\mathrm{MAE_C}$ while having a high $\mathrm{MAE_P}$ following prior backdoor settings (Gao et al., 2023a; Yu et al., 2025).

Taking both $\mathrm{MAE_C}$ and $\mathrm{MAE_P}$ into account, we further propose a new metric, **Forecasting Defense Effectiveness Rating (FDER)**, adapted from the Defense Effective Rate (DER) originally proposed for classification models (Zhu et al., 2023). Unlike DER, which relies on accuracy-based metrics, FDER employs relative error-based measures more suitable for forecasting:

$$\mathrm{FDER} = \frac{\max(0, \rho_{\mathrm{MAE_P}}) - \max(0, \rho_{\mathrm{MAE_C}}) + 1}{2} \in [0, 1],$$

---

**Algorithm 1** Pseudocode for TIMEGUARD

---

**Input:** training set $\mathcal{D}$ from poisoned series $\mathbf{X} \in \mathbb{R}^{T \times C}$; forecaster $f_\theta$; backcaster epochs $T_b$; Stage I epochs $T_1$; Stage II epochs $T_2$; init ratio $\alpha$; max ratio $\beta$; neighbors $K$; candidate scaling factor $\pi$.

**Output:** defended forecaster $f_\theta$.

**# Stage I: Time-aware Reliable Pool Initialization** (Section 4.2)

Initialize backcaster $b_\phi$ with the same architecture as $f_\theta$.

**for** $e = 1$ **to** $T_b$ **do**
    **for all** $(\mathbf{X}_{t,h}, \mathbf{X}_{t,f}) \in \mathcal{D}$ **do**
        $\phi \leftarrow \phi - \nabla_\phi \, \ell\big(b_\phi(\text{Flip}(\mathbf{X}_{t,f})), \, \text{Flip}(\mathbf{X}_{t,h})\big)$
    **end for**
**end for**

**for** $c = 1$ **to** $C$ **do**
    # RCF: Reverse-Consistency Filtering
    Compute $\mathcal{D}_{\text{RCF}}^{(c)}$ using $\Gamma_{\text{RCF}}$ as the $\alpha$-quantile of reverse-consistency losses (Eq. 5).
    # NDF: Neighborhood Diversity Filtering
    Compute neighborhood distances $S^{(c)}(\cdot)$ with $\mathcal{D}^{(c)}$ as neighbors (Eq. 8).
    Select $\mathcal{D}_{\text{NDF}}^{(c)}$ using $\Gamma_{\text{NDF}}$ as the $(1 - \alpha)$-quantile (Eq. 9).
    $\mathcal{D}_{\text{rel}}^{(c)} \leftarrow \mathcal{D}_{\text{RCF}}^{(c)} \cap \mathcal{D}_{\text{NDF}}^{(c)}; \quad \mathcal{D}_{\text{unrel}}^{(c)} \leftarrow \mathcal{D}^{(c)} \setminus \mathcal{D}_{\text{rel}}^{(c)}$
**end for**

Update mask $m_{t,c} \leftarrow \mathbb{1}\Big[ (\mathbf{x}_{t,h}^{(c)}, \mathbf{x}_{t,f}^{(c)}) \in \mathcal{D}_{\text{rel}}^{(c)} \Big]$ for all $(t, c)$.

**for** $e = 1$ **to** $T_1$ **do**
    **for all** $(\mathbf{X}_{t,h}, \mathbf{X}_{t,f}) \in \mathcal{D}$ **do**
        $\theta \leftarrow \theta - \nabla_\theta \, \mathcal{L}_{\text{def}}(\theta; m)$ (Eq. 3).
    **end for**
**end for**

**# Stage II: Distance-Regularized Loss Selection** (Section 4.3)

**for** $e = 1$ **to** $T_2$ **do**
    $\gamma \leftarrow \alpha + \frac{\beta - \alpha}{T_2 - 1}(e - 1)$ {Current target clean ratio (treat 0/0 as 0).}
    **for** $c = 1$ **to** $C$ **do**
        # DRLS: Distance-Regularized Loss Selection
        Compute $S^{(c)}(\cdot)$ with $\mathcal{D}_{\text{unrel}}^{(c)}$ as neighbors (Eq. 8).
        Select candidate set $\mathcal{D}_{\text{NDF}}^{\text{cand}(c)}$ using $\Gamma_{\text{NDF}}$ as the $(1 - \pi\gamma)$-quantile (Eq. 9). {top $100\pi\gamma\%$ of $\mathcal{D}^{(c)}$}
        Update $\mathcal{D}_{\text{rel}}^{(c)}$ using $\Gamma_{\text{DRLS}}$ as the $(1/\pi)$-quantile of losses over $\mathcal{D}_{\text{DRLS}}^{\text{cand}(c)}$ (Eq. 10). {equivalent of $100\gamma\%$ of $\mathcal{D}^{(c)}$}
        $\mathcal{D}_{\text{unrel}}^{(c)} \leftarrow \mathcal{D}^{(c)} \setminus \mathcal{D}_{\text{rel}}^{(c)}$
    **end for**
    Update mask $m_{t,c} \leftarrow \mathbb{1}\Big[ (\mathbf{x}_{t,h}^{(c)}, \mathbf{x}_{t,f}^{(c)}) \in \mathcal{D}_{\text{rel}}^{(c)} \Big]$ for all $(t, c)$.
    **for all** $(\mathbf{X}_{t,h}, \mathbf{X}_{t,f}) \in \mathcal{D}$ **do**
        $\theta \leftarrow \theta - \nabla_\theta \, \mathcal{L}_{\text{def}}(\theta; m)$ (Eq. 3).
    **end for**
**end for**

---

where the relative clean gain ($\rho_{\mathrm{MAE_C}}$) and relative attack gain ($\rho_{\mathrm{MAE_P}}$) are defined as:

$$\rho_{\mathrm{MAE_C}} = 1 - \frac{\mathrm{MAE_C^{und}}}{\mathrm{MAE_C}}, \qquad \rho_{\mathrm{MAE_P}} = 1 - \frac{\mathrm{MAE_P^{und}}}{\mathrm{MAE_P}}.$$

Here $\rho_{\mathrm{MAE_C}}$ quantifies the relative increase in clean forecasting error (performance overhead), while $\rho_{\mathrm{MAE_P}}$ quantifies the relative increase in attack forecasting error (robustness gain) after defense. $\mathrm{MAE_C^{und}}$ and $\mathrm{MAE_P^{und}}$ denote the clean and attack forecasting errors of undefended model. A higher FDER value indicates stronger defense effectiveness with smaller degradation of clean forecasting performance.

For inference-time defenses, which aim to identify triggered input samples during prediction, following Liu et al. (2023a), we adopt two evaluation metrics: (i) the **AUROC**, which measures the trade-off between true and false detection rates, and (ii) **F1 score**, which measures the harmonic mean of precision and recall, reflecting the overall detection performance. Higher AUROC and F1 scores indicate stronger detection capability and more reliable inference-time defense performance.

In our setting, benign means good forecasting on clean inputs (low $\mathrm{MAE_C}$); malicious success means that triggered inputs are steered toward the attacker's target (low $\mathrm{MAE_P}$); and simply wrong means the model performs poorly in general, which is also reflected by high error on clean inputs. We also do not assume that poisoned TSF samples must always have globally distinct trajectories from benign ones, since both the trigger and target patterns are attacker-defined. When these patterns mimic common clean motifs, poisoned and clean samples can indeed become ambiguous. Therefore, defense success should not be judged by trajectory separability, but by whether a method preserves benign forecasting utility while disrupting malicious target alignment.

# F. Experimental Protocol

## F.1. Environments

All experiments are implemented in PyTorch 2.1.0+cu118 and run on a Linux 22.04.5 LTS server equipped with $4\times$ NVIDIA RTX A6000 Ada GPUs.

## F.2. Dataset Description

*Table 9.* Dataset statistics.

| Dataset | # Timestamps | # Variables (channels) |
|---------|--------------|------------------------|
| PEMS03 | 26208 | 358 |
| Weather | 52696 | 21 |
| ETTm1 | 69680 | 7 |

We primarily evaluate TIMEGUARD on three real-world multivariate forecasting benchmarks spanning traffic, meteorology, and energy systems: PEMS03 (Song et al., 2020), Weather (Wu et al., 2021), and ETTm1 (Zhou et al., 2022). Table 9 summarizes their basic statistics; we briefly describe each dataset below.

- **PEMS03.** A traffic forecasting dataset built from Caltrans' Performance Measurement System (PeMS) loop-detector data. We use 5-minute aggregated measurements from 358 sensors (Sep-Nov 2018). PeMS provides standard traffic signals such as flow, speed, and occupancy.

- **Weather.** Hourly weather-station observations from NOAA NCEI Local Climatological Data, covering nearly 1,600 U.S. locations from 2010–2013. We forecast wet-bulb temperature using accompanying meteorological variables.

- **ETTm1.** A 15-minute-resolution subset of the Electricity Transformer Temperature (ETT) collection, containing 7 channels (oil temperature as the target and 6 load-related variables) over roughly two years.

We use the preprocessed versions of all datasets provided by TSLib[1], consistent with the data pipeline used in BackTime[2].

---

[1] https://github.com/thuml/Time-Series-Library
[2] https://github.com/xiaolin-cs/BackTime

### F.3. Forecasting Models

To evaluate whether TIMEGUARD and other defenses are model-agnostic, we primarily apply them to three representative forecasting backbones under backdoor attacks:

- **FEDformer** (Zhou et al., 2022). A Transformer-based forecaster that combines seasonal–trend decomposition with frequency-domain modeling (e.g., Fourier bases) to capture global patterns efficiently.[3]

- **TimesNet** (Wu et al., 2023). A period-aware architecture that maps 1D sequences into structured 2D representations and applies an inception-style block to model temporal variations across discovered periods.[4]

- **SimpleTM** (Chen et al., 2025a). A lightweight multivariate forecasting baseline that tokenizes each channel via a stationary wavelet transform and models cross-channel dependencies with a simple interaction module.[5]

For each backbone, we use the authors' official implementation and follow the default training configuration as closely as possible. When the released code provides multiple recommended settings (e.g., varying by dataset or prediction horizon), we adopt the most commonly used configuration. All exact hyperparameters for each model are provided in our code release.

### F.4. Attack Methods

To assess how well each defense generalizes across different TSF backdoor strategies, we evaluate robustness under the following attacks:

- **BackTime** (Lin et al., 2024). A state-of-the-art TSF backdoor attack that selects vulnerable timestamps and synthesizes *sample-dependent* triggers via a GNN-based generator, leveraging inter-variable correlations. We follow BackTime and constrain the trigger perturbation by a budget $\Delta_{\text{tgr}}$.

- **Random**. A simple BadNets-inspired (Gu et al., 2019) baseline that injects a *fixed* random trigger shared across all poisoned timestamps. We sample the trigger from $\mathcal{U}[-\Delta_{\text{tgr}}, \Delta_{\text{tgr}}]$.

- **FreqBack-TSF**. An adaptation of FreqBack (Huang et al., 2025b) to forecasting that utilizes a *learned universal* trigger guided by frequency-domain analysis. Concretely, we replace BackTime's sample-dependent GNN trigger generator with a single trainable trigger tensor and optimize it using FreqBack's frequency-guided objective (frequency and regularization terms), together with the standard target-pattern construction loss. We estimate the frequency heatmap of the trigger position for each selected poisoned channel. Since the original paper does not specify the perturbation-norm weighting, we set $\lambda=1$ and keep all other hyperparameters consistent with the official implementation.

In addition to the above three attacks, we report results for the **Manhattan** baseline from BackTime (Lin et al., 2024), which uses triggers that mimic common temporal patterns. Specifically, Manhattan retrieves segments closest to the target pattern under the Manhattan (L1) distance and uses the preceding window as the trigger. Unless otherwise specified, we follow the default BackTime setting with window lengths $L_{\text{in}}=L_{\text{out}}=12$, temporal injection rate $\eta_{\text{T}}=0.03$, and spatial injection rate $\eta_{\text{S}}=0.3$. We use the **cone-shaped attack pattern** by default following BackTime; details of the attack patterns are provided in Section F.5.

### F.5. Attack Patterns

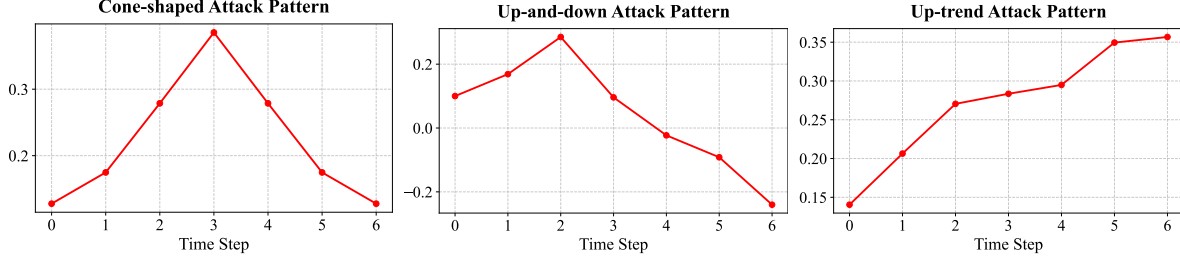

*Figure 7.* Attack pattern shapes evaluated in this paper, covering diverse temporal trends as in BackTime (Lin et al., 2024).

---

[3] https://github.com/MAZiqing/FEDformer
[4] https://github.com/thuml/TimesNet
[5] https://github.com/vsingh-group/SimpleTM

To evaluate TIMEGUARD under diverse attack scenarios, we consider three attack-pattern shapes **P** following the BackTime setup for a fair comparison (Lin et al., 2024). For each poisoned timestamp of the selected channel, the attacker injects the standardized attack pattern into the forecasting horizon. The three pattern shapes (cone, up-trend, and up-and-down) are illustrated in Figure 7.

### F.6. TIMEGUARD Settings

For TIMEGUARD implementation, we follow the training pipeline of BackTime (Lin et al., 2024) as closely as possible to ensure a fair comparison. Unless otherwise specified, we use Adam (Kingma, 2014) with learning rate $1 \times 10^{-4}$ for both the forecaster $f_\theta$ and the backcaster $b_\phi$, batch size 64, and `SmoothL1Loss` as the default training loss. We adopt the default input/output window lengths $L_{in}$=12 and $L_{out}$=12. To match BackTime's default budget of 100 training epochs, we set Stage I and Stage II to $T_1$=10 and $T_2$=90 epochs, respectively. We additionally train the backcaster $b_\phi$ for $T_b$=10 epochs.

We set the initial reliable-pool ratio to $\alpha$=0.2 and the final ratio to $\beta$=0.5, and use a linear schedule for $\gamma$ that increases from $\alpha$ to $\beta$ throughout Stage II. We grid-search the scaling factor $\pi \in \{1.25, 1.5\}$ and the neighborhood size $K \in \{20, 32\}$. We use a 6:2:2 train/validation/test split and report performance on the test set.

### F.7. Baseline Defenses and TSF Adaptation

Since TSF-specific backdoor defenses remain limited, we adapt **13** representative defenses originally proposed for classification, spanning all four stages of the model life cycle and covering diverse defense paradigms (Wu et al., 2025a; Li et al., 2022a). For fairness, we start from each method's official (or widely used) implementation and make only the minimal modifications required to support forecasting.

In general, we replace accuracy-based criteria with MAE-based counterparts and substitute the entropy loss with a regression loss. For inference-time and input-transformation defenses, we tailor the perturbation/augmentation operators to time-series inputs; otherwise, we keep the original procedures unchanged. By default, we follow BackdoorBench implementations when available (Wu et al., 2025a); for methods not included, we adapt the authors' original repositories as fair as possible. Below, we summarize the key adaptation choices and the settings that differ from the original defaults, grouped by life-cycle stage.

**Pre-training-stage defenses.**

- **Spectral** (Tran et al., 2018). Spectral detects poisons by SVD-based outlier scoring in learned representations within each label group, removing top-scoring points before retraining. For TSF, we use penultimate-layer sample representations, flatten them, obtain pseudo-labels via $k$-means, and apply the original per-cluster scoring/removal. We tune $k \in \{5, 10, 20\}$ and use the best-performing setting.

- **TED** (Mo et al., 2024). TED flags backdoor samples by tracking how a sample's neighborhood structure evolves across layers: at selected layers, it records the rank of the nearest neighbor from the predicted group and uses the resulting rank trajectory for PCA-based outlier detection. For TSF, we assign pseudo-labels via $k$-means (as in Spectral) and compute rank trajectories within each cluster using flattened layer representations; we extract features from $M$ evenly spaced layers, with $M$=20 for SimpleTM and $M$=5 for FEDformer/TimesNet due to memory limits, and tune $k \in \{5, 10, 20\}$.

- **TED++** (Le et al., 2025). TED++ extends TED by explicitly modelling a layer-wise tubular neighbourhood around each class's hidden-feature submanifold, then applying Locally Adaptive Ranking (LAR) that assigns worst-case ranks to activations falling outside the tube. It aggregates the LAR ranks across layers into a trajectory and flags outliers using a PCA reconstruction-error test. For TSF, we use the same adaptation settings as TED.

**In-training-stage defenses.**

- **ABL** (Li et al., 2021a). ABL identifies suspicious easy-to-fit poisoned samples from training dynamics and then performs an unlearning stage to suppress their influence. For TSF, we replace the cross-entropy loss with its regression counterparts and otherwise follow the original procedure, using learning rate $10^{-4}$ for standard training and $10^{-5}$ for unlearning, which is the same as the TSF training pipeline of BackTime (Lin et al., 2024).

- **PDB** (Wei et al., 2024). PDB is a model-agnostic defense that mitigates unknown backdoors by proactively injecting a defender-chosen backdoor: it trains on $(\mathbf{x} \oplus \Delta_1, h(\mathbf{y}))$ with a reversible mapping $h$ and an auxiliary augmentation term (weight $\lambda_2$), then stamps $\Delta_1$ and applies $h^{-1}$ at inference. For TSF, we set $h(\mathbf{y}) = \mathbf{y} + \delta$ and $h^{-1}(\mathbf{y}) = \mathbf{y} - \delta$ on the target window, and use a fixed defensive trigger of value $-1$ (after normalization) over a specified span across all channels; we note this unrealistically assumes the defender knows the trigger length, otherwise performance degrades

substantially. We tune $\lambda_2 \in \{0.0, 0.1, 1\}$ and $\delta \in \{0.001, 0.01, 0.1\}$.

- **ESTI** (Yu et al., 2025). ESTI is a two-stage training-time defense that iteratively splits data into clean/poison pools using a KDE-based loss threshold (via benign vs. backdoor-sensitive training), and then isolates the suspected poison by training a trap model on a trap label. For TSF, we replace classification loss with per-window forecasting loss (SmoothL1) for KDE splitting, set the base learning rate to $10^{-4}$, and keep the original relative scaling of stage-specific learning rates.

**Post-training-stage defenses.**

- **Fine-tuning** (Gu et al., 2019). Fine-tuning is a post-training repair baseline that continues training the (potentially backdoored) model on a small trusted clean subset, with the goal of reducing backdoor behavior while preserving clean performance. In TSF, we fine-tune on $5\%$ clean training windows using the default forecasting loss and learning rate $10^{-4}$.

- **Fine-pruning** (Liu et al., 2018). Fine-pruning removes neurons that are rarely activated by clean inputs (ranked by average activation on a clean validation set) and then fine-tunes the pruned model to restore clean performance. For TSF, we prune units in ascending activation order on clean validation windows, iteratively removing a fraction $n$ per round until the validation MAE increases by more than $\delta$ relative to the unpruned model, and then fine-tune with the same setting as above. We grid-search $\delta \in \{0.01, 0.1, 0.2\}$ and $n \in \{0.01, 0.05\}$.

- **NAD** (Li et al., 2021b). NAD performs teacher–student fine-tuning: a teacher is first fine-tuned on a small trusted set, then the backdoored student is fine-tuned on the same set with an additional attention-distillation loss (weighted by $\beta$) that aligns intermediate attention maps. For TSF, we use the same $5\%$ clean windows as the fine-tuning baseline for 50 epochs of each model, and tune NAD by scaling each default $\beta$ in the released implementation by $\{0.1, 1, 100, 1000\}$.

- **IMS** (Dunnett et al., 2025). IMS mitigates backdoors by learning an invertible pruning mask via bilevel optimization: an inner step generates bounded perturbations through the inverse mask, and an outer step updates the mask to reduce backdoor behavior while preserving clean accuracy. For TSF, we replace the classification agree/disagree terms with regression versions based on $d = \mathrm{MSE}(\hat{\mathbf{y}}_1, \hat{\mathbf{y}}_2)$, i.e., $p_{\mathrm{agree}} = \exp(-\alpha d)$, $L_{\mathrm{agree}} = -\log(p_{\mathrm{agree}} + \epsilon)$, and $L_{\mathrm{dis}} = -\log(1 - p_{\mathrm{agree}} + \epsilon)$, and tune the perturbation norm bound in $\{0.02, 0.2, 1.0\}$.

**Inference-time defenses.**

- **STRIP** (Gao et al., 2019). STRIP perturbs a test input by repeatedly superimposing it with randomly sampled clean windows and measures prediction randomness; triggered inputs tend to yield abnormally low randomness under such perturbations. For TSF, we replace class entropy with a forecast-dispersion score based on the normalized variance of predictions across perturbed copies, averaged over channels and horizon. We sample 100 clean windows per test input from a pool of 10,000 and tune the mixing strength $\alpha \in \{0.1, 0.5, 1.0\}$.

- **TeCo** (Liu et al., 2023a). TeCo applies multiple input corruptions with increasing severity and flags inputs whose robustness responses are inconsistent across corruption types. For TSF, we replace hard-label "prediction change" with a deviation-based transition score computed from relative prediction distances. We use four time-series corruptions: Gaussian noise, late cutout, local permutation, and moving-average smoothing, each with four severity levels (noise $\{0.1, 0.2, 0.3, 0.4\}$; cutout ratio $\{0.1, 0.2, 0.3, 0.4\}$; permutation length $\{T/6, T/4, T/3, T/2\}$; smoothing kernel $\{3, 5, 7, 9\}$). The TeCo score is the dispersion of normalized prediction deviations across corruption families.

- **IBD-PSC** (Hou et al., 2024). IBD-PSC scales the affine parameters of late normalization layers by a factor $\omega$ and flags inputs whose predictions remain unusually consistent across scaled model variants. For TSF, we scale BN/LayerNorm affine parameters from the last layers backward and compute the score from prediction deviations. We select the scaling depth using relative clean-performance degradation and tune $\omega \in \{1.25, 1.5, 1.75\}$.

## G. Additional Experiment Results

### G.1. Full Defense Performance Results

**Complete results across datasets and attacks.** Tables 10–17 report the full performance of all baselines and TIMEGUARD under four representative TSF backdoor attacks (including Manhattan attack). Tables 10 and 11 summarize results over the three datasets , averaged across FEDformer (Zhou et al., 2022), SimpleTM (Chen et al., 2025a), and TimesNet (Wu et al., 2023), while Tables 12–17 provide per-architecture breakdowns. Overall, the appendix results are consistent with the findings and conclusions discussed in Sections 3 and 5.1.

*Table 10.* Full main results of backdoor defenses against TSF backdoor attacks, averaged over FEDformer, SimpleTM, and TimesNet. Best results are in **bold**. Lower $MAE_C$ indicates better performance, while higher $MAE_P$ and FDER indicate better performance.

| Dataset | Attack → 
 Defense ↓ | Random 
 $MAE_C$ ↓ | $MAE_P$ ↑ | FDER ↑ | Manhattan 
 $MAE_C$ ↓ | $MAE_P$ ↑ | FDER ↑ | FreqBack-TSF 
 $MAE_C$ ↓ | $MAE_P$ ↑ | FDER ↑ | BackTime 
 $MAE_C$ ↓ | $MAE_P$ ↑ | FDER ↑ |
|---|---|---|---|---|---|---|---|---|---|---|---|---|---|
| PEMS03 | No Defense | 17.634 | 17.772 | – | 17.722 | 20.266 | – | 17.583 | 14.683 | – | 17.607 | 14.201 | – |
| | Spectral | 18.389 | 18.356 | 0.502 | 19.444 | 20.417 | 0.475 | 18.765 | 14.027 | 0.475 | 18.666 | 15.245 | 0.539 |
| | TED | 18.434 | 20.063 | 0.528 | 19.427 | 20.298 | 0.467 | 18.785 | 13.984 | 0.473 | 18.606 | 13.953 | 0.495 |
| | TED++ | 19.197 | 19.184 | 0.499 | 18.992 | 20.659 | 0.479 | 18.706 | 13.445 | 0.473 | 18.565 | 14.541 | 0.513 |
| | Fine-tuning | 19.003 | 30.909 | 0.625 | 19.661 | 30.995 | 0.608 | 18.837 | 22.479 | 0.641 | 18.934 | 18.196 | 0.594 |
| | Fine-pruning | 19.020 | 31.643 | 0.633 | 19.595 | 34.447 | 0.624 | 19.073 | 23.543 | 0.647 | 18.686 | 19.736 | 0.623 |
| | NAD | 18.795 | 26.809 | 0.600 | 19.260 | 26.181 | 0.566 | 18.539 | 20.297 | 0.614 | 18.584 | 18.158 | 0.600 |
| | IMS | 19.239 | 17.731 | 0.466 | 19.370 | 20.178 | 0.466 | 18.521 | 14.570 | 0.479 | 18.418 | 14.351 | 0.509 |
| | ABL | 19.637 | 19.104 | 0.493 | 19.649 | 20.106 | 0.462 | 18.649 | 15.055 | 0.501 | 18.761 | 14.481 | 0.509 |
| | PDB | 18.630 | 54.690 | 0.693 | 19.308 | 60.477 | 0.708 | 19.512 | 26.014 | 0.652 | 18.967 | 22.397 | 0.639 |
| | ESTI | 19.910 | 17.186 | 0.454 | 19.460 | 18.960 | 0.458 | 18.793 | 14.684 | 0.475 | 19.219 | 15.897 | 0.532 |
| | TIMEGUARD | **17.928** | **104.677** | **0.868** | **17.850** | **97.370** | **0.854** | **17.628** | **57.759** | **0.847** | 18.048 | **39.303** | **0.808** |
| Weather | No Defense | 11.210 | 14.991 | – | 11.506 | 38.944 | – | 10.115 | 13.449 | – | 10.768 | 15.913 | – |
| | Spectral | 11.189 | 20.422 | 0.628 | 12.454 | 44.360 | 0.528 | 11.993 | 14.439 | 0.492 | 14.745 | 20.389 | 0.488 |
| | TED | 12.131 | 21.282 | 0.618 | 11.826 | 38.960 | 0.500 | 14.691 | 16.245 | 0.501 | 14.682 | 24.410 | 0.539 |
| | TED++ | 15.968 | 32.296 | 0.644 | 14.984 | 42.390 | 0.466 | 13.633 | 19.164 | 0.585 | 13.221 | 19.713 | 0.498 |
| | Fine-tuning | 12.027 | 41.019 | 0.716 | 11.808 | 71.443 | 0.711 | 13.045 | 53.864 | 0.770 | 11.589 | 51.120 | 0.743 |
| | Fine-pruning | 11.759 | 44.333 | 0.733 | 11.655 | 74.261 | 0.727 | 12.054 | 51.888 | 0.799 | 11.493 | 48.343 | 0.762 |
| | NAD | 11.804 | 27.080 | 0.646 | 11.687 | 69.082 | 0.711 | 11.631 | 39.104 | 0.745 | 11.920 | 43.684 | 0.720 |
| | IMS | 11.207 | 14.947 | 0.502 | 11.514 | 39.117 | 0.501 | 10.110 | 13.194 | 0.500 | 10.770 | 15.929 | 0.501 |
| | ABL | 13.845 | 20.264 | 0.527 | 15.081 | 43.216 | 0.472 | 13.671 | 18.693 | 0.529 | 13.047 | 20.018 | 0.539 |
| | PDB | 12.305 | 91.237 | 0.841 | 12.540 | 86.136 | 0.745 | 14.406 | 58.349 | 0.784 | 11.732 | 56.439 | 0.827 |
| | ESTI | 15.731 | 20.971 | 0.569 | 16.342 | 82.196 | 0.672 | 14.102 | 81.121 | 0.663 | 13.441 | 20.086 | 0.507 |
| | TIMEGUARD | **10.587** | **177.583** | **0.942** | **10.986** | **101.476** | **0.800** | **10.804** | **188.781** | **0.919** | 10.716 | **66.534** | **0.874** |
| ETTm1 | No Defense | 1.144 | 1.059 | – | 1.142 | 1.438 | – | 1.117 | 0.752 | – | 1.114 | 0.805 | – |
| | Spectral | 1.259 | 1.165 | 0.505 | 1.288 | 1.490 | 0.494 | 1.215 | 0.927 | 0.552 | 1.218 | 0.930 | 0.534 |
| | TED | **1.226** | 1.208 | 0.527 | 1.270 | 1.462 | 0.479 | 1.200 | 0.839 | 0.526 | **1.195** | 0.955 | 0.529 |
| | TED++ | 1.270 | 1.202 | 0.516 | 1.264 | 1.409 | 0.477 | **1.194** | 0.889 | 0.536 | 1.219 | 0.945 | 0.524 |
| | Fine-tuning | 1.269 | 1.895 | 0.664 | 1.265 | 2.603 | 0.676 | 1.254 | 1.365 | 0.658 | 1.249 | 1.286 | 0.623 |
| | Fine-pruning | 1.266 | 1.931 | 0.671 | 1.262 | 2.774 | 0.688 | 1.243 | 1.330 | 0.664 | 1.241 | 1.291 | 0.636 |
| | NAD | 1.276 | 1.555 | 0.607 | 1.226 | 2.137 | 0.624 | 1.235 | 1.125 | 0.613 | 1.244 | 1.208 | 0.579 |
| | IMS | 1.284 | 1.166 | 0.498 | **1.142** | 1.452 | 0.504 | 1.199 | 0.847 | 0.518 | 1.202 | 1.005 | 0.545 |
| | ABL | 1.351 | 1.341 | 0.529 | 1.362 | 1.616 | 0.474 | 1.307 | 1.143 | 0.582 | 1.256 | 1.014 | 0.526 |
| | PDB | 1.230 | 2.972 | 0.766 | 1.353 | 3.669 | 0.681 | 1.294 | 1.418 | 0.663 | 1.274 | 1.422 | 0.648 |
| | ESTI | 1.390 | 2.409 | 0.637 | 1.356 | 1.952 | 0.541 | 1.218 | 1.082 | 0.607 | 1.244 | 1.075 | 0.551 |
| | TIMEGUARD | 1.235 | **6.481** | **0.881** | 1.250 | **6.651** | **0.849** | 1.321 | **2.053** | **0.736** | 1.268 | **1.443** | **0.652** |

*Table 11.* Detection performance comparison of inference-time defenses on three datasets, averaged over FEDformer, SimpleTM, and TimesNet. Best results are in **bold**. Higher AUC and F1 indicates better detection performance.

| Dataset | Defense | Random 
 AUC ↑ | F1 ↑ | Manhattan 
 AUC ↑ | F1 ↑ | FreqBack-TSF 
 AUC ↑ | F1 ↑ | BackTime 
 AUC ↑ | F1 ↑ | AVERAGE 
 AUC ↑ | F1 ↑ |
|---|---|---|---|---|---|---|---|---|---|---|---|
| **PEMS03** | No Defense | 0.500 | 0.500 | 0.500 | 0.500 | 0.500 | 0.500 | 0.500 | 0.500 | 0.500 | 0.500 |
| | STRIP | 0.518 | 0.532 | 0.523 | 0.537 | **0.481** | 0.513 | **0.501** | 0.516 | 0.506 | 0.525 |
| | TeCo | **0.563** | **0.564** | **0.563** | **0.563** | 0.431 | 0.506 | 0.478 | 0.512 | **0.509** | **0.536** |
| | IBD-PSC | 0.364 | 0.514 | 0.402 | 0.522 | 0.416 | **0.519** | 0.486 | **0.535** | 0.417 | 0.523 |
| **Weather** | No Defense | 0.500 | 0.500 | 0.500 | 0.500 | 0.500 | 0.500 | 0.500 | 0.500 | 0.500 | 0.500 |
| | STRIP | 0.300 | 0.510 | 0.461 | 0.518 | **0.589** | 0.591 | 0.497 | 0.531 | 0.462 | 0.538 |
| | TeCo | **0.581** | **0.590** | 0.458 | 0.517 | 0.466 | 0.534 | **0.547** | **0.574** | **0.513** | **0.554** |
| | IBD-PSC | 0.317 | 0.519 | **0.521** | **0.546** | 0.369 | 0.556 | 0.390 | 0.534 | 0.399 | 0.539 |
| **ETTm1** | No Defense | 0.500 | 0.500 | 0.500 | 0.500 | 0.500 | 0.500 | 0.500 | 0.500 | 0.500 | 0.500 |
| | STRIP | 0.490 | 0.525 | 0.448 | 0.506 | 0.480 | 0.507 | 0.477 | 0.506 | 0.474 | 0.511 |
| | TeCo | **0.614** | **0.591** | **0.519** | **0.544** | **0.640** | **0.612** | **0.524** | **0.521** | **0.574** | **0.567** |
| | IBD-PSC | 0.378 | 0.513 | 0.464 | 0.523 | 0.497 | 0.532 | 0.486 | 0.518 | 0.456 | 0.522 |

*Table 12.* Full main results of backdoor defenses against TSF backdoor attacks on *FEDformer* model. Best results are in **bold**. Lower MAE$_C$ indicates better performance, while higher MAE$_P$ and FDER indicate better performance.

| Dataset | Attack → Defense ↓ | Random | | | Manhattan | | | FreqBack-TSF | | | BackTime | | |
|---|---|---|---|---|---|---|---|---|---|---|---|---|---|
| | | MAE$_C$ ↓ | MAE$_P$ ↑ | FDER ↑ | MAE$_C$ ↓ | MAE$_P$ ↑ | FDER ↑ | MAE$_C$ ↓ | MAE$_P$ ↑ | FDER ↑ | MAE$_C$ ↓ | MAE$_P$ ↑ | FDER ↑ |
| PEMS03 | No Defense | 16.286 | 14.959 | – | 16.411 | 17.984 | – | 16.179 | 9.436 | – | 16.093 | 10.760 | – |
| | Spectral | 16.930 | 15.658 | 0.503 | 16.567 | 18.960 | 0.521 | 16.232 | 9.627 | 0.508 | 16.484 | 11.221 | 0.509 |
| | TED | 16.607 | 15.402 | 0.505 | 16.667 | 17.639 | 0.492 | 16.378 | 9.496 | 0.497 | **16.284** | 10.828 | 0.497 |
| | TED++ | 18.093 | 19.247 | 0.561 | 17.542 | 18.627 | 0.485 | 16.103 | 9.257 | 0.500 | 16.332 | 10.882 | 0.498 |
| | Fine-tuning | 16.758 | 48.550 | 0.832 | 16.887 | 41.641 | 0.770 | 16.835 | 18.616 | 0.727 | 16.414 | 17.767 | 0.687 |
| | Fine-pruning | 16.836 | 49.054 | 0.831 | 16.951 | 51.561 | 0.810 | 16.871 | 20.123 | 0.745 | 16.408 | 21.445 | 0.740 |
| | NAD | 16.599 | 39.003 | 0.799 | 16.616 | 32.060 | 0.713 | 16.558 | 15.765 | 0.689 | 16.377 | 17.941 | 0.691 |
| | IMS | **16.286** | 14.953 | 0.500 | **16.411** | 17.982 | 0.500 | **16.179** | 9.430 | 0.500 | 16.684 | 13.569 | 0.586 |
| | ABL | 17.591 | 18.677 | 0.562 | 17.141 | 17.898 | 0.479 | 16.990 | 10.627 | 0.532 | 16.803 | 11.183 | 0.498 |
| | PDB | 16.774 | 25.809 | 0.696 | 17.254 | 37.089 | 0.733 | 16.914 | 17.076 | 0.702 | 17.040 | 14.511 | 0.601 |
| | ESTI | 18.896 | 15.727 | 0.455 | 18.102 | 17.401 | 0.453 | 16.432 | 9.199 | 0.492 | **16.284** | 11.212 | 0.514 |
| | TIMEGUARD | 16.607 | **100.436** | **0.916** | 16.578 | **94.212** | **0.900** | 16.496 | **38.147** | **0.867** | 16.840 | **41.232** | **0.847** |
| Weather | No Defense | 9.282 | 13.400 | – | 8.781 | 22.145 | – | 9.434 | 9.423 | – | 9.609 | 8.020 | – |
| | Spectral | 9.286 | 18.636 | 0.640 | 9.495 | 22.391 | 0.468 | 10.107 | 7.347 | 0.467 | 9.460 | 8.280 | 0.516 |
| | TED | 9.161 | 19.594 | 0.658 | 9.271 | 19.516 | 0.474 | 9.670 | 6.263 | 0.488 | 9.775 | 10.295 | 0.602 |
| | TED++ | 9.412 | 45.224 | 0.845 | 10.853 | 21.064 | 0.405 | 9.506 | 19.342 | 0.753 | 9.517 | 8.135 | 0.507 |
| | Fine-tuning | 9.174 | 77.684 | 0.914 | 9.252 | 71.884 | 0.820 | 9.785 | **85.835** | 0.927 | 9.517 | **69.535** | **0.942** |
| | Fine-pruning | 9.155 | 86.216 | 0.922 | 9.127 | 72.853 | 0.829 | 9.802 | 70.408 | 0.914 | 9.600 | 58.004 | 0.931 |
| | NAD | 9.111 | 44.531 | 0.850 | 9.032 | 67.813 | 0.823 | 9.801 | 60.086 | 0.903 | 9.584 | 55.976 | 0.928 |
| | IMS | 9.282 | 13.020 | 0.500 | 8.785 | 22.210 | 0.501 | 9.430 | 8.868 | 0.500 | 9.610 | 8.029 | 0.501 |
| | ABL | 10.044 | 11.555 | 0.462 | 9.633 | 22.443 | 0.462 | 9.823 | 7.480 | 0.480 | 11.159 | 12.711 | 0.615 |
| | PDB | 9.890 | 41.380 | 0.807 | 9.619 | 54.644 | 0.754 | 9.609 | 30.494 | 0.836 | 10.254 | 35.951 | 0.857 |
| | ESTI | **9.076** | 23.477 | 0.715 | **8.569** | 102.159 | 0.892 | **8.440** | 5.191 | 0.500 | **8.882** | 4.855 | 0.500 |
| | TIMEGUARD | 9.162 | **102.996** | **0.935** | 9.584 | 96.013 | 0.843 | 9.536 | 76.651 | **0.933** | 10.089 | 43.244 | 0.883 |
| ETTm1 | No Defense | 1.121 | 1.218 | – | 1.109 | 1.662 | – | 1.111 | 0.671 | – | 1.085 | 0.911 | – |
| | Spectral | 1.134 | 1.306 | 0.528 | 1.142 | 1.826 | 0.531 | 1.138 | 0.794 | 0.565 | 1.160 | 0.775 | 0.468 |
| | TED | **1.096** | 1.201 | 0.500 | 1.128 | 1.695 | 0.502 | 1.100 | 0.566 | 0.500 | 1.125 | 0.997 | 0.525 |
| | TED++ | 1.130 | 1.258 | 0.512 | 1.145 | 1.498 | 0.484 | **1.088** | 0.708 | 0.526 | 1.106 | 0.851 | 0.490 |
| | Fine-tuning | 1.180 | 2.273 | 0.707 | 1.173 | 2.534 | 0.645 | 1.217 | 1.759 | 0.766 | 1.191 | 1.698 | 0.687 |
| | Fine-pruning | 1.181 | 2.229 | 0.701 | 1.161 | 2.612 | 0.660 | 1.176 | 1.541 | 0.754 | 1.177 | 1.659 | 0.686 |
| | NAD | 1.155 | 1.994 | 0.680 | 1.148 | 2.403 | 0.637 | 1.163 | 1.310 | 0.721 | 1.161 | 1.667 | 0.694 |
| | IMS | 1.121 | 1.229 | 0.504 | **1.109** | 1.665 | 0.501 | 1.110 | 0.670 | 0.500 | **1.080** | 1.103 | 0.587 |
| | ABL | 1.275 | 1.630 | 0.566 | 1.345 | 2.052 | 0.507 | 1.327 | 1.492 | 0.693 | 1.296 | 1.152 | 0.523 |
| | PDB | 1.142 | 3.479 | 0.816 | 1.343 | 2.681 | 0.603 | 1.220 | 1.078 | 0.644 | 1.261 | 1.715 | 0.664 |
| | ESTI | 1.301 | 1.464 | 0.514 | 1.398 | 1.764 | 0.426 | 1.193 | 0.888 | 0.587 | 1.261 | 1.406 | 0.606 |
| | TIMEGUARD | 1.220 | **6.664** | **0.868** | 1.213 | **7.138** | **0.841** | 1.298 | **1.912** | **0.752** | 1.256 | **1.304** | **0.582** |

*Table 13.* Detection performance comparison of inference-time defenses on three datasets on *FEDformer* model. Best results are in **bold**. Higher AUC and F1 indicates better detection performance.

| Dataset | Defense | Random | | Manhattan | | FreqBack-TSF | | BackTime | | AVERAGE | |
|---|---|---|---|---|---|---|---|---|---|---|---|
| | | AUC ↑ | F1 ↑ | AUC ↑ | F1 ↑ | AUC ↑ | F1 ↑ | AUC ↑ | F1 ↑ | AUC ↑ | F1 ↑ |
| PEMS03 | No Defense | 0.500 | 0.500 | 0.500 | 0.500 | 0.500 | 0.500 | 0.500 | 0.500 | 0.500 | 0.500 |
| | STRIP | 0.502 | 0.522 | 0.523 | 0.532 | 0.500 | 0.520 | 0.504 | 0.517 | **0.507** | 0.523 |
| | TeCo | **0.573** | **0.557** | **0.577** | **0.559** | 0.403 | 0.500 | 0.465 | 0.500 | 0.505 | 0.529 |
| | IBD-PSC | 0.284 | 0.500 | 0.398 | 0.507 | **0.552** | **0.555** | **0.627** | **0.603** | 0.465 | **0.541** |
| Weather | No Defense | 0.500 | 0.500 | 0.500 | 0.500 | 0.500 | 0.500 | 0.500 | 0.500 | 0.500 | 0.500 |
| | STRIP | 0.422 | 0.507 | 0.481 | 0.524 | 0.526 | 0.547 | **0.527** | **0.539** | 0.489 | 0.529 |
| | TeCo | **0.599** | **0.587** | **0.554** | **0.541** | **0.587** | **0.571** | 0.442 | 0.521 | **0.546** | **0.555** |
| | IBD-PSC | 0.331 | 0.501 | 0.505 | 0.520 | 0.455 | 0.567 | 0.362 | 0.502 | 0.413 | 0.523 |
| ETTm1 | No Defense | 0.500 | 0.500 | 0.500 | 0.500 | 0.500 | 0.500 | 0.500 | 0.500 | 0.500 | 0.500 |
| | STRIP | 0.406 | 0.501 | 0.401 | 0.501 | 0.496 | 0.513 | 0.466 | 0.507 | 0.442 | 0.506 |
| | TeCo | **0.674** | **0.630** | **0.577** | **0.559** | **0.560** | **0.561** | **0.504** | 0.507 | **0.579** | **0.564** |
| | IBD-PSC | 0.405 | 0.517 | 0.384 | 0.510 | 0.511 | 0.519 | 0.491 | **0.522** | 0.448 | 0.517 |

*Table 14.* Full main results of backdoor defenses against TSF backdoor attacks on *SimpleTM* model. Best results are in **bold**. Lower MAE$_C$ indicates better performance, while higher MAE$_P$ and FDER indicate better performance.

| Dataset | Attack → | **Random** | | | **Manhattan** | | | **FreqBack-TSF** | | | **BackTime** | | |
|---|---|---|---|---|---|---|---|---|---|---|---|---|---|
| | Defense ↓ | MAE$_C$ ↓ | MAE$_P$ ↑ | FDER ↑ | MAE$_C$ ↓ | MAE$_P$ ↑ | FDER ↑ | MAE$_C$ ↓ | MAE$_P$ ↑ | FDER ↑ | MAE$_C$ ↓ | MAE$_P$ ↑ | FDER ↑ |
| PEMS03 | No Defense | 17.510 | 19.007 | – | 17.539 | 22.532 | – | 17.335 | 15.468 | – | 17.268 | 9.131 | – |
| | Spectral | 17.746 | 20.820 | 0.537 | 17.596 | 22.835 | 0.505 | 18.015 | 13.304 | 0.481 | 17.621 | 13.971 | 0.663 |
| | TED | 17.578 | 25.544 | 0.626 | 17.529 | 22.801 | 0.506 | 17.707 | 13.417 | 0.489 | 17.671 | 10.242 | 0.543 |
| | TED++ | 17.807 | 19.156 | 0.496 | 17.529 | 22.863 | 0.507 | 17.785 | 12.099 | 0.487 | 17.328 | 11.401 | 0.598 |
| | Fine-tuning | 17.397 | 23.898 | 0.602 | 17.619 | 27.930 | 0.594 | 17.464 | 27.846 | 0.719 | 17.355 | 13.287 | 0.654 |
| | Fine-pruning | **17.396** | 25.735 | 0.631 | 17.665 | 28.338 | 0.599 | 17.460 | 29.782 | 0.737 | 17.363 | 13.950 | 0.670 |
| | NAD | 17.516 | 21.572 | 0.559 | 17.571 | 24.708 | 0.543 | 17.411 | 25.277 | 0.692 | 17.299 | 13.245 | 0.654 |
| | IMS | 17.513 | 19.014 | 0.500 | 17.539 | 22.540 | 0.500 | 17.335 | 15.480 | 0.500 | **16.520** | 8.206 | 0.500 |
| | ABL | 17.740 | 19.722 | 0.512 | 17.717 | 23.135 | 0.508 | 17.665 | 16.424 | 0.520 | 17.465 | 11.219 | 0.587 |
| | PDB | 17.740 | 117.954 | 0.913 | 17.527 | 120.846 | 0.907 | 18.889 | 38.543 | 0.758 | 18.025 | 26.746 | 0.808 |
| | ESTI | **17.396** | 16.826 | 0.500 | 17.380 | 20.347 | 0.500 | 17.188 | 15.309 | 0.500 | 18.952 | 14.763 | 0.646 |
| | TIMEGUARD | 17.489 | **173.700** | **0.945** | 17.284 | **157.870** | **0.929** | 16.780 | **94.224** | **0.918** | 17.243 | **36.626** | **0.875** |
| Weather | No Defense | 7.693 | 18.888 | – | 7.711 | 64.020 | – | 7.761 | 19.205 | – | 7.752 | 15.301 | – |
| | Spectral | 7.868 | 26.086 | 0.627 | 7.792 | 65.176 | 0.504 | 7.875 | 19.109 | 0.493 | 7.851 | 15.079 | 0.494 |
| | TED | 7.764 | 28.075 | 0.659 | 7.676 | 63.941 | 0.500 | 7.836 | 16.691 | 0.495 | 7.979 | 15.343 | 0.487 |
| | TED++ | 7.729 | 19.507 | 0.514 | 7.674 | 61.888 | 0.500 | 7.862 | 16.510 | 0.494 | 7.849 | 15.187 | 0.494 |
| | Fine-tuning | 7.850 | 23.020 | 0.580 | 8.089 | 72.225 | 0.533 | 8.148 | 42.259 | 0.749 | 7.960 | 17.028 | 0.538 |
| | Fine-pruning | 7.860 | 25.415 | 0.618 | 8.092 | 77.976 | 0.566 | 8.005 | 45.018 | 0.771 | 8.009 | 19.544 | 0.593 |
| | NAD | 7.835 | 19.659 | 0.511 | 8.056 | 73.444 | 0.543 | 7.866 | 27.959 | 0.650 | 8.095 | 16.989 | 0.529 |
| | IMS | **7.692** | 19.130 | 0.506 | 7.712 | 64.484 | 0.504 | **7.755** | 18.969 | 0.500 | 7.753 | 15.263 | 0.500 |
| | ABL | 7.887 | 35.276 | 0.720 | 8.007 | 64.383 | 0.484 | 7.927 | 21.299 | 0.539 | 8.021 | 16.695 | 0.525 |
| | PDB | 7.836 | 192.806 | 0.942 | 7.902 | 108.511 | 0.693 | 8.041 | 98.519 | 0.885 | 8.040 | 50.108 | 0.829 |
| | ESTI | 7.733 | 16.628 | 0.497 | **7.469** | 74.586 | 0.571 | 7.896 | 210.368 | 0.946 | **7.689** | 15.785 | 0.515 |
| | TIMEGUARD | 7.716 | **351.059** | **0.972** | 7.699 | **114.876** | **0.721** | 7.973 | **416.357** | **0.964** | 7.934 | **69.357** | **0.878** |
| ETTm1 | No Defense | 1.206 | 0.966 | – | 1.203 | 1.558 | – | 1.165 | 0.870 | – | 1.170 | 0.508 | – |
| | Spectral | 1.215 | 1.107 | 0.560 | 1.224 | 1.317 | 0.492 | 1.186 | 0.922 | 0.519 | 1.185 | 0.602 | 0.572 |
| | TED | 1.189 | 1.318 | 0.633 | 1.199 | 1.448 | 0.500 | 1.164 | 0.902 | 0.518 | 1.174 | 0.517 | 0.507 |
| | TED++ | 1.189 | 1.197 | 0.596 | 1.190 | 1.474 | 0.500 | 1.165 | 0.893 | 0.513 | 1.181 | 0.549 | 0.533 |
| | Fine-tuning | 1.210 | 2.075 | 0.766 | 1.186 | 3.142 | 0.752 | 1.182 | 1.171 | 0.621 | 1.186 | 0.683 | 0.622 |
| | Fine-pruning | 1.209 | 2.171 | 0.776 | 1.194 | 3.716 | 0.790 | 1.185 | 1.295 | 0.655 | 1.188 | 0.794 | 0.673 |
| | NAD | 1.215 | 1.341 | 0.636 | 1.195 | 2.557 | 0.695 | 1.170 | 1.005 | 0.565 | 1.183 | 0.501 | 0.495 |
| | IMS | 1.208 | 1.018 | 0.525 | 1.204 | 1.599 | 0.513 | 1.166 | 0.893 | 0.512 | 1.171 | 0.504 | 0.500 |
| | ABL | 1.229 | 1.061 | 0.536 | 1.226 | 1.564 | 0.492 | 1.182 | 0.944 | 0.532 | 1.190 | 0.505 | 0.492 |
| | PDB | **1.142** | 3.659 | 0.868 | **1.154** | 6.487 | 0.880 | **1.142** | 1.447 | 0.699 | **1.135** | 0.799 | 0.683 |
| | ESTI | 1.285 | 4.140 | 0.853 | 1.276 | 2.649 | 0.677 | 1.281 | 1.345 | 0.631 | 1.259 | 0.461 | 0.465 |
| | TIMEGUARD | 1.247 | **6.928** | **0.914** | 1.245 | **7.802** | **0.883** | 1.287 | **1.795** | **0.710** | 1.268 | **0.923** | **0.687** |

*Table 15.* Detection performance comparison of inference-time defenses on three datasets on *SimpleTM* model. Best results are in **bold**. Higher AUC and F1 indicates better detection performance.

| Dataset | Defense | **Random** | | **Manhattan** | | **FreqBack-TSF** | | **BackTime** | | **AVERAGE** | |
|---|---|---|---|---|---|---|---|---|---|---|---|
| | | AUC ↑ | F1 ↑ | AUC ↑ | F1 ↑ | AUC ↑ | F1 ↑ | AUC ↑ | F1 ↑ | AUC ↑ | F1 ↑ |
| **PEMS03** | No Defense | 0.500 | 0.500 | 0.500 | 0.500 | 0.500 | 0.500 | 0.500 | 0.500 | 0.500 | 0.500 |
| | STRIP | 0.517 | 0.529 | 0.506 | 0.516 | 0.486 | **0.517** | 0.494 | 0.518 | 0.501 | 0.520 |
| | TeCo | **0.680** | **0.628** | **0.670** | **0.627** | 0.510 | **0.517** | **0.541** | **0.535** | **0.600** | **0.577** |
| | IBD-PSC | 0.436 | 0.525 | 0.434 | 0.548 | 0.263 | 0.500 | 0.364 | 0.500 | 0.374 | 0.518 |
| **Weather** | No Defense | 0.500 | 0.500 | 0.500 | 0.500 | 0.500 | 0.500 | 0.500 | 0.500 | 0.500 | 0.500 |
| | STRIP | 0.308 | 0.521 | 0.395 | 0.515 | **0.737** | **0.701** | 0.420 | 0.515 | 0.465 | 0.563 |
| | TeCo | **0.747** | **0.680** | 0.463 | 0.506 | 0.439 | 0.531 | **0.770** | **0.700** | **0.605** | **0.604** |
| | IBD-PSC | 0.047 | 0.500 | **0.560** | **0.597** | 0.055 | 0.500 | 0.245 | 0.520 | 0.227 | 0.529 |
| **ETTm1** | No Defense | 0.500 | 0.500 | 0.500 | 0.500 | 0.500 | 0.500 | 0.500 | 0.500 | 0.500 | 0.500 |
| | STRIP | 0.464 | 0.500 | 0.447 | 0.500 | 0.485 | 0.506 | 0.491 | 0.509 | 0.472 | 0.504 |
| | TeCo | **0.533** | **0.539** | **0.594** | **0.572** | **0.597** | 0.575 | **0.541** | **0.533** | **0.566** | **0.555** |
| | IBD-PSC | 0.421 | 0.523 | 0.507 | 0.544 | 0.526 | **0.575** | 0.482 | 0.532 | 0.484 | 0.544 |

*Table 16.* Full main results of backdoor defenses against TSF backdoor attacks on *TimesNet* model. Best results are in **bold**. Lower MAE$_C$ indicates better performance, while higher MAE$_P$ and FDER indicate better performance.

| Dataset | Attack → Defense ↓ | Random MAE$_C$ ↓ | MAE$_P$ ↑ | FDER ↑ | Manhattan MAE$_C$ ↓ | MAE$_P$ ↑ | FDER ↑ | FreqBack-TSF MAE$_C$ ↓ | MAE$_P$ ↑ | FDER ↑ | BackTime MAE$_C$ ↓ | MAE$_P$ ↑ | FDER ↑ |
|---|---|---|---|---|---|---|---|---|---|---|---|---|---|
| PEMS03 | No Defense | 19.104 | 19.351 | – | 19.216 | 20.283 | – | 19.234 | 19.146 | – | 19.459 | 22.713 | – |
| | Spectral | 20.492 | 18.591 | 0.466 | 24.168 | 19.455 | 0.398 | 22.047 | 19.149 | 0.436 | 21.891 | 20.544 | 0.444 |
| | TED | 21.116 | 19.244 | 0.452 | 24.086 | 20.453 | 0.403 | 22.270 | 19.040 | 0.432 | 21.862 | 20.789 | 0.445 |
| | TED++ | 21.692 | 19.150 | 0.440 | 21.906 | 20.487 | 0.444 | 22.228 | 18.979 | 0.433 | 22.033 | 21.340 | 0.442 |
| | Fine-tuning | 22.852 | 20.279 | 0.441 | 24.476 | 23.412 | 0.459 | 22.211 | 20.975 | 0.477 | 23.032 | 23.534 | 0.440 |
| | Fine-pruning | 22.828 | 20.139 | 0.438 | 24.168 | 23.441 | 0.465 | 22.887 | 20.725 | 0.458 | 22.286 | 23.813 | 0.460 |
| | NAD | 22.270 | 19.851 | 0.442 | 23.592 | 21.775 | 0.442 | 21.647 | 19.849 | 0.462 | 22.076 | 23.288 | 0.453 |
| | IMS | 23.919 | 19.225 | 0.399 | 24.160 | 20.013 | 0.398 | 22.048 | 18.800 | 0.436 | 22.049 | 21.278 | 0.441 |
| | ABL | 23.579 | 18.912 | 0.405 | 24.091 | 19.283 | 0.399 | 21.293 | 18.113 | 0.452 | 22.014 | 21.043 | 0.442 |
| | PDB | 21.375 | 20.306 | 0.470 | 23.142 | 23.497 | 0.484 | 22.733 | 22.423 | 0.496 | 21.836 | 25.933 | 0.508 |
| | ESTI | 23.440 | 19.006 | 0.408 | 22.899 | 19.134 | 0.420 | 22.760 | 19.543 | 0.433 | 22.420 | 21.717 | 0.434 |
| | **TIMEGUARD** | **19.687** | **39.894** | **0.743** | **19.689** | **40.029** | **0.735** | **19.609** | **40.905** | **0.756** | **20.061** | **40.052** | **0.701** |
| Weather | No Defense | 16.653 | 12.684 | – | 18.026 | 30.666 | – | 13.148 | 11.719 | – | 14.943 | 24.417 | – |
| | Spectral | 16.412 | 16.542 | 0.617 | 20.076 | 45.515 | 0.612 | 17.997 | 16.863 | 0.518 | 26.925 | 37.809 | 0.455 |
| | TED | 19.469 | 16.176 | 0.536 | 18.531 | 33.423 | 0.528 | 26.568 | 25.782 | 0.520 | 26.293 | 47.592 | 0.528 |
| | TED++ | 30.764 | 32.157 | 0.573 | 26.426 | 44.219 | 0.494 | 23.532 | 21.640 | 0.509 | 22.297 | 35.818 | 0.494 |
| | Fine-tuning | 19.056 | 22.354 | 0.653 | 18.082 | 70.219 | 0.780 | 21.202 | 33.497 | 0.635 | 17.291 | 66.797 | 0.749 |
| | Fine-pruning | 18.261 | 21.368 | 0.659 | 17.745 | 71.955 | 0.787 | 18.355 | 40.239 | 0.713 | 16.871 | 67.479 | 0.762 |
| | NAD | 18.466 | 17.050 | 0.579 | 17.971 | 65.990 | 0.768 | 17.227 | 29.268 | 0.681 | 18.082 | 58.089 | 0.703 |
| | IMS | 16.648 | 12.691 | 0.500 | 18.044 | 30.657 | 0.499 | **13.144** | 11.745 | 0.501 | 14.948 | 24.494 | 0.501 |
| | ABL | 23.605 | 13.962 | 0.399 | 27.601 | 42.822 | 0.468 | 23.263 | 27.300 | 0.568 | 19.962 | 30.649 | 0.476 |
| | PDB | 19.188 | 39.525 | 0.773 | 20.099 | **95.252** | 0.787 | 25.566 | 46.034 | 0.630 | 16.903 | 83.259 | 0.795 |
| | ESTI | 30.383 | 22.809 | 0.496 | 32.988 | 69.844 | 0.554 | 25.970 | 27.804 | 0.542 | 23.750 | 39.619 | 0.506 |
| | **TIMEGUARD** | **14.883** | **78.694** | **0.919** | **15.675** | 93.538 | **0.836** | 14.902 | **73.334** | **0.861** | **14.125** | **87.000** | **0.860** |
| ETTm1 | No Defense | 1.106 | 0.992 | – | 1.113 | 1.094 | – | 1.074 | 0.714 | – | 1.086 | 0.996 | – |
| | Spectral | 1.427 | 1.080 | 0.428 | 1.497 | 1.327 | 0.460 | 1.320 | 1.067 | 0.572 | 1.308 | 1.412 | 0.563 |
| | TED | 1.394 | 1.106 | 0.448 | 1.484 | 1.243 | 0.435 | 1.337 | 1.049 | 0.561 | 1.287 | 1.352 | 0.554 |
| | TED++ | 1.491 | 1.152 | 0.440 | 1.458 | 1.253 | 0.446 | 1.329 | 1.065 | 0.569 | 1.369 | 1.434 | 0.550 |
| | Fine-tuning | 1.417 | 1.338 | 0.520 | 1.434 | 2.133 | 0.632 | 1.363 | 1.165 | 0.587 | 1.371 | 1.477 | 0.559 |
| | Fine-pruning | 1.407 | 1.392 | 0.536 | 1.430 | 1.994 | 0.615 | 1.368 | 1.153 | 0.583 | 1.359 | 1.421 | 0.549 |
| | NAD | 1.459 | 1.330 | 0.506 | 1.335 | 1.450 | 0.540 | 1.372 | 1.059 | 0.554 | 1.387 | 1.457 | 0.550 |
| | IMS | 1.523 | 1.250 | 0.466 | **1.114** | 1.093 | 0.500 | 1.320 | 0.980 | 0.542 | 1.355 | 1.409 | 0.548 |
| | ABL | 1.548 | 1.331 | 0.485 | 1.515 | 1.232 | 0.424 | 1.412 | 0.992 | 0.520 | 1.281 | 1.386 | 0.565 |
| | PDB | 1.407 | 1.779 | 0.614 | 1.561 | 1.838 | 0.559 | 1.519 | 1.728 | 0.647 | 1.426 | 1.752 | 0.597 |
| | ESTI | 1.583 | 1.624 | 0.544 | 1.395 | 1.443 | 0.520 | **1.180** | 1.014 | 0.603 | **1.213** | 1.359 | 0.581 |
| | **TIMEGUARD** | **1.239** | **5.852** | **0.862** | 1.291 | **5.013** | **0.822** | 1.378 | **2.453** | **0.744** | 1.279 | **2.103** | **0.688** |

*Table 17.* Detection performance comparison of inference-time defenses on three datasets on *TimesNet* model. Best results are in **bold**. Higher AUC and F1 indicates better detection performance.

| Dataset | Defense | Random AUC ↑ | F1 ↑ | Manhattan AUC ↑ | F1 ↑ | FreqBack-TSF AUC ↑ | F1 ↑ | BackTime AUC ↑ | F1 ↑ | AVERAGE AUC ↑ | F1 ↑ |
|---|---|---|---|---|---|---|---|---|---|---|---|
| PEMS03 | No Defense | 0.500 | 0.500 | 0.500 | 0.500 | 0.500 | 0.500 | 0.500 | 0.500 | 0.500 | 0.500 |
| | STRIP | **0.536** | **0.545** | **0.539** | **0.563** | **0.457** | **0.502** | **0.507** | **0.513** | **0.510** | **0.531** |
| | TeCo | 0.437 | 0.506 | 0.442 | 0.503 | 0.379 | 0.501 | 0.428 | 0.500 | 0.422 | 0.503 |
| | IBD-PSC | 0.372 | 0.516 | 0.375 | 0.511 | 0.434 | **0.502** | 0.468 | 0.500 | 0.412 | 0.507 |
| Weather | No Defense | 0.500 | 0.500 | 0.500 | 0.500 | 0.500 | 0.500 | 0.500 | 0.500 | 0.500 | 0.500 |
| | STRIP | 0.171 | 0.503 | **0.507** | 0.514 | 0.505 | 0.525 | 0.545 | 0.537 | 0.432 | 0.520 |
| | TeCo | 0.398 | 0.503 | 0.357 | 0.504 | 0.372 | 0.500 | 0.431 | 0.500 | 0.390 | 0.502 |
| | IBD-PSC | **0.573** | **0.555** | 0.497 | **0.521** | **0.596** | **0.599** | **0.563** | **0.580** | **0.557** | **0.564** |
| ETTm1 | No Defense | 0.500 | 0.500 | 0.500 | 0.500 | 0.500 | 0.500 | 0.500 | 0.500 | 0.500 | 0.500 |
| | STRIP | 0.601 | 0.573 | 0.497 | **0.518** | 0.458 | 0.502 | 0.475 | 0.503 | 0.508 | 0.524 |
| | TeCo | **0.633** | **0.604** | 0.386 | 0.500 | **0.763** | **0.700** | **0.527** | **0.523** | **0.577** | **0.582** |
| | IBD-PSC | 0.307 | 0.500 | **0.501** | 0.513 | 0.455 | 0.501 | 0.484 | 0.501 | 0.437 | 0.504 |

**Robustness against BadTime attack.** Beyond evaluating defenses against three SOTA attacks, we also evaluate TIME-GUARD against the recent TSF backdoor attack BadTime (Xiang et al., 2025). BadTime leverages inter-variable correlations, temporal lags, and data-driven initialization to construct distributed, lag-aware triggers for effective and stealthy attacks. For our BadTime implementation, we attempt to replicate the method using its default hyperparameters. For defense evaluation, after obtaining the fixed BadTime trigger, we poison the datasets and then apply the PDB and TIMEGUARD training pipelines. Table 18 shows that TIMEGUARD remains effective under this recent attack setting and achieves the best overall trade-off, attaining the highest FDER of 0.847 while maintaining competitive clean forecasting performance, even outperforming vanilla training in terms of clean $MAE_C$.

*Table 18.* Defense performance of PDB (Wei et al., 2024) and TIMEGUARD under BadTime (Xiang et al., 2025) a on PEMS03 dataset, where FEDFormer, SimpleTM, and TimesNet are the victim models. Best results are in **bold**.

| Model → | FEDformer | | | SimpleTM | | | TimesNet | | | AVERAGE | | |
|---|---|---|---|---|---|---|---|---|---|---|---|---|
| Defense ↓ | $MAE_C$ ↓ | $MAE_P$ ↑ | FDER ↑ | $MAE_C$ ↓ | $MAE_P$ ↑ | FDER ↑ | $MAE_C$ ↓ | $MAE_P$ ↑ | FDER ↑ | $MAE_C$ ↓ | $MAE_P$ ↑ | FDER ↑ |
| No Defense | 20.399 | 17.797 | – | 20.940 | 18.640 | – | 23.755 | 26.316 | – | 21.698 | 20.918 | – |
| PDB (Wei et al., 2024) | 19.372 | 36.976 | 0.759 | 21.455 | **38.732** | 0.747 | 24.390 | 37.878 | 0.640 | 21.739 | 37.862 | 0.715 |
| **TIMEGUARD** | **17.287** | **37.524** | **0.867** | **19.109** | 37.165 | **0.918** | **21.600** | **39.563** | **0.756** | **19.332** | **38.084** | **0.847** |

**Generalization to additional architectures.** Beyond the three backbone forecasters used in our main experiments, two Transformer-based models (FEDformer (Zhou et al., 2022) and SimpleTM (Chen et al., 2025a)) and one CNN-based model (TimesNet (Wu et al., 2023)), we further evaluate TIMEGUARD on a broader set of TSF architectures under the Random and BackTime attacks on PEMS03 dataset. Concretely, we include SegRNN (Lin et al., 2023) (RNN-based), SOFTS (Han et al., 2024) and TimeMixer (Wang et al., 2024) (MLP-based), and AutoTimes (Liu et al., 2024c) as an emerging LLM-based forecaster with two large LLM variants (GPT2 (Radford et al., 2019) and OPT-1.3B (Zhang et al., 2022)). As shown in Table 19, TIMEGUARD consistently attains $MAE_P$ above 32 and FDER above 0.68, while incurring at most a 10% relative increase in $MAE_C$ across two attacks. Specifically, on the LLM-based method (AutoTimes), TIMEGUARD yielding at least a $5.14\times$ $MAE_P$ gain with only a $3.8\%$ change in clean $MAE_C$. Overall, these results support that TIMEGUARD is architecture-agnostic and remains effective across diverse forecasting architectures.

*Table 19.* Defense performance across 8 models with different architectures under Random and BackTime attacks on PEMS03 dataset.

| Attack → | Random | | | | | BackTime | | | | |
|---|---|---|---|---|---|---|---|---|---|---|
| Defense → | No Defense | | TIMEGUARD | | | No Defense | | TIMEGUARD | | |
| Model ↓ | $MAE_C$ ↓ | $MAE_P$ ↑ | $MAE_C$ ↓ | $MAE_P$ ↑ | FDER ↑ | $MAE_C$ ↓ | $MAE_P$ ↑ | $MAE_C$ ↓ | $MAE_P$ ↑ | FDER ↑ |
| FEDformer (Zhou et al., 2022) | 16.286 | 14.959 | 16.607 | 100.436 | 0.916 | 16.093 | 10.760 | 16.840 | 41.232 | 0.847 |
| SimpleTM (Chen et al., 2025a) | 17.510 | 19.007 | 17.489 | 173.700 | 0.945 | 17.268 | 9.131 | 17.243 | 36.626 | 0.875 |
| TimesNet (Wu et al., 2023) | 19.104 | 19.351 | 19.687 | 39.894 | 0.743 | 19.459 | 22.713 | 20.061 | 40.052 | 0.701 |
| SegRNN (Lin et al., 2023) | 19.889 | 8.953 | 20.469 | 205.044 | 0.964 | 19.980 | 6.927 | 20.718 | 33.941 | 0.880 |
| SOFTS (Han et al., 2024) | 16.263 | 2.930 | 16.871 | 170.169 | 0.973 | 16.227 | 3.185 | 17.451 | 33.340 | 0.917 |
| TimeMixer (Wang et al., 2024) | 21.540 | 19.917 | 21.351 | 220.274 | 0.955 | 21.484 | 21.053 | 21.440 | 33.662 | 0.687 |
| AutoTimes$_{GPT2}$ (Liu et al., 2024c) | 20.984 | 19.346 | 21.292 | 215.993 | 0.948 | 21.006 | 6.239 | 21.805 | 32.046 | 0.884 |
| AutoTimes$_{OPT1B}$ (Liu et al., 2024c) | 20.911 | 22.946 | 21.054 | 221.067 | 0.945 | 20.921 | 6.162 | 21.196 | 33.931 | 0.903 |

**Defense performance under large-scale TSF foundation models.** Beyond AutoTimes$_{GPT2}$ and AutoTimes$_{OPT1B}$ (Liu et al., 2024c), we further evaluate AutoTimes$_{LLaMA7B}$, an AutoTimes variant built on the large-scale LLaMA-7B foundation model (Touvron et al., 2023). Table 20 shows that TIMEGUARD still outperforms PDB while keeping the total training time to $1.431\times$ that of undefended training and comparable to PDB, i.e., approximately 70,000 seconds. These results suggest that TIMEGUARD transfers beyond standard TSF backbones and remains effective for large-scale TSF foundation models.

*Table 20.* Defense performance and training time (in seconds) of PDB (Wei et al., 2024) and TIMEGUARD under BackTime on the PEMS03 dataset, using AutoTimes$_{LLaMA7B}$ (Liu et al., 2024c) as the victim model. Best results are shown in **bold**.

| Defense | Training time (s) ↓ | $MAE_C$ ↓ | $MAE_P$ ↑ | FDER ↑ |
|---|---|---|---|---|
| No Defense | 49027.6 | 20.977 | 6.004 | – |
| PDB (Wei et al., 2024) | **69997.6** | 22.657 | 25.798 | 0.847 |
| **TIMEGUARD** | 70162.8 | **21.385** | **32.792** | **0.899** |

**Generalization to different poisoning rates.** We evaluate the robustness of TIMEGUARD under varying attack budgets by adjusting the temporal poisoning rate $\eta_T$ and spatial poisoning rate $\eta_S$ using the BackTime attack on the PEMS03 dataset. As shown in Figures 8–10, TIMEGUARD consistently maintains strong defense effectiveness across all poisoning rates, with $MAE_P$ remaining above 35 for all three models. Additionally, clean performance stays within 5% even at high poisoning rates ($\eta_T = 0.04$, $\eta_S = 0.4$). These results demonstrate that TIMEGUARD is robust to varying poisoning rates while maintaining reasonable clean performance.

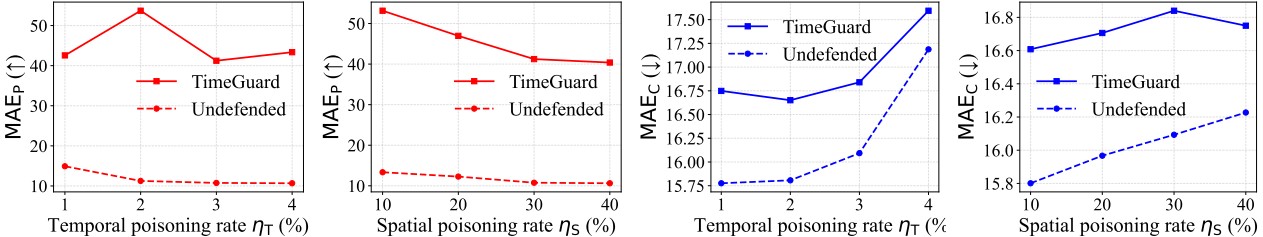

*Figure 8.* Defense performance of TIMEGUARD ($MAE_P$ and $MAE_C$) under varying temporal and spatial poisoning rates of the BackTime attack on the PEMS03 dataset with the *FEDformer* model.

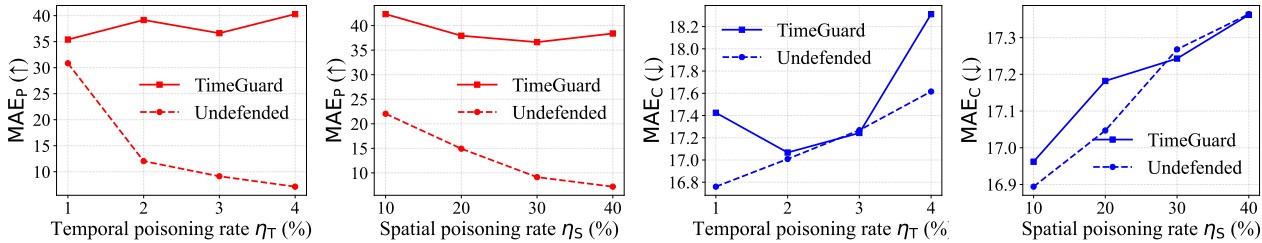

*Figure 9.* Defense performance of TIMEGUARD ($MAE_P$ and $MAE_C$) under varying temporal and spatial poisoning rates of the BackTime attack on the PEMS03 dataset with the *SimpleTM* model.

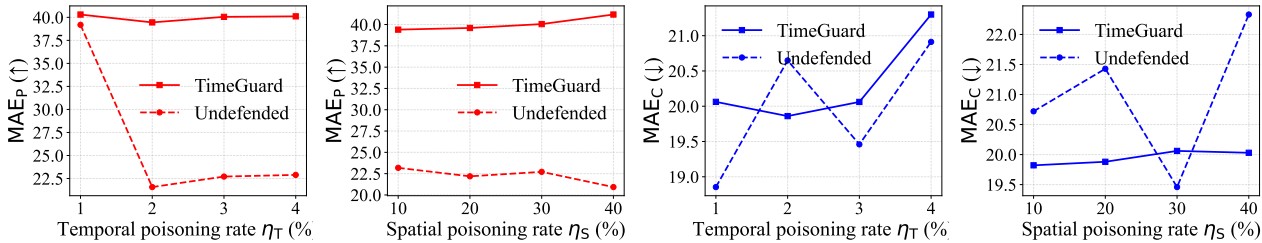

*Figure 10.* Defense performance of TIMEGUARD ($MAE_P$ and $MAE_C$) under varying temporal and spatial poisoning rates of the BackTime attack on the PEMS03 dataset with the *TimesNet* model.

**Generalization to the extreme case of full-channel poisoning.** Our motivation is strongest under partial-channel poisoning, which is the common setting in existing multivariate TSF backdoor attacks; as the channel poisoning ratio increases, attacks generally become less stealthy and easier to detect. Nevertheless, TIMEGUARD does not require poisoning to affect only a strict subset of channels: its channel-wise formulation remains applicable even when poisoning is dense across channels. To directly test the all-channel case, we evaluate TIMEGUARD on PEMS03 under BackTime with spatial poisoning ratio $\eta_S = 1.0$, meaning that all channels are poisoned. As shown in Table 21, TIMEGUARD remains effective in this setting and achieves the highest FDER of 0.748.

*Table 21.* Defense performance of PDB and TIMEGUARD under BackTime attack on the PEMS03 dataset with full-channel poisoning, i.e., $\eta_S = 1.0$, where FEDformer, SimpleTM, and TimesNet are used as victim models. Best results are shown in **bold**.

| Model → | FEDformer | | | SimpleTM | | | TimesNet | | | AVERAGE | | |
|---|---|---|---|---|---|---|---|---|---|---|---|---|
| Defense ↓ | $MAE_C$ ↓ | $MAE_P$ ↑ | FDER ↑ | $MAE_C$ ↓ | $MAE_P$ ↑ | FDER ↑ | $MAE_C$ ↓ | $MAE_P$ ↑ | FDER ↑ | $MAE_C$ ↓ | $MAE_P$ ↑ | FDER ↑ |
| No Defense | 18.025 | 11.586 | – | 18.567 | 5.817 | – | 27.473 | 21.503 | – | 21.355 | 12.969 | – |
| PDB (Wei et al., 2024) | 18.308 | 16.074 | 0.632 | **19.114** | 13.355 | **0.768** | 23.731 | 32.485 | 0.669 | 20.384 | 20.638 | 0.690 |
| **TIMEGUARD** | **18.176** | **26.268** | **0.775** | 19.464 | **13.938** | 0.768 | **21.974** | **35.797** | **0.700** | **19.871** | **25.335** | **0.748** |

**Generalization to different attack patterns.** We further evaluate the robustness of TIMEGUARD under three attack patterns from the original BackTime work (Lin et al., 2024) (described in Appendix F.5) on the PEMS03 dataset. As shown in Table 22, TIMEGUARD consistently demonstrates strong robustness across all attack patterns, including Random, Manhattan, and BackTime attacks, averaged over the three models. Compared to the state-of-the-art defense PDB, TIMEGUARD significantly improves both MAE$_P$ and FDER, while maintaining clean performance with minimal degradation. The results for the up-and-down and up-trend attack patterns, broken down by model, are provided in Tables 23 and 24, respectively.

*Table 22.* Defense performance of TIMEGUARD across three different attack patterns under Random, Manhattan, and BackTime attacks on PEMS03, *average* over FEDFormer, SimpleTM, and TimesNet models. Best results are in **bold**.

| Attack Pattern | Attack → Defense ↓ | Random | | | Manhattan | | | BackTime | | |
|---|---|---|---|---|---|---|---|---|---|---|
| | | MAE$_C$ ↓ | MAE$_P$ ↑ | FDER ↑ | MAE$_C$ ↓ | MAE$_P$ ↑ | FDER ↑ | MAE$_C$ ↓ | MAE$_P$ ↑ | FDER ↑ |
| **Cone** | No Defense | 17.634 | 17.772 | – | 17.722 | 20.266 | – | 17.607 | 14.201 | – |
| | Fine-tuning | 19.003 | 30.909 | 0.625 | 19.661 | 30.995 | 0.608 | 18.934 | 18.196 | 0.594 |
| | Fine-pruning | 19.020 | 31.643 | 0.633 | 19.595 | 34.447 | 0.624 | 18.686 | 19.736 | 0.623 |
| | PDB | 18.630 | 54.690 | 0.693 | 19.308 | 60.477 | 0.708 | 18.967 | 22.397 | 0.639 |
| | **TIMEGUARD** | **17.928** | **104.677** | **0.868** | **17.850** | **97.370** | **0.854** | **18.048** | **39.303** | **0.808** |
| **Up & Down** | No Defense | 17.628 | 17.903 | – | 17.679 | 20.213 | – | 17.389 | 16.853 | – |
| | Fine-tuning | 19.161 | 26.878 | 0.609 | 19.196 | 28.158 | 0.588 | 18.772 | 19.479 | 0.576 |
| | Fine-pruning | 19.078 | 29.024 | 0.629 | 19.193 | 30.267 | 0.606 | 18.743 | 20.980 | 0.593 |
| | PDB | 19.150 | 45.243 | 0.669 | 19.146 | 42.813 | 0.649 | 20.028 | 21.720 | 0.567 |
| | **TIMEGUARD** | **18.055** | **80.871** | **0.835** | **17.985** | **75.454** | **0.810** | **18.158** | **30.165** | **0.709** |
| **Up Trend** | No Defense | 17.721 | 18.472 | – | 17.733 | 21.010 | – | 17.615 | 13.674 | – |
| | Fine-tuning | 19.223 | 32.865 | 0.636 | 19.540 | 33.298 | 0.624 | 18.935 | 19.706 | 0.629 |
| | Fine-pruning | 19.211 | 33.532 | 0.640 | 19.620 | 34.720 | 0.622 | 19.028 | 21.771 | 0.648 |
| | PDB | 19.227 | 58.857 | 0.683 | 19.201 | 70.032 | 0.687 | 19.783 | 22.195 | 0.645 |
| | **TIMEGUARD** | **18.022** | **111.006** | **0.872** | **17.985** | **101.135** | **0.856** | **18.273** | **46.106** | **0.834** |

**Generalization to different forecasting horizons.** Beyond the default forecasting horizon used in BackTime (Lin et al., 2024) ($L_{out}$=12), we further evaluate TIMEGUARD under longer horizons with $L_{out} \in \{24, 36, 48\}$. Following BackTime's protocol, we assume the attacker knows the forecasting horizon used by the victim model.

As shown in Figures 11–13, TIMEGUARD maintains competitive clean performance (MAE$_C$) across all horizons, and even outperforms undefended training on TimesNet in some cases. As $L_{out}$ increases, BackTime itself becomes less effective (e.g., poisoned MAE$_P$ exceeds 30 across models when $L_{out}$=48), which correspondingly lowers FDER; nevertheless, TIMEGUARD remains robust, with defended MAE$_P$ staying above 28.4 in all settings. We observe that, except for TimesNet, the defense effectiveness of TIMEGUARD lightly decreases for FEDformer and SimpleTM at longer horizons ($L_{out} \in 36, 48$). A plausible explanation is that losses over longer target windows become more diluted across distant time steps, reducing the discriminability used by DRLS (Eq. 10). A potential remedy is to adopt a weighted loss that prioritizes nearer horizons, which we leave for future work. Overall, these results indicate that TIMEGUARD remains effective against TSF backdoor attacks under different forecasting horizons.

*Table 23.* Defense performance of TIMEGUARD under Random, Manhattan, and BackTime attacks with a *up-and-down attack pattern* on the PEMS03 dataset, where FEDFormer, SimpleTM, and TimesNet are the victim models. Best results are in **bold**.

| Model | Attack → Defense ↓ | Random | | | Manhattan | | | BackTime | | |
|---|---|---|---|---|---|---|---|---|---|---|
| | | MAE$_C$ ↓ | MAE$_P$ ↑ | FDER ↑ | MAE$_C$ ↓ | MAE$_P$ ↑ | FDER ↑ | MAE$_C$ ↓ | MAE$_P$ ↑ | FDER ↑ |
| **FEDformer** | No Defense | 16.370 | 15.193 | – | 16.413 | 18.700 | – | 15.943 | 10.461 | – |
| | Fine-tuning | 16.802 | 35.086 | 0.771 | 16.998 | 35.427 | 0.719 | 16.559 | 18.153 | 0.693 |
| | Fine-pruning | 16.772 | 38.811 | 0.792 | 16.882 | 39.564 | 0.750 | 16.650 | 23.142 | 0.753 |
| | PDB | 16.898 | 21.148 | 0.625 | 17.259 | 24.828 | 0.599 | 17.273 | 16.246 | 0.640 |
| | **TIMEGUARD** | **16.551** | **67.162** | **0.881** | **16.564** | **65.542** | **0.853** | **16.776** | **27.529** | **0.785** |
| **SimpleTM** | No Defense | 17.428 | 19.133 | – | 17.461 | 20.735 | – | 17.209 | 12.936 | – |
| | Fine-tuning | 17.534 | 24.886 | 0.613 | 17.515 | 27.539 | 0.622 | 17.247 | 16.747 | 0.613 |
| | Fine-pruning | 17.468 | 27.910 | 0.656 | 17.568 | 29.781 | 0.649 | 17.264 | 16.289 | 0.601 |
| | PDB | 17.555 | 91.863 | 0.892 | 17.610 | 79.668 | 0.866 | 18.996 | 22.338 | 0.663 |
| | **TIMEGUARD** | **17.335** | **140.540** | **0.932** | **17.024** | **126.706** | **0.918** | **17.188** | **28.395** | **0.772** |
| **TimesNet** | No Defense | 19.087 | 19.383 | – | 19.161 | 21.202 | – | 19.016 | 27.161 | – |
| | Fine-tuning | 23.149 | 20.661 | 0.443 | 23.073 | 21.508 | 0.422 | 22.508 | 23.535 | 0.422 |
| | Fine-pruning | 22.994 | 20.352 | 0.439 | 23.131 | 21.456 | 0.420 | 22.315 | 23.509 | 0.426 |
| | PDB | 22.996 | 22.720 | 0.488 | 22.569 | 23.943 | 0.482 | 23.816 | 26.576 | 0.399 |
| | **TIMEGUARD** | **20.278** | **34.911** | **0.693** | **20.368** | **34.114** | **0.660** | **20.508** | **34.571** | **0.571** |

*Table 24.* Defense performance of TIMEGUARD under Random, Manhattan, and BackTime attacks with *up-trend attack pattern* on PEMS03 dataset, where FEDFormer, SimpleTM, and TimesNet are the victim models. Best results are in **bold**.

| Model | Attack → | Random | | | Manhattan | | | BackTime | | |
|---|---|---|---|---|---|---|---|---|---|---|
| | Defense ↓ | MAE$_C$ ↓ | MAE$_P$ ↑ | FDER ↑ | MAE$_C$ ↓ | MAE$_P$ ↑ | FDER ↑ | MAE$_C$ ↓ | MAE$_P$ ↑ | FDER ↑ |
| **FEDformer** | No Defense | 16.420 | 15.652 | – | 16.434 | 18.461 | – | 16.105 | 10.772 | – |
| | Fine-tuning | 16.863 | 50.608 | 0.832 | 16.864 | 42.857 | 0.772 | 16.594 | 22.653 | 0.748 |
| | Fine-pruning | 16.792 | 51.944 | 0.838 | 16.811 | 49.437 | 0.802 | 16.661 | 28.115 | 0.792 |
| | PDB | 16.875 | 23.665 | 0.656 | 17.156 | 29.193 | 0.663 | 17.347 | 16.996 | 0.647 |
| | **TIMEGUARD** | **16.610** | **102.645** | **0.918** | **16.598** | **98.314** | **0.901** | **16.855** | **50.734** | **0.872** |
| **SimpleTM** | No Defense | 17.613 | 20.562 | – | 17.441 | 24.170 | – | 17.302 | 7.999 | – |
| | Fine-tuning | 17.683 | 26.638 | 0.612 | 17.659 | 33.546 | 0.634 | 17.350 | 12.608 | 0.681 |
| | Fine-pruning | 17.661 | 27.625 | 0.626 | 17.717 | 31.995 | 0.615 | 17.483 | 13.888 | 0.707 |
| | PDB | 17.866 | 130.997 | 0.914 | 17.667 | 157.867 | 0.917 | 18.218 | 23.245 | 0.803 |
| | **TIMEGUARD** | **17.066** | **185.689** | **0.945** | **16.951** | **160.888** | **0.925** | **17.263** | **41.806** | **0.904** |
| **TimesNet** | No Defense | 19.129 | 19.203 | – | 19.325 | 20.399 | – | 19.439 | 22.250 | – |
| | Fine-tuning | 23.121 | 21.347 | 0.464 | 24.097 | 23.492 | 0.467 | 22.862 | 23.857 | 0.459 |
| | Fine-pruning | 23.179 | 21.028 | 0.456 | 24.332 | 22.729 | 0.448 | 22.940 | 23.310 | 0.446 |
| | PDB | 22.939 | 21.908 | 0.479 | 22.780 | 23.035 | 0.481 | 23.785 | 26.345 | 0.486 |
| | **TIMEGUARD** | **20.389** | **44.684** | **0.754** | **20.404** | **44.203** | **0.743** | **20.703** | **45.776** | **0.726** |

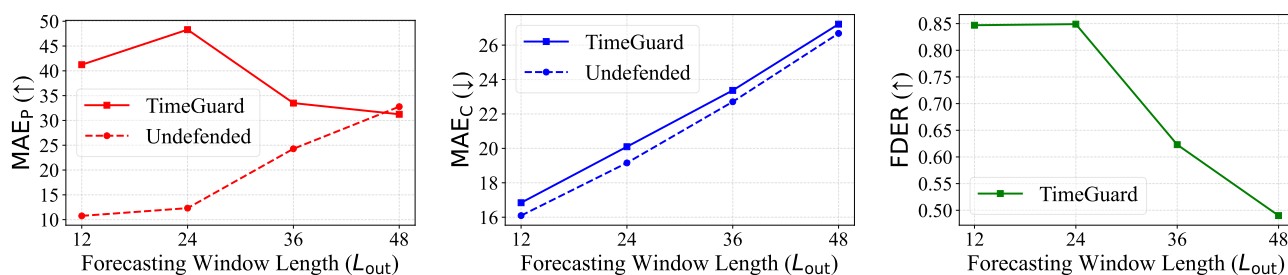

*Figure 11.* Defense performance of TIMEGUARD (MAE$_P$, MAE$_C$, and FDER) under different forecasting window length $L_{out}$ of the BackTime attack on the PEMS03 dataset with the *FEDformer* model.

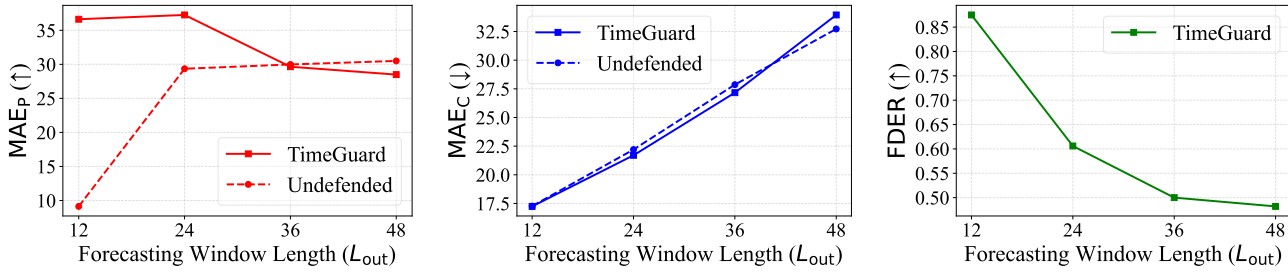

*Figure 12.* Defense performance of TIMEGUARD (MAE$_P$, MAE$_C$, and FDER) under different forecasting window length $L_{out}$ of the BackTime attack on the PEMS03 dataset with the *SimpleTM* model.

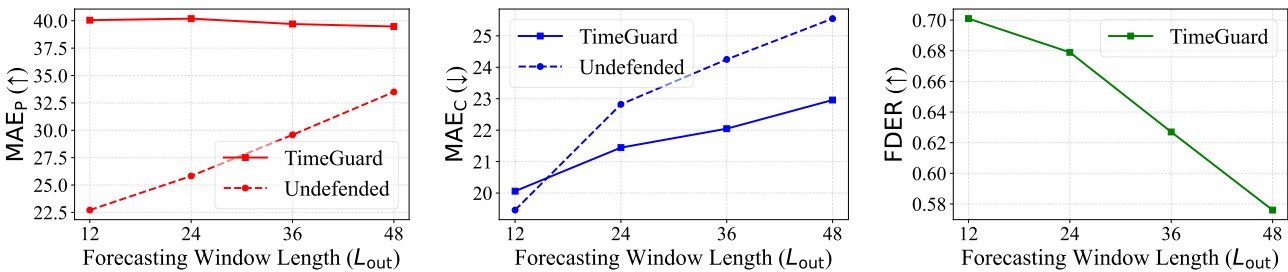

*Figure 13.* Defense performance of TIMEGUARD (MAE$_P$, MAE$_C$, and FDER) under different forecasting window length $L_{out}$ of the BackTime attack on the PEMS03 dataset with the *TimesNet* model.

**Generalization to large-scale datasets.** To evaluate the scalability of TIMEGUARD, we further do experiments on GBA (Liu et al., 2023b), using its 2019 subset, which is a much larger traffic forecasting benchmark ($35040 \times 2352$) than the datasets used in our main experiments. Even at this scale, TIMEGUARD achieves the best overall defense performance, with an average FDER of $0.698$. We also explicitly report the increased training cost, showing that TIMEGUARD remains effective on substantially larger datasets, albeit with higher training overhead. This training time is notably higher than that on our previous largest benchmark, PEMS03, i.e., 3372s as reported in Table 6. The overhead corresponds to $\approx 3.53\times$ the cost of undefended training, indicating that TIMEGUARD remains scalable in practice but incurs nontrivial additional cost.

*Table 25.* Defense performance and training time (in seconds) of PDB and TIMEGUARD under BackTime attack on the GBA dataset (Liu et al., 2023b), where FEDformer, SimpleTM, and TimesNet are used as victim models. Best results are shown in **bold**.

| Model → | FEDformer | | | | SimpleTM | | | | TimesNet | | | | AVERAGE | | | |
|---|---|---|---|---|---|---|---|---|---|---|---|---|---|---|---|---|
| Defense ↓ | MAE$_C$ ↓ | MAE$_P$ ↑ | FDER ↑ | Training Time ↓ | MAE$_C$ ↓ | MAE$_P$ ↑ | FDER ↑ | Training Time ↓ | MAE$_C$ ↓ | MAE$_P$ ↑ | FDER ↑ | Training Time ↓ | MAE$_C$ ↓ | MAE$_P$ ↑ | FDER ↑ | Training Time ↓ |
| No Defense | 27.259 | 26.234 | – | 5625.6 | 32.814 | 37.735 | – | 6169.1 | 31.846 | 41.139 | – | 7134.3 | 30.640 | 35.036 | – | 6309.7 |
| PDB (Wei et al., 2024) | **26.705** | 39.016 | 0.664 | 5625.3 | **32.406** | 48.588 | **0.612** | 6419.8 | 32.326 | 51.237 | 0.591 | 8541.8 | **30.479** | 46.280 | 0.622 | **6862.3** |
| TIMEGUARD | 28.463 | **69.200** | **0.789** | 23390.1 | 39.874 | **54.641** | 0.566 | 22641.2 | **31.647** | **79.109** | **0.740** | 20745.8 | 33.328 | **67.650** | **0.698** | 22259.0 |

**Generalization to discrete, count-valued datasets.** While our main evaluation focuses on continuous-valued datasets, we further assess TIMEGUARD on a discrete, count-valued dataset. Specifically, we use the hourly subset of Bike Sharing (Fanaee-T, 2013), which records hourly bike rental counts from 2011 to 2012 in the Capital Bikeshare system, and evaluate under the Random attack. For preprocessing, we retain "temp", "atemp", "hum", "windspeed", and "cnt", and use "cnt", a discrete variable representing the number of rental bikes, as the target variable, resulting in 17,379 time stamps. As shown in Table 26, TIMEGUARD still achieves the best defense performance, with an average FDER of $0.831$. This suggests preliminary transfer beyond continuous-valued TSF, while broader adaptation to discrete and count-valued forecasting remains future work.

*Table 26.* Defense performance of PDB and TIMEGUARD under Random attack on the Bike Sharing dataset (Fanaee-T, 2013), where FEDformer, SimpleTM, and TimesNet are used as victim models. Best results are shown in **bold**.

| Model → | FEDformer | | | SimpleTM | | | TimesNet | | | AVERAGE | | |
|---|---|---|---|---|---|---|---|---|---|---|---|---|
| Defense ↓ | MAE$_C$ ↓ | MAE$_P$ ↑ | FDER ↑ | MAE$_C$ ↓ | MAE$_P$ ↑ | FDER ↑ | MAE$_C$ ↓ | MAE$_P$ ↑ | FDER ↑ | MAE$_C$ ↓ | MAE$_P$ ↑ | FDER ↑ |
| No Defense | 23.921 | 66.894 | – | 24.143 | 66.465 | – | 20.377 | 47.904 | – | 22.814 | 60.421 | – |
| PDB (Wei et al., 2024) | **22.368** | 120.193 | 0.722 | 24.472 | 124.615 | 0.727 | **19.131** | 103.550 | 0.769 | **21.990** | 116.119 | 0.739 |
| TIMEGUARD | 28.355 | **249.084** | **0.788** | **20.699** | **243.449** | **0.863** | 22.751 | **227.498** | **0.843** | 23.935 | **240.010** | **0.831** |

**Robustness on distribution-shifted and nonstationary datasets.** Since TIMEGUARD relies on a hand-designed neighborhood metric, its estimates may degrade under strong distribution shift and nonstationarity. To evaluate this scenario, we test TIMEGUARD on Exchange, a financial forecasting benchmark with evolving dynamics that contains daily exchange rates from 8 countries between 1990 and 2016 (Lai et al., 2018), totaling 7,588 time steps, under the BackTime attack. The results in Table 27 show that, despite these challenging evolving dynamics, TIMEGUARD remains effective and outperforms PDB. This suggests that TIMEGUARD remains practically effective even under stronger distribution shift and nonstationarity.

*Table 27.* Defense performance of PDB and TIMEGUARD under BackTime attack on the Exchange dataset (Lai et al., 2018), where FEDformer, SimpleTM, and TimesNet are used as victim models. Best results are shown in **bold**.

| Model → | FEDformer | | | SimpleTM | | | TimesNet | | | AVERAGE | | |
|---|---|---|---|---|---|---|---|---|---|---|---|---|
| Defense ↓ | MAE$_C$ ↓ | MAE$_P$ ↑ | FDER ↑ | MAE$_C$ ↓ | MAE$_P$ ↑ | FDER ↑ | MAE$_C$ ↓ | MAE$_P$ ↑ | FDER ↑ | MAE$_C$ ↓ | MAE$_P$ ↑ | FDER ↑ |
| No Defense | 0.00967 | 0.02143 | – | 0.00699 | 0.01927 | – | 0.03089 | 0.10875 | – | 0.01585 | 0.04982 | – |
| PDB (Wei et al., 2024) | 0.01654 | 0.08107 | 0.66040 | 0.00737 | 0.10169 | 0.87944 | 0.05331 | **0.16554** | 0.46127 | 0.02574 | 0.11610 | 0.66704 |
| TIMEGUARD | **0.00803** | **0.11179** | **0.90417** | **0.00673** | **0.10451** | **0.90782** | **0.03829** | 0.14101 | **0.51782** | **0.01768** | **0.11911** | **0.77660** |

**Robustness under nonstationary settings and concept drift.** To further examine robustness under mild distribution change and concept drift, we conduct an additional experiment on PEMS03 under the BackTime attack by introducing synthetic distribution shifts at test time. Specifically, let $x_{t,c}$ denote the value at time step $t$ and channel $c$, and let $\sigma_c$ denote the standard deviation of channel $c$ computed from the training dataset. We consider three perturbations:

- Scale shift: $x'_{t,c} = (1 + \alpha)x_{t,c}$

- Mean shift: $x'_{t,c} = x_{t,c} + \alpha\sigma_c$

- Linear trend: $x'_{t,c} = x_{t,c} + \alpha\sigma_c \frac{t}{T}$, where $T$ is the length of test split.

We test two shift strengths, $\alpha \in \{0.1, 0.2\}$. As shown in Table 28, the performance of all methods declines slightly under synthetic distribution shift. However, TIMEGUARD consistently achieves the best defense performance across all six shifted

settings. This suggests that although non-stationarity affects performance, its negative impact is moderate rather than catastrophic, and TIMEGUARD remains stable in practice under mild distribution shifts. Under strongly non-stationary or concept-drift scenarios, any training-phase defense is likely to face challenges, and TIMEGUARD is no exception, as also reflected by the degraded performance of both undefended training and PDB.

*Table 28.* Defense performance of TIMEGUARD and PDB under BackTime attack on PEMS03 dataset under mild distribution shift, where FEDFormer, SimpleTM, and TimesNet are the victim models. Best results in each scenario are shown in **bold**.

| Shift / Strength | Model → Defense ↓ | FEDformer | | | SimpleTM | | | TimesNet | | | AVERAGE | | |
|---|---|---|---|---|---|---|---|---|---|---|---|---|---|
| | | MAE$_C$ ↓ | MAE$_P$ ↑ | FDER ↑ | MAE$_C$ ↓ | MAE$_P$ ↑ | FDER ↑ | MAE$_C$ ↓ | MAE$_P$ ↑ | FDER ↑ | MAE$_C$ ↓ | MAE$_P$ ↑ | FDER ↑ |
| No shift | No Defense | 16.688 | 13.577 | – | 16.519 | 8.218 | – | 22.041 | 21.293 | – | 18.416 | 14.363 | – |
| | PDB (Wei et al., 2024) | 17.420 | 16.038 | 0.556 | 18.283 | 25.700 | 0.792 | 23.586 | 26.898 | 0.571 | 19.763 | 22.879 | 0.640 |
| | TIMEGUARD | 16.850 | 42.101 | 0.834 | 17.688 | 36.386 | 0.854 | 20.562 | 40.005 | 0.734 | 18.367 | 39.497 | 0.807 |
| Scale shift ($\alpha = 0.1$) | No Defense | 18.323 | 14.860 | – | 18.171 | 8.696 | – | 25.510 | 24.590 | – | 20.668 | 16.049 | – |
| | PDB (Wei et al., 2024) | 19.039 | 17.560 | 0.558 | 20.229 | 26.254 | 0.784 | 27.108 | 30.703 | 0.570 | 22.125 | 24.839 | 0.637 |
| | TIMEGUARD | 18.508 | 42.423 | 0.820 | 19.457 | 36.707 | 0.849 | 23.672 | 40.808 | 0.699 | 20.546 | 39.979 | 0.789 |
| Scale shift ($\alpha = 0.2$) | No Defense | 19.977 | 16.170 | – | 19.823 | 9.241 | – | 30.968 | 29.982 | – | 23.590 | 19.682 | – |
| | PDB (Wei et al., 2024) | 20.734 | 19.108 | 0.482 | 22.384 | 27.480 | 0.775 | 32.602 | 36.076 | 0.559 | 25.240 | 27.555 | 0.605 |
| | TIMEGUARD | 20.225 | 42.811 | 0.762 | 21.226 | 37.050 | 0.842 | 28.465 | 42.762 | 0.649 | 23.305 | 40.874 | 0.751 |
| Linear trend ($\alpha = 0.1$) | No Defense | 16.689 | 14.860 | – | 16.519 | 8.696 | – | 21.761 | 24.590 | – | 18.323 | 16.049 | – |
| | PDB (Wei et al., 2024) | 17.446 | 16.029 | 0.515 | 18.409 | 25.862 | 0.781 | 23.117 | 28.007 | 0.532 | 19.657 | 23.300 | 0.609 |
| | TIMEGUARD | 16.841 | 42.123 | 0.819 | 17.688 | 36.386 | 0.847 | 20.523 | 41.380 | 0.703 | 18.351 | 39.963 | 0.790 |
| Linear trend ($\alpha = 0.2$) | No Defense | 16.690 | 13.515 | – | 16.518 | 8.218 | – | 22.140 | 29.982 | – | 18.450 | 17.239 | – |
| | PDB (Wei et al., 2024) | 17.482 | 16.028 | 0.556 | 18.641 | 26.046 | 0.785 | 23.364 | 29.682 | 0.474 | 19.829 | 23.919 | 0.605 |
| | TIMEGUARD | 16.836 | 42.139 | 0.835 | 17.688 | 36.401 | 0.854 | 20.990 | 42.854 | 0.650 | 18.505 | 40.465 | 0.780 |
| Mean shift ($\alpha = 0.1$) | No Defense | 16.688 | 13.515 | – | 16.519 | 8.217 | – | 22.051 | 23.388 | – | 18.419 | 15.040 | – |
| | PDB (Wei et al., 2024) | 17.474 | 16.016 | 0.556 | 18.607 | 26.066 | 0.786 | 23.242 | 29.628 | 0.580 | 19.774 | 23.903 | 0.641 |
| | TIMEGUARD | 16.838 | 42.166 | 0.835 | 17.688 | 36.422 | 0.854 | 20.901 | 42.946 | 0.728 | 18.476 | 40.512 | 0.806 |
| Mean shift ($\alpha = 0.2$) | No Defense | 16.693 | 13.462 | – | 16.519 | 8.217 | – | 23.937 | 26.798 | – | 19.050 | 16.159 | – |
| | PDB (Wei et al., 2024) | 17.559 | 16.028 | 0.555 | 19.207 | 26.066 | 0.772 | 24.941 | 33.902 | 0.585 | 20.569 | 25.332 | 0.637 |
| | TIMEGUARD | 16.836 | 42.227 | 0.836 | 17.688 | 36.422 | 0.854 | 22.566 | 46.094 | 0.709 | 19.030 | 41.581 | 0.800 |

## G.2. Ablation Study Full Results

We provide per-model ablation results on the PEMS03 dataset under the Random, Manhattan, and BackTime attacks in Table 29 with FEDformer, SimpleTM, and TimesNet. Overall, these results are consistent with the model-averaged trends reported in Section 5.2.

## G.3. Hyperparameter Sensitivity Full Results

**Influence of $\alpha$ and $\beta$.** Figures 14–16 report the TIMEGUARD defense performance with FEDformer, SimpleTM, and TimesNet on PEMS03 under the BackTime attack while varying $\alpha \in \{0.10, 0.15, 0.20, 0.25, 0.30\}$ and $\beta \in \{0.40, 0.50, 0.60, 0.70, 0.80\}$ combination. Consistent with the model-averaged trends in Section 5.2, it recommends choosing $\alpha \in [0.15, 0.25]$ and $\beta \in [0.5, 0.7]$ to balance clean performance and robustness.

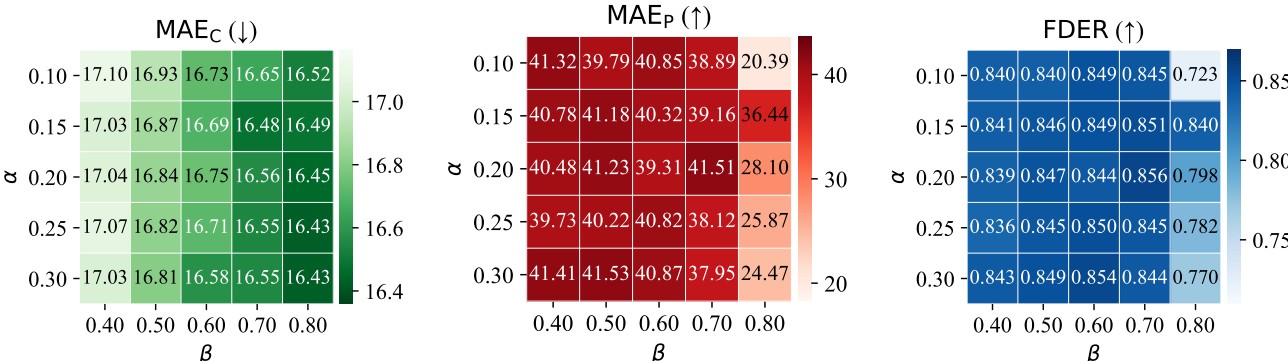

*Figure 14.* Defense performance of TIMEGUARD (MAE$_P$, MAE$_C$, and FDER) with different initial reliable-pool ratio $\alpha$ and final ratio $\beta$ under **BackTime** attack on the **PEMS03** dataset with the **FEDformer** model.

*Table 29.* Per-model ablation results of TIMEGUARD on PEMS03 under the Random, Manhattan, and BackTime attacks, with FEDformer, SimpleTM, and TimesNet as victim models. The AVERAGE row reports the mean across the three models, matching Table 5.

| Model | Attack → / Defense ↓ | Random | | | Manhattan | | | BackTime | | |
|---|---|---|---|---|---|---|---|---|---|---|
| | | MAE$_C$ ↓ | MAE$_P$ ↑ | FDER ↑ | MAE$_C$ ↓ | MAE$_P$ ↑ | FDER ↑ | MAE$_C$ ↓ | MAE$_P$ ↑ | FDER ↑ |
| **FEDformer** | No Defense | 16.286 | 14.959 | – | 16.411 | 17.984 | – | 16.093 | 10.760 | – |
| | **TIMEGUARD** | 16.607 | **100.436** | **0.916** | 16.578 | **94.212** | **0.900** | 16.840 | **41.232** | **0.847** |
| | w/o Channel-wise | 16.558 | 14.915 | 0.492 | 16.660 | 18.462 | 0.505 | 17.959 | 12.832 | 0.529 |
| | w/o NDF | 16.717 | 91.682 | 0.906 | 16.695 | 87.981 | 0.889 | 16.947 | 39.474 | 0.839 |
| | w/o RCF | 16.622 | 99.747 | 0.915 | 16.639 | 89.769 | 0.893 | 16.962 | 40.913 | 0.843 |
| | w/o NDF+RCF | **16.549** | 83.125 | 0.902 | **16.549** | 82.560 | 0.887 | **16.740** | 40.111 | **0.847** |
| | w/o DRLS | 17.727 | 15.775 | 0.485 | 17.618 | 16.000 | 0.466 | 17.711 | 9.509 | 0.454 |
| **SimpleTM** | No Defense | 17.510 | 19.007 | – | 17.539 | 22.532 | – | 17.268 | 9.131 | – |
| | **TIMEGUARD** | 17.489 | 173.700 | **0.945** | 17.284 | 157.870 | **0.929** | 17.243 | 36.626 | **0.875** |
| | w/o Channel-wise | **16.826** | 14.307 | 0.500 | 16.660 | 18.462 | 0.500 | 17.666 | 6.615 | 0.489 |
| | w/o NDF | 18.740 | 180.126 | 0.914 | 16.695 | 87.981 | 0.872 | 17.400 | 36.145 | 0.870 |
| | w/o RCF | 17.311 | 172.719 | **0.945** | **16.639** | 89.769 | 0.874 | 17.703 | **37.717** | 0.867 |
| | w/o NDF+RCF | 17.577 | 151.760 | 0.935 | 17.793 | 139.796 | 0.912 | **17.094** | 35.681 | 0.872 |
| | w/o DRLS | 19.970 | **194.031** | 0.889 | 19.945 | **173.531** | 0.875 | 20.628 | 31.919 | 0.776 |
| **TimesNet** | No Defense | 19.104 | 19.351 | – | 19.216 | 20.283 | – | 19.459 | 22.713 | – |
| | **TIMEGUARD** | **19.687** | 39.894 | **0.743** | **19.689** | **40.029** | **0.735** | **20.061** | 40.052 | **0.701** |
| | w/o Channel-wise | 21.577 | 19.212 | 0.443 | 21.389 | 20.662 | 0.458 | 21.580 | 25.328 | 0.502 |
| | w/o NDF | 20.286 | **41.563** | 0.738 | 20.370 | 39.967 | 0.718 | 20.908 | 39.428 | 0.677 |
| | w/o RCF | 20.255 | 40.748 | 0.734 | 20.043 | 39.957 | 0.726 | 21.158 | **40.207** | 0.677 |
| | w/o NDF+RCF | 20.880 | 40.456 | 0.718 | 20.366 | 39.382 | 0.714 | 20.985 | 39.889 | 0.679 |
| | w/o DRLS | 21.545 | 19.520 | 0.448 | 21.723 | 21.164 | 0.463 | 21.905 | 27.327 | 0.529 |
| **AVERAGE** | No Defense | 17.634 | 17.772 | – | 17.722 | 20.266 | – | 17.607 | 14.201 | – |
| | **TIMEGUARD** | 17.928 | **104.677** | **0.868** | 17.850 | **97.370** | **0.854** | 18.048 | 39.303 | **0.808** |
| | w/o Channel-wise | 18.320 | 16.145 | 0.478 | 18.236 | 19.195 | 0.488 | 19.068 | 14.925 | 0.507 |
| | w/o NDF | 18.581 | 104.457 | 0.853 | 17.920 | 71.976 | 0.826 | 18.418 | 38.349 | 0.795 |
| | w/o RCF | 18.063 | 104.405 | 0.865 | **17.774** | 73.165 | 0.831 | 18.608 | **39.612** | 0.796 |
| | w/o NDF+RCF | 18.336 | 91.780 | 0.852 | 18.236 | 87.246 | 0.838 | 18.273 | 38.560 | 0.799 |
| | w/o DRLS | 19.748 | 76.442 | 0.607 | 19.762 | 70.232 | 0.601 | 20.081 | 22.918 | 0.586 |

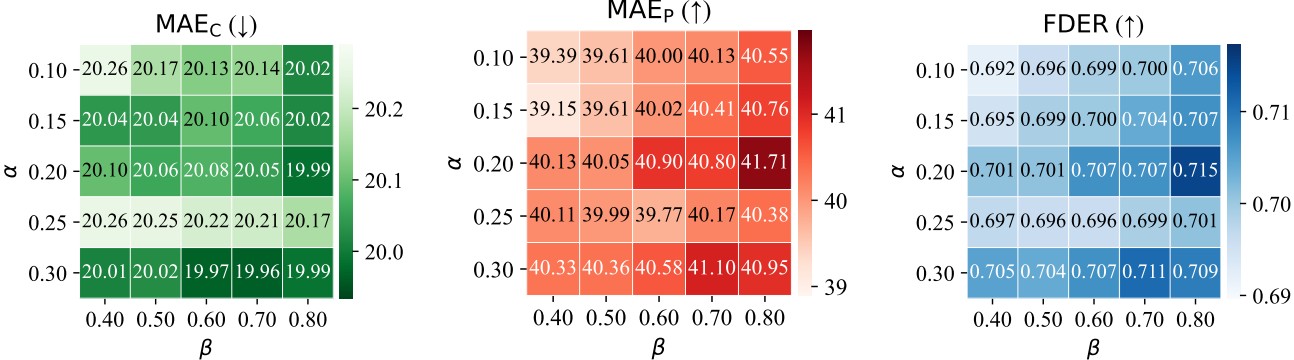

*Figure 15.* Defense performance of TIMEGUARD (MAE$_P$, MAE$_C$, and FDER) with different initial reliable-pool ratio $\alpha$ and final ratio $\beta$ under **BackTime** attack on the **PEMS03** dataset with the **SimpleTM** model.

*Figure 16.* Defense performance of TIMEGUARD (MAE$_P$, MAE$_C$, and FDER) with different initial reliable-pool ratio $\alpha$ and final ratio $\beta$ under **BackTime** attack on the **PEMS03** dataset with the **TimesNet** model.

We further evaluate the effects of pool sizes $\alpha$ and $\beta$ on the Weather dataset under the BackTime attack, as shown in Figure 17 and Figure 18. Specifically, we vary $\alpha \in \{0.10, 0.15, 0.20, 0.25, 0.30\}$ with $\beta$ fixed at 0.50, and vary $\beta \in \{0.40, 0.50, 0.60, 0.70, 0.80\}$ with $\alpha$ fixed at 0.20. Overall, TIMEGUARD remains effective across all settings, achieving FDER above 0.75 in every case. These results lead to the same conclusion as in Section 5.1 on PEMS03 dataset: $\alpha \in [0.15, 0.25]$ and $\beta \in [0.5, 0.7]$ provide the best trade-off, as reflected by FDER.

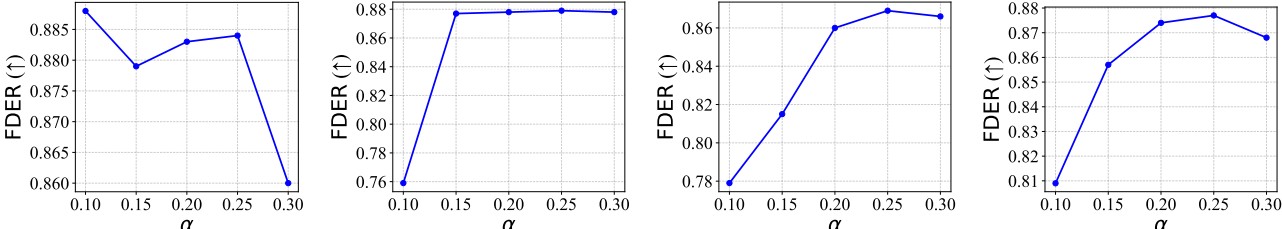

*Figure 17.* Defense performance of TIMEGUARD in terms of FDER with different initialization ratios $\alpha$ under the **BackTime** attack on the **Weather** dataset, reported for FEDformer, SimpleTM, TimesNet, and their average.

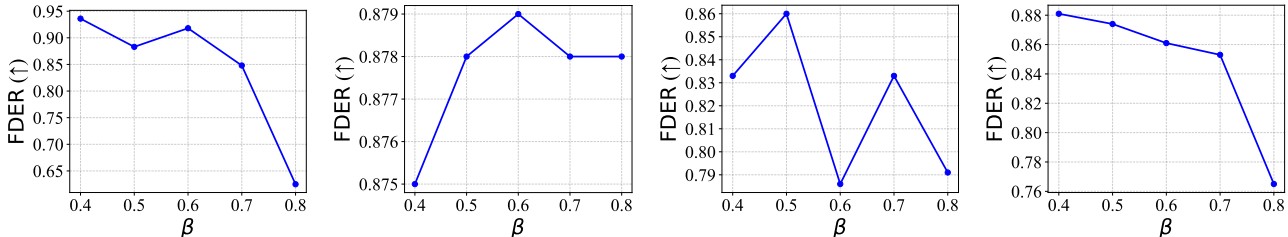

*Figure 18.* Defense performance of TIMEGUARD in terms of FDER with different maximum pool ratios $\beta$ under the **BackTime** attack on the **Weather** dataset, reported for FEDformer, SimpleTM, TimesNet, and their average.

**Influence of $K$ and $\pi$.** Figures 19 and 20 report the TIMEGUARD defense performance with FEDformer, SimpleTM, and TimesNet on PEMS03 under the BackTime attack while varying the neighborhood size $K \in \{10, 20, 32, 48, 64\}$ and the scaling factor $\pi \in \{1.05, 1.15, 1.25, 1.35, 1.50, 1.65\}$, respectively. Consistent with the model-averaged trends in Section 5.2, TIMEGUARD is relatively insensitive to the choice of $K$, while we recommend selecting $\pi \le 1.5$.

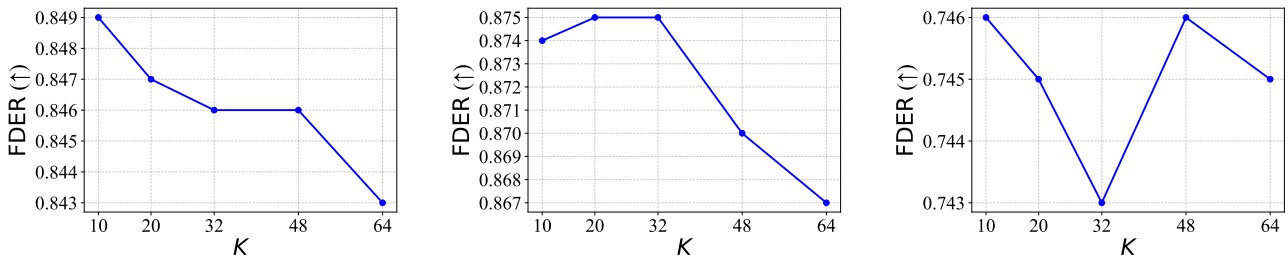

*Figure 19.* Defense performance of TIMEGUARD (FDER) with different neighborhood size $K$ under **BackTime** attack on the **PEMS03** dataset with the FEDformer, SimpleTM, and TimesNet, respectively.

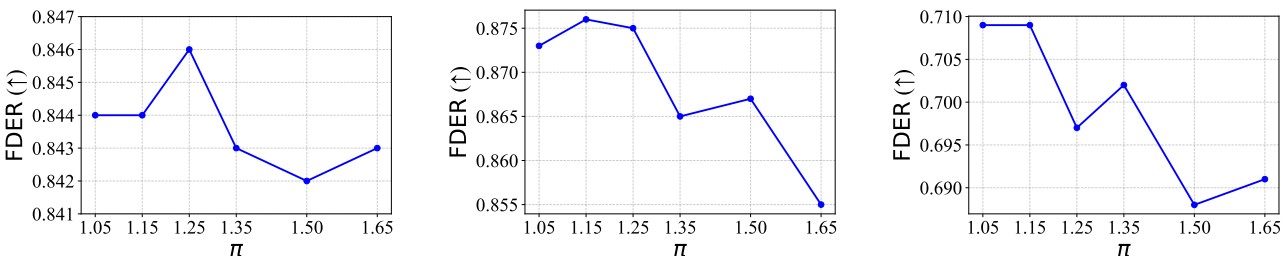

*Figure 20.* Defense performance of TIMEGUARD (FDER) with different scaling factor $\pi$ under **BackTime** attack on the **PEMS03** dataset with the FEDformer, SimpleTM, and TimesNet, respectively.

We also vary $K$ and $\pi$ under the BackTime attack on the Weather dataset, as shown in Figure 21–22, and under the Random attack on the PEMS03 dataset, as shown in Figure 23–24. Consistent with the results on PEMS03 under BackTime in Section 5.1, TIMEGUARD is relatively insensitive to the choice of $K$, and we recommend selecting $\pi \leq 1.5$. These results also suggest that the effects of $K$ and $\pi$ in TIMEGUARD are consistent across different datasets and attack scenarios.

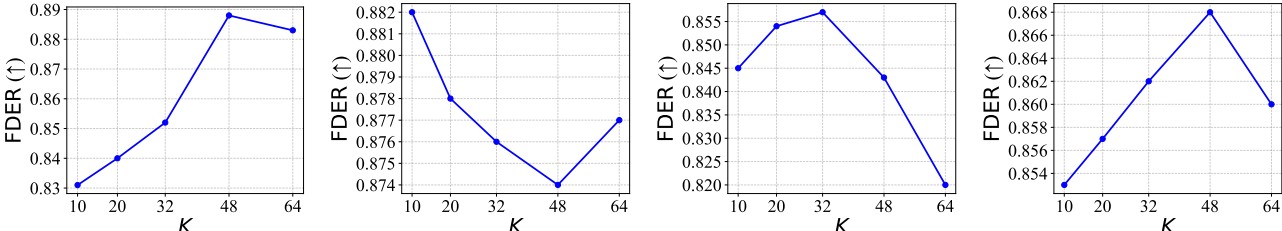

*Figure 21.* Defense performance of TIMEGUARD in terms of FDER with different neighborhood sizes $K$ under the **BackTime** attack on the **Weather** dataset, reported for FEDformer, SimpleTM, TimesNet, and their average.

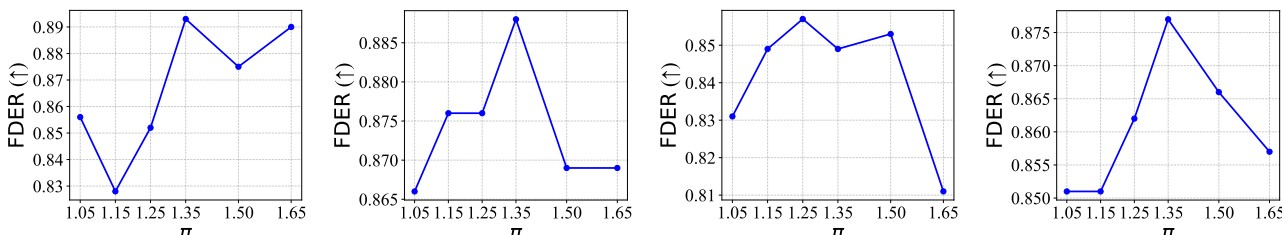

*Figure 22.* Defense performance of TIMEGUARD in terms of FDER with different scaling factors $\pi$ under the **BackTime** attack on the **Weather** dataset, reported for FEDformer, SimpleTM, TimesNet, and their average.

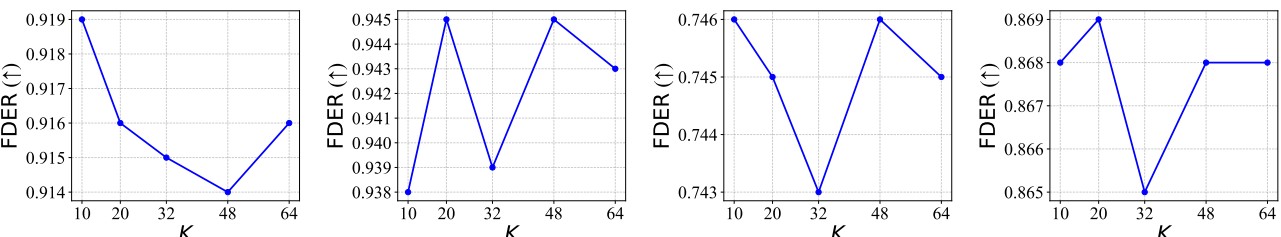

*Figure 23.* Defense performance of TIMEGUARD in terms of FDER with different neighborhood sizes $K$ under the **Random** attack on the **PEMS03** dataset, reported for FEDformer, SimpleTM, TimesNet, and their average.

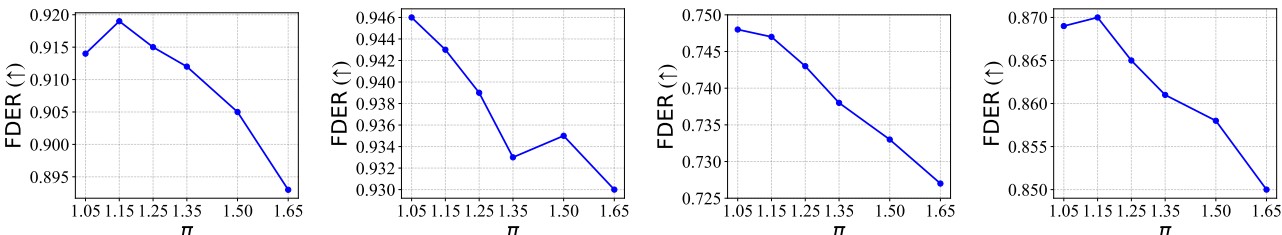

*Figure 24.* Defense performance of TIMEGUARD in terms of FDER with different scaling factors $\pi$ under the **Random** attack on the **PEMS03** dataset, reported for FEDformer, SimpleTM, TimesNet, and their average.

**Influence of $T_b$.** We further study the sensitivity to the number of backcaster training epochs $T_b$ used in the BLS module. Figure 25 reports the defense performance of TIMEGUARD on PEMS03 under the BackTime attack for all three victim models while $T_b \in \{0, 5, 10, 20, 40, 60\}$. When $T_b < 10$, the backcaster is likely under-trained, making its loss signal less reliable for separating clean and poisoned samples, which results in suboptimal defense performance. To balance robustness and training cost, we set $T_b=10$ by default, which adds only roughly 10% overhead to the training pipeline while reducing the risk of overfitting, where the backcaster may start fitting poisoned hard samples.

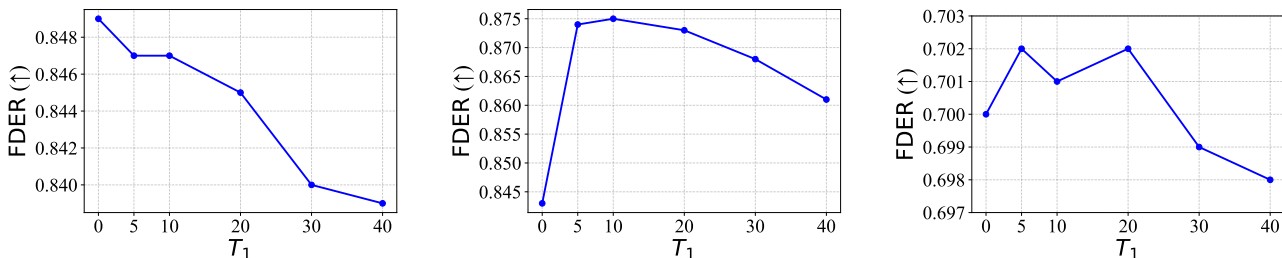

*Figure 25.* Defense performance of TIMEGUARD (FDER) with varying backcaster $b_\phi$ training epoch $T_b$ under BackTime attack on the PEMS03 dataset with the FEDformer, SimpleTM, and TimesNet, respectively.

**Influence of $T_1$ and $T_2$.** We study the sensitivity to the stage-wise training budgets $T_1$ (Stage I) and $T_2$ (Stage II) of TIMEGUARD, while fixing the total budget to $T_1 + T_2 = 100$. Figure 26 reports the defense performance on PEMS03 under the BackTime attack for all three victim models, varying $T_1 \in \{0, 5, 10, 20, 30, 40\}$ with the corresponding $T_2 \in \{100, 95, 90, 80, 70, 60\}$. Overall, performance tends to degrade as $T_1$ increases, suggesting that overly long Stage I training may overfit the initial reliable pool and leave insufficient budget for incorporating newly admitted reliable samples in Stage II. Notably, for SimpleTM and TimesNet, a short Stage I training ($T_1 \leq 10$) improves subsequent progressive training, since Stage II relies on the current model's loss signal for selecting reliable samples. We set $T_1{=}10$ and $T_2{=}90$ by default, which is also consistent with common two-stage training schedules used in backdoor defenses (Li et al., 2021a; Gao et al., 2023a).

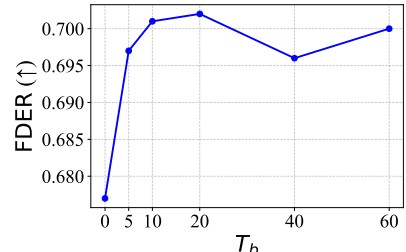

*Figure 26.* Defense performance of TIMEGUARD (FDER) with varying training epoch $T_1$ under BackTime attack on the PEMS03 dataset with the FEDformer, SimpleTM, and TimesNet, respectively.

### G.4. Detailed Efficiency Analysis

As shown in Table 6, TIMEGUARD introduces additional training-time overhead, mainly from Stage I backcaster training and neighborhood-based filtering. However, this cost is limited and temporary: Stage I runs for only $T_b$ epochs, which we set to approximately $10\%$ of the total training budget by default. After Stage I, the backcaster is discarded, so its additional memory overhead does not persist into the main training stage. Importantly, this differs from the repeated inference-time overhead discussed in Table 2, since TIMEGUARD introduces no additional latency during deployment.

To reduce this cost, we implement neighborhood search in a precompute-and-reuse manner instead of recomputing all distances repeatedly. Specifically, we compute the channel-wise neighbor graph only once before the main Stage II training, cache the top-$K_{\max}$ neighbors for each sample, and reuse these cached neighborhoods throughout filtering and selection rather than recomputing kNN every epoch. By default, we set $K_{\max} = 2K$; our preliminary experiments show that TIMEGUARD is insensitive to the choice of $K_{\max}$. This keeps the defense model-agnostic while avoiding repeated full-distance searches during training. As shown in Table 20, TIMEGUARD incurs training time comparable to PDB, the leading baseline, while achieving better defense performance on large-scale time-series foundation models based on LLaMA-7B (Touvron et al., 2023). Similar scalability trends are also observed on large-scale datasets, as shown in Table 25.

### G.5. Potential Adaptive Attacks

We consider a worst-case scenario in which the attacker deliberately adapts the attack strategy to circumvent our defense.

**Design.** We construct an adaptive variant of BackTime (Lin et al., 2024) by augmenting the trigger-generator training objective. To challenge our unidirectional trigger-to-target assumption, we assume the attacker has access to a pre-trained backcaster $b_\phi$ trained on the clean dataset. The attacker then adds a reverse-consistency regularizer $L_{\text{uni}}$ that encourages

*Table 30.* Defense performance of TIMEGUARD under BackTime and adaptive attacks on PEMS03 dataset, where FEDFormer, SimpleTM, and TimesNet are the victim models. Best results under adaptive attack are in **bold**.

| Attack | Model → Defense ↓ | FEDformer | | | SimpleTM | | | TimesNet | | | AVERAGE | | |
|---|---|---|---|---|---|---|---|---|---|---|---|---|---|
| | | $\text{MAE}_C$ ↓ | $\text{MAE}_P$ ↑ | FDER ↑ | $\text{MAE}_C$ ↓ | $\text{MAE}_P$ ↑ | FDER ↑ | $\text{MAE}_C$ ↓ | $\text{MAE}_P$ ↑ | FDER ↑ | $\text{MAE}_C$ ↓ | $\text{MAE}_P$ ↑ | FDER ↑ |
| BackTime | No Defense | 16.093 | 10.760 | – | 17.268 | 9.131 | – | 19.459 | 22.713 | – | 17.607 | 14.201 | – |
| | TIMEGUARD | 16.840 | 41.232 | 0.847 | 17.243 | 36.626 | 0.875 | 20.061 | 40.052 | 0.701 | 18.048 | 39.303 | 0.808 |
| Adaptive | No Defense | 16.383 | 13.779 | – | 17.491 | 10.744 | – | 22.498 | 21.507 | – | 18.791 | 15.343 | – |
| | TIMEGUARD | 17.066 | **30.035** | **0.751** | 17.707 | **22.393** | **0.754** | 20.540 | **39.298** | **0.726** | **18.438** | **30.575** | **0.744** |
| | TIMEGUARD w/o NDF | **17.012** | 29.622 | 0.749 | **17.287** | 22.053 | 0.756 | 21.392 | 37.411 | 0.713 | 18.564 | 29.695 | 0.739 |
| | TIMEGUARD w/o DRLS | 19.035 | 14.558 | 0.457 | 21.093 | 15.808 | 0.575 | 22.460 | 26.712 | 0.597 | 20.863 | 19.026 | 0.543 |

the induced target pattern (specified by the attack target $\mathbf{P}$) to reconstruct the generated trigger $\mathbf{G}$, making the trigger and target more mutually predictive. In addition, to weaken our neighborhood similarity signal based on weighted Pearson correlation, the attacker introduces a similarity regularizer $L_{\text{sim}}$ that maintains a buffer of previously generated poisoned samples and penalizes the distance between the current poisoned sample and the buffer, thereby encouraging diversification. The resulting adaptive objective is:

$$L_{\text{adap}} = L_{bd} + \lambda_1 L_{\text{uni}} + \lambda_2 L_{\text{sim}},$$

where $L_{bd}$ is the original BackTime backdoor objective and $\lambda_1, \lambda_2$ are attacker-controlled hyperparameters. In our implementation, we grid-search $\lambda_1 \in \{0.1, 0.5, 1\}$ and $\lambda_2 \in \{10, 100, 1000\}$.

**Results.** Table 30 shows that, on PEMS03, this adaptive attack attains an average $\text{MAE}_C$ of 18.791 and an average $\text{MAE}_P$ of 15.343, which is slightly worse than the original BackTime attack (17.607 $\text{MAE}_C$ and 14.201 $\text{MAE}_P$, averaged over models). This suggests that enforcing additional constraints, namely reverse consistency and similarity regularization, can hinder the attacker. This observation aligns with our analysis that effective TSF backdoors rely on generating highly similar trigger-induced patterns, as discussed in Theorem 4.1.

Meanwhile, TIMEGUARD remains effective under this adaptive threat, achieving 18.438 $\text{MAE}_C$, 30.575 $\text{MAE}_P$, and 0.744 FDER, which remains within a strong defense-performance range. This trend is consistent across all forecasting models. We attribute this robustness primarily to the distance-aware criteria, which exploit the attacker's structural need to produce highly correlated poisoned samples for successful backdoor activation.

### G.6. Neighborhood Distance Analysis

As TIMEGUARD relies on a hand-designed neighborhood metric, e.g., correlation-/distance-based $k$NN on normalized windows, input-space distances may degrade under strong distribution shifts or nonstationarity. To examine whether learned representations can provide a more robust neighborhood signal, we replace our Gaussian-weighted input-space distance with a TS2Vec embedding distance (Yue et al., 2022) on the Weather dataset under the BackTime attack. For implementation, we train TS2Vec on the full training set with hidden dimension 64, output dimension 128, and depth 6, and then use the resulting sample embeddings to compute neighborhood distances. Since this representation-learning step is computationally expensive, taking approximately 80,000 seconds, we evaluate this variant only on the moderate-scale Weather dataset.

*Table 31.* Defense performance of PDB, TIMEGUARD, and TIMEGUARD$_{\text{emb}}$, a variant of TIMEGUARD that uses TS2Vec embeddings (Yue et al., 2022) as sample representations for neighborhood-distance computation, under the BackTime attack on the Weather dataset. FEDformer, SimpleTM, and TimesNet are used as victim models. Best results are shown in **bold**.

| Model → Defense ↓ | FEDformer | | | SimpleTM | | | TimesNet | | | AVERAGE | | |
|---|---|---|---|---|---|---|---|---|---|---|---|---|
| | $\text{MAE}_C$ ↓ | $\text{MAE}_P$ ↑ | FDER ↑ | $\text{MAE}_C$ ↓ | $\text{MAE}_P$ ↑ | FDER ↑ | $\text{MAE}_C$ ↓ | $\text{MAE}_P$ ↑ | FDER ↑ | $\text{MAE}_C$ ↓ | $\text{MAE}_P$ ↑ | FDER ↑ |
| No Defense | 9.609 | 8.020 | – | 7.752 | 15.301 | – | 14.943 | 24.417 | – | 10.768 | 15.913 | – |
| PDB (Wei et al., 2024) | 10.254 | 35.951 | 0.857 | 8.040 | 50.108 | 0.829 | 16.903 | 83.259 | 0.795 | 11.732 | 56.439 | 0.827 |
| TIMEGUARD | **10.089** | **43.244** | **0.883** | **7.934** | **69.357** | **0.878** | **14.125** | **87.000** | **0.860** | **10.716** | **66.534** | **0.874** |
| TIMEGUARD$_{\text{emb}}$ | 10.142 | 34.681 | 0.858 | 8.256 | 64.011 | 0.850 | 17.289 | 85.908 | 0.790 | 11.896 | 61.534 | 0.833 |

**Results.** As shown in Table 31, the embedding-based variant of TIMEGUARD still outperforms PDB across all models. However, it does not improve over the original input-space version. This suggests that generic learned embeddings such as TS2Vec do not automatically provide a stronger signal for TSF backdoor defense in our setting, consistent with previous observations on the limited effectiveness of generic data embeddings for time series forecasting (Nematirad et al., 2025). We leave the design of more dedicated embeddings for TSF backdoor defense to future work.

## G.7. Clean Performance under No Attack

*Table 32.* Clean performance (MAE$_C$ ↓) of in-training backdoor defenses under no attack scenario on PEMS03. Best results are in **bold**.

| Model →
Defense ↓ | SimpleTM | FEDformer | TimesNet | AVERAGE |
|---|---|---|---|---|
| Vanilla Training | 16.794 | 15.680 | 20.257 | 17.577 |
| ABL | **17.129** | 16.928 | 21.011 | 18.356 |
| PDB | 17.828 | 16.843 | 22.308 | 18.993 |
| ESTI | 17.396 | **15.915** | 20.119 | **17.810** |
| **TIMEGUARD** | 17.197 | 16.695 | **19.804** | 17.899 |

In realistic scenarios, the defender may not know whether the training set has been poisoned. We therefore evaluate an extreme setting where no poisoning is present (*No Attack*). Table 32 reports the clean forecasting performance of four in-training defenses on the PEMS03 dataset under this setting. Overall, ESTI and TIMEGUARD preserve clean accuracy well, with at most a 3.5% degradation across three models compared to vanilla training, and they even improve performance for TimesNet in some cases; both outperform ABL and PDB in this no-attack regime. However, ESTI incurs substantially higher training cost and is more prone to failing under attacks in our TSF setting (Appendix G.4 and Section 5.1). Taken together, these results suggest that TIMEGUARD offers a more practical trade-off when the poisoning status of the training data is unknown.

## G.8. Reliable Pool Dynamics Illustration

To illustrate how TIMEGUARD maintains a reliable pool during training, we plot (i) the number of poisoned samples admitted into the reliable pool at each epoch, together with the corresponding (ii) clean performance (MAE$_C$) and poisoned performance (MAE$_P$) of FEDformer. We compare TIMEGUARD against its loss-only variant (w/o NDF+DRLS) under the Random attack on three datasets, as shown in Figures 27–29 for PEMS03, Weather, and ETTm1, respectively. Overall, TIMEGUARD consistently admits fewer poisoned samples into the reliable pool than the loss-only variant, helping explain its strong robustness and competitive clean performance. These dynamics also highlight the importance of incorporating neighborhood-distance cues beyond loss-only criteria.

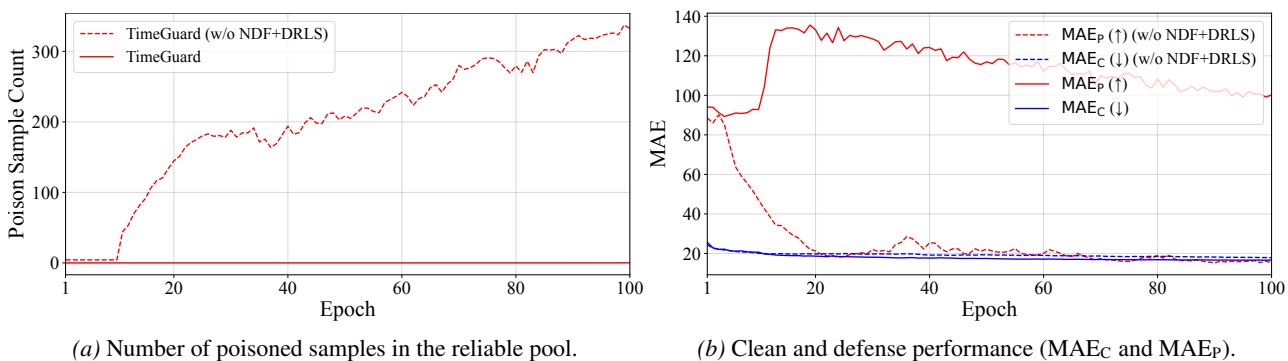

*(a)* Number of poisoned samples in the reliable pool.  *(b)* Clean and defense performance (MAE$_C$ and MAE$_P$).

*Figure 27.* Dynamic illustration of TIMEGUARD at each training epoch under Random attack on PEMS03 dataset of FEDformer model.

# H. Showcases

To better visualize the effectiveness of TIMEGUARD, we provide an inference-time prediction example for the FEDformer model under the BackTime attack on PEMS03 in Figure 27, where the showcased triggers are sampled from different channels and different test samples. Overall, TIMEGUARD preserves accurate forecasts on clean channels while substantially mitigating trigger-induced manipulation on poisoned channels. Moreover, even when the input window is perturbed by the trigger, TIMEGUARD can partially recover the underlying future trend. We also observe that the generated triggers exhibit similar shapes despite BackTime using sample-dependent triggers, which further supports our analysis in Theorem 4.1.

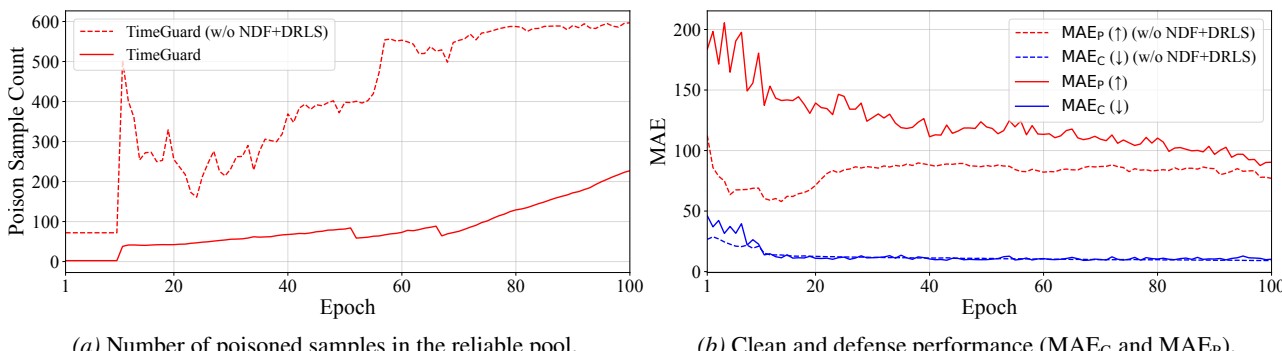

*(a)* Number of poisoned samples in the reliable pool.

*(b)* Clean and defense performance (MAE$_C$ and MAE$_P$).

*Figure 28.* Dynamic illustration of TIMEGUARD at each training epoch under Random attack on Weather dataset of FEDformer model.

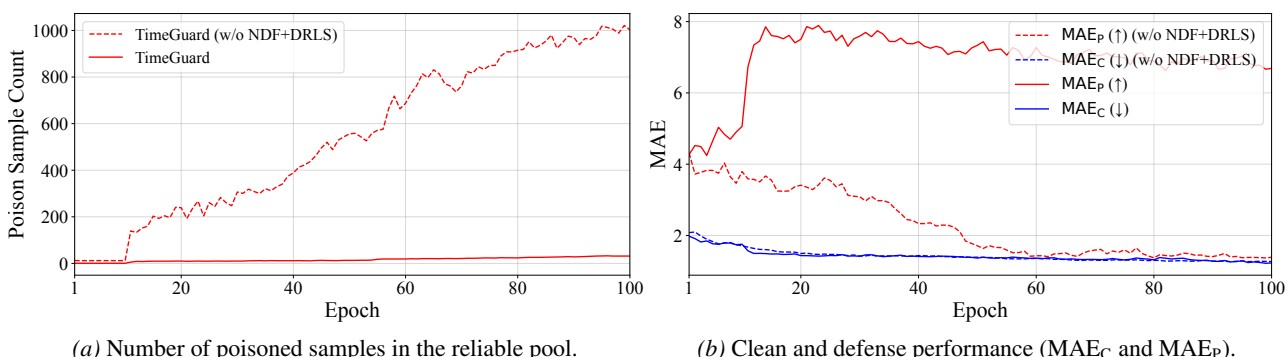

*(a)* Number of poisoned samples in the reliable pool.

*(b)* Clean and defense performance (MAE$_C$ and MAE$_P$).

*Figure 29.* Dynamic illustration of TIMEGUARD at each training epoch under Random attack on ETTm1 dataset of FEDformer model.

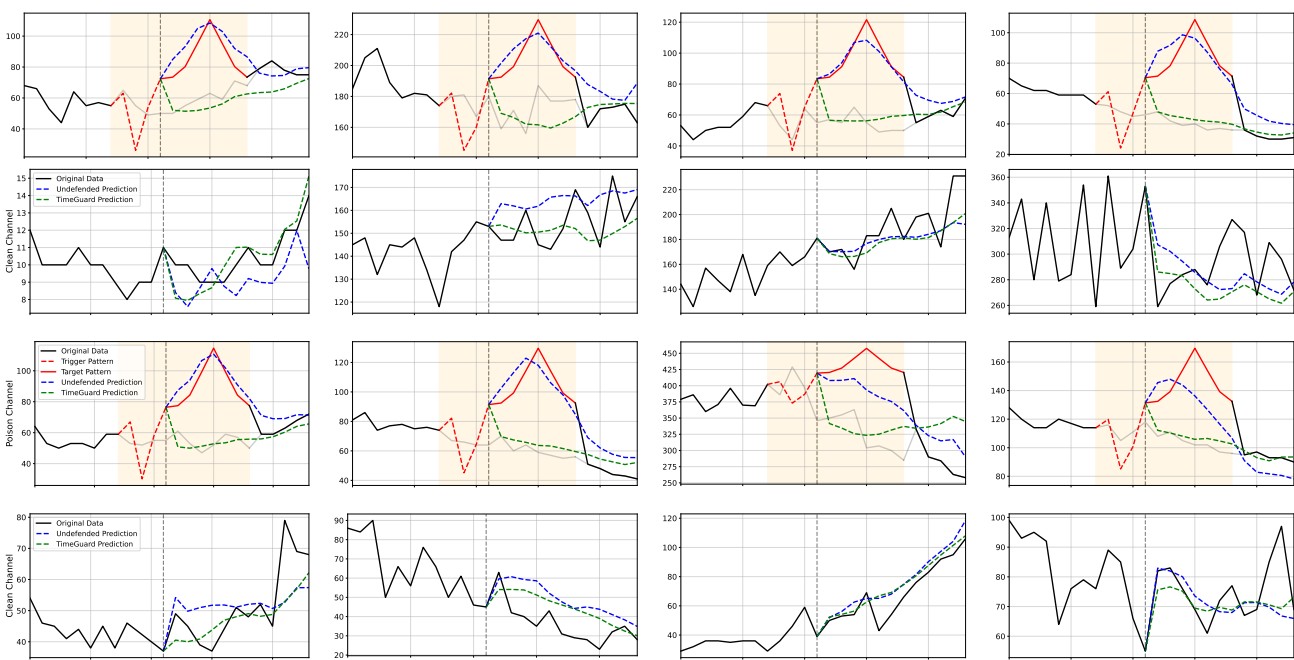

*Figure 30.* Inference-time prediction showcases of TIMEGUARD under the BackTime attack on PEMS03 of FEDformer model, visualized on alternating poisoned and clean channels. We display a randomly selected test sample with a randomly selected channel.

# I. Limitations and Future Work

**Limitations.** First, TIMEGUARD relies on a hand-designed neighborhood metric, e.g., correlation-/distance-based $k$NN on normalized windows, to construct and refine the reliable pool. Such input-space distances may degrade under strong distribution shifts or nonstationarity. Our preliminary experiments suggest that this degradation is moderate rather than catastrophic; however, existing training-phase defenses also suffer under these challenging settings, as discussed in Appendix G.1. Second, TSF backdoor defenses in general face an inherent precision–recall trade-off when the trigger and induced target are not "out-of-distribution" relative to clean dynamics. If the trigger and target mimic prevalent motifs (e.g., a near-linear upward trend), poisoned and clean windows can be ambiguous in both learning-based and neighborhood structure: filtering/detecting may remove frequent clean patterns, while retaining them may preserve backdoor influence.

**Future work.** A natural direction is to augment our input-space $k$NN with TSF-specific embedding spaces where neighborhoods better reflect forecasting semantics, e.g., via self-supervised or contrastive representations (Zhang et al., 2024a; Zheng et al., 2025). Although we conduct preliminary experiments using TS2Vec embeddings (Yue et al., 2022), as discussed in Appendix G.6, the results remain unsatisfactory and do not outperform our original input-space implementation. Future work could explore representations specifically tailored to TSF backdoor defense. Another direction is to utilize (rather than discard) the unreliable pool with semi-supervised learning (Cho & Lee, 2025); however, current TSF semi-supervised methods are often architecture-dependent, motivating deeper study of architecture-agnostic formulations under backdoor settings. Finally, multivariate TSF offers opportunities to leverage cross-channel structure (e.g., dependency graphs or causal signals (Qiu et al., 2025; Han et al., 2025)) to localize corrupted channels while improving clean and recovery forecasting performance.

More broadly, we hope this work encourage TSF-specific backdoor defense research and time series security in general, including standardized benchmarks, stronger adaptive attacks/defenses, and principled evaluation protocols.

