# OpenReview forum: "TimeGuard: Channel-wise Pool Training for Backdoor Defense in Time Series Forecasting"
_ICML.cc/2026/Conference — ICML 2026 regular_

### Official Review · Reviewer_JgZW · 2026-03-03

**Soundness:** 3
**Presentation:** 3
**Significance:** 2
**Originality:** 3
**Overall Recommendation:** 4
**Confidence:** 4

**Summary:**

This paper systematically investigates the vulnerabilities of Time Series Forecasting (TSF) models to backdoor attacks and explores the failure modes of existing defense strategies. Based on these insights, the authors introduce TIMEGUARD, an unsupervised, training-stage backdoor defense framework. The proposed method shifts the paradigm from traditional sample-level defense to a fine-grained, channel-wise strategy. Extensive experiments demonstrate that TIMEGUARD can effectively mitigate various backdoor attacks without relying on any external clean data, while preserving high forecasting accuracy on clean data.

**Compliance With Llm Reviewing Policy:**

Affirmed.

**Final Justification:**

I appreciate the author's rebuttal, which have addressed my concerns. Therefore, I raised my score to weak accept. I suggest the authors put the additional experiment results into the revised version of the paper. I believe this will further strengthen the paper.

**Key Questions For Authors:**

- Considering that NDF strongly relies on the high similarity of poisoned samples, would the NDF mechanism still be effective if an attacker is aware of the TIMEGUARD defense mechanism (i.e., a white-box or adaptive attack) and employs dynamic triggers or optimization-based non-clustered triggers?
- In practical applications, time series data frequently experiences concept drift. When normal data undergoes drastic trend changes, it exhibits large neighborhood distances and high prediction losses, which closely resemble the characteristics of the rejected samples described by the authors. How does the TIMEGUARD framework ensure that these critical clean samples, which represent new trends, are not permanently isolated in the unreliable pool?
- Minor typos
  - line 207, right column, tofine-tuning-basedd

**Limitations:**

yes

**Strengths And Weaknesses:**

### Strengths

- **Clear motivation based on preliminary experiments**: The analysis regarding why existing backdoor defenses fail in the TSF domain is thorough, providing a highly compelling and clear motivation for the work.
- **Comprehensive methodology**: The authors design RCF and NDF specifically around the temporal asymmetry and local channel characteristics unique to TSF data, resulting in a logically rigorous framework.
- **Generalizability**: The defense does not require access to internal model features or a clean validation dataset. It can be seamlessly integrated into various mainstream TSF architectures, demonstrating significant practical utility.

### Weaknesses

- **Robustness to non-stationary data**: The Neighborhood Diversity Filtering (NDF) relies on distance metrics that might misclassify natural sudden changes as poisoned samples when applied to highly non-stationary time series or data exhibiting severe concept drift.
- **Increased computational overhead**: Training an auxiliary backcaster and repeatedly computing Gaussian-weighted neighborhood distance matrices across the training process inevitably leads to noticeably higher training times and computational costs compared to standard training. How is such latency measured and compared with baselines?
- **Strong assumption of channel-wise poisoning**: The proposed defense assumes that backdoor injection in multivariate TSF only modifies a subset of data channels. It seems that this condition doe not necessarily hold for all attacks. Why cannot one perform poisoning on all channels of a time series for a short time window? How would the proposed defense perform under such conditions?

---

> ### Author Rebuttal · Authors · 2026-03-31
>
> Thank you for your valuable comments. Below we address the main concerns.
> > **Q1. Robustness to non-stationary data.**
>
> **A:** Thank you for your comment. We clarify that NDF does not reject samples with large neighborhood distance; rather, poisoned windows are expected to exhibit abnormally small neighborhood distances, so natural sudden changes are not automatically treated as poisoned by NDF alone. More importantly, TimeGuard does not permanently discard samples once excluded. Through DRLS, the reliable pool is progressively expanded during training, so clean samples associated with new trends can be re-admitted once the model adapts and their loss decreases. We agree that under severe non-stationarity or concept drift this process becomes more challenging, and we will clarify this limitation more explicitly in the revision.
>
> > **Q2.  Increased computational overhead.**
>
> **A:** Thank you for your comment and question. TimeGuard does introduce additional training overhead from two sources: the auxiliary backcaster in Stage I and the computation of Gaussian-weighted neighborhood distances. We measure this overhead using wall-clock training time and compare it directly against No Defense and PDB under the same experimental setup. For the backcaster cost, please refer to reviewer RWuK's Q1; for the efficient neighborhood-search implementation that avoids repeatedly recomputing distances, please refer to reviewer xoJG's Q4. We also report efficiency analysis in Appendix G.5, and refer to xoJG's Q3 and Q7 for additional large-scale runtime results.
>
> > **Q3. Strong assumption of channel-wise poisoning.**
>
> **A:** Thank you for your concern. Our motivation is strongest under partial-channel poisoning, since this is the common setting in existing multivariate TSF attacks, but TimeGuard itself does not require poisoning to affect only a strict subset of channels. The channel-wise formulation remains applicable even when poisoning becomes dense across channels. To directly test the all-channel case, we evaluate TimeGuard on PEMS03 under BackTime with spatial poisoning ratio $\eta_S=1.0$, meaning that all channels are poisoned. As shown below, TimeGuard remains effective in this setting and achieves the highest FDER ($0.748$). We will clarify in the revision that the method is not restricted to partial-channel attacks, although the channel-wise motivation is most natural there.
>
> | Method        |    $\mathrm{MAE}\_\mathrm{C}$  ↓ |     $\mathrm{MAE}\_\mathrm{P}$  ↑ |    $\mathrm{FDER}$ ↑ |
> | ------------- | ---------: | ----------: | --------: |
> | No Defense    |     21.355 |      12.969 |         - |
> | PDB           | 20.384	| 20.638	| 0.690|
> | **TimeGuard** |     **19.871** |	**25.335**|	**0.748**|
>
>
> > **Q4. Would NDF still remain effective under a white-box adaptive attack using dynamic or non-clustered triggers?**
>
> **A:** Thank you for your question. We evaluate a representative adaptive setting in Section 5 and Appendix G.4, where the attacker uses a dynamic, optimization-based trigger by encouraging reverse consistency and penalizing high correlation among poisoned samples. Even under this stronger white-box attack, TimeGuard still achieves the best overall trade-off. We further evaluate ablations without the neighborhood-based components (w/o NDF and w/o DRLS). As shown below, removing NDF slightly reduces FDER, while removing DRLS causes a much larger drop, indicating that neighborhood-based cues remain effective even under this adaptive setting.
>
> | Method        |    $\mathrm{MAE}\_\mathrm{C}$  ↓ |     $\mathrm{MAE}\_\mathrm{P}$  ↑ |    $\mathrm{FDER}$ ↑ |
> | ------------------ | --------------------------: | --------------------------: | ----------------: |
> | No Defense         | 18.791 | 15.343 | - |
> | TimeGuard     | **18.438** | **30.575** |  **0.744** |
> | TimeGuard w/o DRLS |                      20.863 |                      19.026 |             0.543 |
> | TimeGuard w/o NDF  |                      18.564 |                      29.695 |             0.739 |
>
> > **Q5. How does TimeGuard avoid permanently excluding clean samples under concept drift or sudden trend changes?**
>
> **A:** Thank you for your concern and question. TimeGuard does not permanently isolate samples once they are excluded early on. Instead, the reliable pool is progressively expanded during training through distance-regularized loss selection (DRLS), which admits samples that become both low-loss and still maintain sufficient neighborhood diversity. Therefore, clean samples corresponding to new trends may be excluded at first, but can be gradually incorporated once the model adapts to the new regime and their prediction loss decreases. We also provide additional experiments under strong concept drift in Reviewer xoJG's Q1/Q9.
>
> > **Q6. Minor typos line 207, right column, tofine-tuning-basedd.**
>
> **A:** Thank you, we will revise this in the revision.

---

> > ### Author Rebuttal · Reviewer_JgZW · 2026-04-02
> >
> > Thanks for the authors' detailed response. The rebuttal clarifies several concerns (e.g., NDF mechanism) and provides additional experiments (e.g., all-channel and adaptive attacks), which strengthen the empirical support.
> > However, my concern regarding robustness under non-stationary settings and concept drift is not fully resolved. I suggest authors put them into the limitation discussions of the paper.

---

> > > ### Author Response · Authors · 2026-04-04
> > >
> > > Thank you for your acknowledgement and the remaining concern. Below we address the concern.
> > >
> > >
> > > > **Q. However, my concern regarding robustness under non-stationary settings and concept drift is not fully resolved. I suggest authors put them into the limitation discussions of the paper.**
> > >
> > > **A:** We agree that TimeGuard may face limitations under non-stationary settings and concept drift, as discussed in Appendix I. To further examine robustness under mild distribution change, we conduct an additional experiment on PEMS03 under the BackTime attack by introducing synthetic distribution shifts at test time. Specifically, let $x_{t,c}$ denote the value at time step $t$ and channel $c$,  and let $\sigma_c$ denote the standard deviation of channel $c$ computed from the training dataset. We consider three perturbations:
> > >
> > > - Scale shift: $x'\_{t, c} = (1 + \alpha)x\_{t,c}$
> > > - Mean shift: $x'\_{t, c} = x\_{t,c} + \alpha\sigma_c$
> > > - Linear trend: $x'\_{t,c} = x\_{t,c} + \alpha\sigma_c\frac{t}{T}$, where $T$ is the length of test split.
> > >
> > > We test two shift strengths, $\alpha \in \\{0.1, 0.2\\}$. The results are shown below.
> > >
> > > | Shift / Strength | Defense       |    $\mathrm{MAE}_\mathrm{C} ↓$  |    $\mathrm{MAE}_\mathrm{P} ↑$ |  $\mathrm{FDER} ↑$ |
> > > | ---------------- | ------------- | ---------: | ---------: | --------: |
> > > | No shift         | No Defense    |     17.607 |     14.201 |         - |
> > > |                  | PDB           |     18.967 |     22.397 |     0.639 |
> > > |                  | **TimeGuard** | **18.048** | **39.303** | **0.808** |
> > > | Scale shift ($\alpha = 0.1$)      | No Defense    |     20.668 |     16.049 |         - |
> > > |                  | PDB           |     22.125 |     24.839 |     0.637 |
> > > |                  | **TimeGuard** | **20.546** | **39.979** | **0.789** |
> > > | Scale shift ($\alpha = 0.2$)      | No Defense    |     23.590 |     19.682 |         - |
> > > |                  | PDB           |     25.240 |     27.555 |     0.605 |
> > > |                  | **TimeGuard** | **23.305** | **40.874** | **0.751** |
> > > | Linear trend ($\alpha = 0.1$)        | No Defense    | 18.323|     16.049 |         - |
> > > |                  | PDB           |     19.657 |     23.300 |     0.609 |
> > > |                  | **TimeGuard** |     **18.351** | **39.963** | **0.790** |
> > > | Linear trend ($\alpha = 0.2$)        | No Defense    | 18.450 |     17.239 |         - |
> > > |                  | PDB           |     19.829 |     23.919 |     0.605 |
> > > |                  | **TimeGuard** |     **18.505** | **40.465** | **0.780** |
> > > | Mean shift ($\alpha = 0.1$)       | No Defense    | 18.419 |     15.040 |         - |
> > > |                  | PDB           |     19.774 |     23.903 |     0.641 |
> > > |                  | **TimeGuard** |     **18.476** | **40.512** | **0.806** |
> > > | Mean shift ($\alpha = 0.2$)      | No Defense    |     19.050 |     16.159 |         - |
> > > |                  | PDB           |     20.569 |     25.332 |     0.637 |
> > > |                  | **TimeGuard** | **19.030** | **41.581** | **0.800** |
> > >
> > >
> > > As shown above, the performance of all methods declines slightly under synthetic distribution shift. However, TimeGuard consistently achieves the best defense performance across all six shifted settings. This suggests that, although non-stationarity does affect performance, the negative impact is moderate rather than catastrophic, and TimeGuard remains stable in practice under mild distribution shift.
> > >
> > > Under strongly non-stationary or concept drift scenarios, any training-phase defense that will face challenges, and TimeGuard is no exception. That said, because DRLS does not permanently discard excluded samples, some clean samples that are initially filtered out may still be reintroduced as the model gradually adapts. We will clarify this limitation more explicitly and  include the full results in the revision.

---

### Official Review · Reviewer_RWuK · 2026-03-11

**Soundness:** 3
**Presentation:** 3
**Significance:** 3
**Originality:** 3
**Overall Recommendation:** 4
**Confidence:** 2

**Summary:**

The paper systematically evaluates existing backdoor defenses originally designed for classification after adapting them to time-series forecasting (TSF), and finds two main reasons these adapted defenses fail. To address this, the paper proposes TIMEGUARD, a training-time defense that performs channel-wise reliable-pool training. Experiments across multiple TSF datasets, victim forecasters, and attack settings show that TIMEGUARD significantly reduces attack effectiveness.

**Compliance With Llm Reviewing Policy:**

Affirmed.

**Final Justification:**

My concerns have been addressed. Thank you!

**Key Questions For Authors:**

Questions are basically the weaknesses:

1. A backcaster needs to be trained during training phase which increased additional expenses, especially in large-scale multi-channel data. Also add training-time runtime/memory overhead for TIMEGUARD, since the paper critiques inference-time defenses partly on efficiency grounds in Table 2.
2. The assumptions are strong, especially the bounded deviation of all background labels from the backdoor target and local Lipschitzness of the target mapping around poisoned samples.  And the theory supports the method mainly as intuition, not as a close analysis of TIMEGUARD itself.
3. The recommendation π≤1.5 is empirically supported on PEMS03 under BackTime, but it is unclear whether the same threshold transfers across datasets/attacks?
4. How about the defense performance of TimeGuard against Badtime attack [1]?
[1]Xiang K, Yang H, Hao M, et al. BadTime: An Effective Backdoor Attack on Multivariate Long-Term Time Series Forecasting.

**Limitations:**

yes

**Strengths And Weaknesses:**

Strengths:
1. The paper systematically evaluated backdoor defenses for time series forecasting (TSF) and pointed out the two causes for the failure of "directly moving classified defense to TSF".
2. The paper proposed TimeGuard to defend backdoor attacks in TSF without requiring additional samples.
3. Empirical evaluation has a wide coverage (multiple datasets, multiple models, and multiple attacks) and many baseline comparisons and ablation studies.

Weaknesses:
1. A backcaster needs to be trained during training phase which increased additional expenses, especially in large-scale multi-channel data. Also add training-time runtime/memory overhead for TIMEGUARD, since the paper critiques inference-time defenses partly on efficiency grounds in Table 2.
2. The assumptions are strong, especially the bounded deviation of all background labels from the backdoor target and local Lipschitzness of the target mapping around poisoned samples.  And the theory supports the method mainly as intuition, not as a close analysis of TIMEGUARD itself.
3. The recommendation π≤1.5 is empirically supported on PEMS03 under BackTime, but it is unclear whether the same threshold transfers across datasets/attacks?
4. How about the defense performance of TimeGuard against Badtime attack [1]?
[1]Xiang K, Yang H, Hao M, et al. BadTime: An Effective Backdoor Attack on Multivariate Long-Term Time Series Forecasting.

---

> ### Author Rebuttal · Authors · 2026-03-31
>
> Thank you for your comments. Below we address the main concerns.
> > **Q1. "A backcaster needs to be trained during training phase which increased additional expenses, especially in large-scale multi-channel data. Also add training-time runtime/memory overhead for TIMEGUARD, since the paper critiques inference-time defenses partly on efficiency grounds in Table 2."**
>
> **A:** Thank your for your comment. We agree that TimeGuard introduces additional training-time overhead, mainly from Stage I backcaster training and neighborhood-based filtering. However, this cost is limited and temporary: Stage I runs for only $T_b$ epochs, which we set to about $10%$ of the total training budget by default. After Stage I, the backcaster is discarded, so its extra memory overhead does not persist into the main training stage. Appendix G.5 reports the corresponding runtime overhead, and we will make this training-time runtime/memory cost more explicit in the revision. Importantly, this is different from the repeated inference-time overhead discussed in Table 2, since TimeGuard does not add extra latency during deployment.
> > **Q2. "The assumptions are strong, especially the bounded deviation of all background labels from the backdoor target and local Lipschitzness of the target mapping around poisoned samples. And the theory supports the method mainly as intuition, not as a close analysis of TIMEGUARD itself."**
>
> **A:** Thank you for your comment. We agree that the assumptions in Theorem 4.1 are stylized, and that this analysis is better viewed as a mechanism-level explanation rather than a tight end-to-end guarantee for TimeGuard. In particular, the bounded-background-deviation and local-Lipschitz conditions are introduced to make the attack-success bound tractable, not to fully characterize the complete behavior of TimeGuard. Our goal is therefore to explain why successful TSF backdoors tend to induce concentrated poisoned neighborhoods, which motivates NDF/DRLS, rather than to claim a close theoretical analysis of the full method. We will revise the paper to make this positioning more explicit and avoid overstating the scope of the theory. The main support for TimeGuard remains the systematic empirical results, including the adaptive-attack evaluation in Appendix G.4.
> > **Q3. "The recommendation π≤1.5 is empirically supported on PEMS03 under BackTime, but it is unclear whether the same threshold transfers across datasets/attacks?"**
>
> **A:** Thank you for your concern. To assess whether the recommendation on $\pi$ generalizes beyond PEMS03 under BackTime, we additionally conduct sensitivity studies on Weather under BackTime and PEMS03 under Random. The results show a consistent pattern: performance remains stable across a broad range of $\pi \in [1.05, 1.50]$, while it degrades when $\pi$ is increased to 1.65. These results suggest that $\pi \leq 1.5$ should be viewed as a robust empirical operating range, rather than a threshold tuned only to PEMS03 under BackTime. We will include the full results in the revision.
>
> **Weather under BackTime**
>
> | Method / $\pi$ | $\mathrm{MAE}\_\mathrm{C}$  ↓ | $\mathrm{MAE}\_\mathrm{P}$  ↑ | $\mathrm{FDER}$ ↑ |
> | -------------- | ------: | ------: | -----: |
> | No Defense     |  10.768 |  15.913 |      - |
> | $\pi = 1.05$   |  11.497 |  65.994 |  0.851 |
> | $\pi = 1.15$   |  11.183 |  59.566 |  0.851 |
> | $\pi = 1.25$   |  11.169 |  64.773 |  0.862 |
> | $\pi = 1.35$   |  11.098 |  70.362 |  0.877 |
> | $\pi = 1.50$   |  11.218 |  67.949 |  0.866 |
> | $\pi = 1.65$   |  10.972 |  59.584 |  0.857 |
>
>
> **PEMS03 under Random**
> | Method / $\pi$ | $\mathbf{\mathrm{MAE}\_\mathrm{C}}$  ↓ | $\mathrm{MAE}\_\mathrm{P}$  ↑ | $\mathrm{FDER}$ ↑ |
> | -------------- | ------: | ------: | -----: |
> | No Defense     |  17.634 |  17.772 |      - |
> | $\pi = 1.05$   |  17.779 | 107.413 |  0.869 |
> | $\pi = 1.15$   |  17.770 | 104.048 |  0.870 |
> | $\pi = 1.25$   |  18.009 | 103.645 |  0.865 |
> | $\pi = 1.35$   |  18.150 | 101.274 |  0.861 |
> | $\pi = 1.50$   |  18.262 |  98.333 |  0.858 |
> | $\pi = 1.65$   |  18.497 |  97.536 |  0.850 |
> > **Q4. "How about the defense performance of TimeGuard against Badtime attack [5]? "**
>
> **A:** Thank you for your question. We additionally evaluate TimeGuard against BadTime [5] on PEMS03. As shown below, TimeGuard remains effective under this recent attack setting and achieves the best overall trade-off, with the highest FDER while maintaining competitive clean forecasting performance. We will include the full results in the revision.
>
>
> | Method        |    $\mathrm{MAE}\_\mathrm{C}$  ↓ |     $\mathrm{MAE}\_\mathrm{P}$  ↑ |    $\mathrm{FDER}$ ↑ |
> | ------------- | ---------: | ----------: | --------: |
> | No Defense    |     21.698 |      20.918 |         - |
> | PDB           | 21.739	| 37.862	| 0.715|
> | **TimeGuard** |     **19.332** |	**38.084**|	**0.847**|

---

> > ### Author Rebuttal · Reviewer_RWuK · 2026-04-03
> >
> > My concerns have been addressed. Thank you!

---

### Official Review · Reviewer_xoJG · 2026-03-12

**Soundness:** 3
**Presentation:** 3
**Significance:** 4
**Originality:** 3
**Overall Recommendation:** 4
**Confidence:** 3

**Summary:**

Summary
The paper studies the problem of defending against backdoor attacks in time series forecasting (TSF). The research attempts to outline an important concept: security vulnerabilities of forecasting models under poisoned training data. The authors explore an important concept by analyzing why existing backdoor defenses designed for classification fail in TSF settings.

To address these issues, the paper proposes TIMEGUARD, an in-training defense that constructs a channel-wise reliable pool of training samples. The method combines reverse-consistency filtering (RCF), neighborhood diversity filtering (NDF), and distance-regularized loss selection (DRLS) to progressively identify clean samples and reduce the influence of poisoned data during training. Extensive experiments on multiple datasets and forecasting backbones demonstrate that TIMEGUARD effectively mitigates backdoor attacks while preserving clean forecasting performance.

**Compliance With Llm Reviewing Policy:**

Affirmed.

**Final Justification:**

all my concerns have been addressed

**Key Questions For Authors:**

- How sensitive is TIMEGUARD to the choice of neighborhood size K and other hyperparameters in different datasets?

- Could attackers design adaptive triggers that mimic common temporal patterns to evade the neighborhood-based filtering criteria?

- How well would the defense perform under large-scale foundation models for time series forecasting?

- Would embedding-based distances or learned representations provide more robust signals than input-space distance metrics?

**Limitations:**

- The defense depends on input-space neighborhood distances, which may degrade under strong distribution shifts or nonstationary dynamics. :contentReference[oaicite:5]{index=5}

- When triggers mimic common temporal patterns, separating poisoned and clean samples becomes inherently difficult due to the precision–recall trade-off in filtering. :contentReference[oaicite:6]{index=6}

- The method focuses on training-time defenses and does not explore inference-time detection or hybrid defense strategies.

- The approach currently assumes continuous-valued time series and may require adaptation for other types of temporal data.

**Strengths And Weaknesses:**

#  Pros
- The paper studies an important and relatively underexplored problem: backdoor attacks in time series forecasting. This direction is timely as forecasting systems are increasingly deployed in real-world applications. :contentReference[oaicite:0]{index=0}

- The paper provides a clear analysis of why many existing backdoor defenses fail in TSF due to data entanglement and task-formulation differences between classification and forecasting. This insight motivates the proposed defense design. :contentReference[oaicite:1]{index=1}

- The proposed method is conceptually simple and practical. TIMEGUARD does not require additional clean data, internal model features, or additional inference overhead, making it applicable to different forecasting models. :contentReference[oaicite:2]{index=2}

- Experiments are fairly comprehensive. The method is evaluated across multiple datasets, attacks, and forecasting architectures, showing consistent improvements in defense performance. :contentReference[oaicite:3]{index=3}

# cons

- The method relies on hand-crafted distance metrics and kNN-based neighborhood analysis in input space, which may be sensitive to distribution shift and nonstationary time series. :contentReference[oaicite:4]{index=4}

- The defense relies on heuristic filtering rules (e.g., reverse consistency and neighborhood diversity), and the theoretical guarantees of these signals under stronger adaptive attacks remain unclear.

- The evaluation focuses on a limited set of attack settings and datasets, leaving questions about scalability to more diverse forecasting tasks and larger datasets.

- While the method is model-agnostic, the computational cost during training (e.g., neighborhood search and filtering) may increase for very large datasets.

---

> ### Author Rebuttal · Authors · 2026-03-31
>
> Thank you for your valuable comments. Below we address the main concerns.
> > **Q1/Q9. kNN/input-space distances may be sensitive and degrade under shift/nonstationarity.**
>
> **A:** We agree and already note this in Appendix I. We add Exchange/BackTime [7], a financial benchmark with evolving dynamics. TimeGuard remains effective and outperforms PDB under this stronger shift/nonstationarity setting, although we agree that such conditions are inherently more difficult.
>
> | Method | MAE_C↓ | MAE_P↑ | FDER↑ |
> |---|---:|---:|---:|
> | No defense | 0.01585 | 0.04982 | - |
> | PDB | 0.02574 | 0.11610 | 0.66704 |
> | **TimeGuard** | **0.01768** | **0.11911** | **0.77660** |
> > **Q2. Filtering rules are heuristic; guarantees under adaptive attack are unclear.**
>
> **A:** Our filtering rules are heuristic but not arbitrary: they are motivated by Theorem 4.1 and TSF temporal asymmetry. We do not claim a formal guarantee against fully adaptive attackers; instead, we provide mechanism-level and empirical support, including a worst-case adaptive attack in Appendix G.4, under which TimeGuard remains effective.
> > **Q3/Q4. Limited scalability to larger datasets; training cost may grow.**
>
> **A:** We add GBA [8] (2019 subset), a much larger traffic benchmark (35040×2352). TimeGuard still achieves the highest FDER. The cost increases, but this is offline training overhead rather than deployment latency. To reduce it, we precompute the channel-wise neighbor graph once, cache top-$K_{\max}$ neighbors, and reuse them throughout training. For broader task diversity, see Q12.
>
> | Method | MAE_C↓ | MAE_P↑ | FDER↑ | Training time(s) |
> |---|---:|---:|---:|---:|
> | No defense | 30.640 | 35.036 | - | 6309.7 |
> | PDB | **30.479** | 46.280 | 0.622 | 6862.3 |
> | **TimeGuard** | 33.328 | **67.650** | **0.698** | 20745.8 |
> > **Q5. Sensitivity to K and other hyperparameters.**
>
> **A:** On Weather/BackTime, TimeGuard still remains stable: $ K\in[10,64] $ gives FDER 0.831–0.888; $\pi\in[1.05,1.65]$ gives 0.828–0.893; $\alpha\in[0.10,0.30] $ gives 0.860–0.888. The main sensitivity is $\beta$: $\beta\in[0.40,0.80]$ gives 0.625–0.936, with 0.4–0.6 best.
> > **Q6/Q10. Triggers that mimic common temporal patterns.**
>
>
> **A:** We test this with Manhattan [3], which retrieves segments closest to the target under $L_1$ distance and uses preceding windows as triggers. Although this attack overlaps more with clean motifs, TimeGuard still gives the best defense across all three datasets (Appendix G.1, Table 8).
> > **Q7. Performance on large TSF foundation models?**
>
> **A:** Beyond $\text{AutoTimes}\_{\text{GPT2}}$ and $\text{AutoTimes}\_{\text{OPT1B}}$, we further evaluate $\text{AutoTimes}_{\text{LLaMA7B}}$. TimeGuard still outperforms PDB while keeping total training time to 1.431× that of no defense. This suggests transfer beyond standard TSF backbones.
>
> | Method | Training time(s) | MAE_C↓ | MAE_P↑ | FDER↑ |
> |---|---:|---:|---:|---:|
> | No Defense | 49027.6 | 20.977 | 6.004 | - |
> | PDB | **69997.6** | 22.657 | 25.798 | 0.847 |
> | **TimeGuard** | 70162.8 | **21.385** | **32.792** | **0.899** |
> > **Q8. Would learned/embedding distances be better?**
>
> **A:** We replace our Gaussian-weighted input-space distance with a TS2Vec embedding [9] distance on Weather/BackTime. The embedding variant still beats PDB, but does not improve over the original version, so generic embeddings do not automatically provide a stronger signal in our setting [10].
>
> | Method | MAE_C↓ | MAE_P↑ | FDER↑ |
> |---|---:|---:|---:|
> | No defense | 10.768 | 15.913 | - |
> | PDB | 11.732 | 56.439 | 0.827 |
> | **TimeGuard** | **10.716** | **66.534** | **0.874** |
> | TimeGuard_emb | 11.896 | 61.534 | 0.833 |
> > **Q11. Why only training-time defense, not inference-time / hybrid?**
>
> **A:** TSF is often deployed in continuous real-time settings, where inference-time checks can be costly (Appendix B.1). TimeGuard therefore focuses on training-time defense and adds no inference-time overhead. We leave inference-time or hybrid defenseS for future work.
> > **Q12. Continuous-valued assumption; what about other temporal data types?**
>
> **A:** We add Bike Sharing [11], a count-valued dataset, under the Random attack. TimeGuard still achieves the best defense performance, suggesting preliminary transfer beyond continuous-valued TSF, while broader adaptation remains future work.
>
> | Method | MAE_C↓ | MAE_P↑ | FDER↑ |
> |---|---:|---:|---:|
> | No Defense | 22.814 | 60.421 | - |
> | PDB | **21.990** | 116.119 | 0.739 |
> | **TimeGuard** | 23.935 | **240.010** | **0.831** |
>
> ---
> [7] Lai et al. *Modeling Long- and Short-Term Temporal Patterns with Deep Neural Networks*. SIGIR 2018.
>
> [8] Liu at el. *LargeST: A Benchmark Dataset for Large-Scale Traffic Forecasting*. NeurIPS 2023.
>
> [9] Yue at el. *TS2Vec: Towards Universal Representation of Time Series*. AAAI 2022.
>
> [10] Nematirad at el. *Are Data Embeddings Effective in Time Series Forecasting?*. TMLR 2025.
>
> [11] Fanaee-T at el. *Bike Sharing [Dataset]*. UCI Machine Learning Repository 2013.

---

> > ### Author Rebuttal · Reviewer_xoJG · 2026-04-04
> >
> > all my concerns have been addressed

---

### Official Review · Reviewer_rwL3 · 2026-03-13

**Soundness:** 1
**Presentation:** 2
**Significance:** 2
**Originality:** 3
**Overall Recommendation:** 4
**Confidence:** 4

**Summary:**

This paper addresses backdoor defense in multivariate time series forecasting (TSF). The authors first conduct an  evaluation of 13 existing backdoor defenses adapted to TSF, identifying two failure modes They then propose timeguard, an in-training defense that operates at channel-wise granularity rather than sample level.

**Compliance With Llm Reviewing Policy:**

Affirmed.

**Final Justification:**

After reading authors' feedback, I changed the rating to weak accept.

**Key Questions For Authors:**

See previous comments.

**Strengths And Weaknesses:**

++ An important area that is worth developing.

-- Core failure mode argument is not sound

The paper argues that representation dilution is TSF-specific because clean channels dominate the joint representation. However, image backdoors also go through convolution, pooling, and mixing with a small patch — yet defenses work. The authors do not provide support or distinguish why the mixing argument applies differently here.

Similarly, the paper argues that because poisoned and clean sample losses converge within a few epochs (Figure 3), defenses relying on loss-based separation fail in TSF due to the regression objective. However, In any setting, classification or regression, once the model has absorbed the backdoor pattern into its weights, training loss on poisoned samples will naturally drop. A well-executed image backdoor would produce exactly the same loss convergence behavior. The authors do not demonstrate empirically that loss-based separation remains reliable in classification under comparably strong attacks, nor do they cite evidence that this distinction holds in practice.

-- Using the loss function as the decision for attack/defense success or failure is misleading.

A high MAEP is interpreted as the defense succeeding. MAEP can increase simply because the model's overall forecasting quality has degraded, not because the backdoor was specifically suppressed. A model that forecasts poorly on everything will trivially have high MAEP. The authors further proposes FDER to address the issue, which further complicates the support.

However, the author didn't address the root cause. The underlying problem is that the paper didn't have a clear definition between what is benign, what is malicious and what is simply wrong. Without this measure, a wrong example can be simply a successful attack without constraint and a successful defense can be merely a degraded model. The author needs to answer the question, why backdoor sample in time series should have a different trajectory than the bad pattern? Should they share similar trajectory as the benign sample?

---

> ### Author Rebuttal · Authors · 2026-03-31
>
> Thank you for your comments. Below we address the main concerns.
> > **Q1. "Core failure mode argument is not sound."**
>
> **A:** We agree that our original claim could have been clearer. Our point is not that these phenomena occur exclusively in TSF, but that when existing backdoor defenses are transferred to multivariate TSF, the signals they rely on become substantially less reliable.
>
>
> For the representation-dilution point, the key issue is not simply that signals are mixed, but that the attack granularity and defense granularity are mismatched. In multivariate TSF, channel structure is a core part of both modeling and attack design [1, 2], and under current TSF threat models [3, 4], attackers typically poison only a subset of channels. As a result, transferred defenses that make only sample-level decisions can be dominated by the many clean channels within the same window, making poisoned signals much harder to localize. Our method therefore reformulates defense from a sample-level decision into a time × channel-wise one. Empirically, our ablation shows that removing the channel-wise formulation substantially reduces defense effectiveness, supporting the importance of matching defense granularity to the attack setting (Table 5).
>
> For the loss-convergence point, we agree that fast loss convergence alone is not unique to TSF. Our claim is narrower: when loss-based defenses are transferred to TSF, forward-loss-only early separation becomes particularly weak due to the continuous regression objective together with overlapping input-output windows (Figure 3 and Table 1). This is why TimeGuard does not rely on forward loss alone, but instead combines reverse-consistency and neighborhood-diversity signals. Consistently, replacing Distance-Regularized Loss Selection with loss-only selection reduces effectiveness by 28% on average in our ablation (Table 5).
>
> We will revise the paper to make this scope clearer and avoid overstating these effects as exclusive to TSF, while retaining our main conclusion: these signals become markedly less reliable when existing defenses are transferred to multivariate TSF.
> > **Q2. "Using the loss function as the decision for attack/defense success or failure is misleading."**
>
> **A:** We agree that $\mathrm{MAE}\_\mathrm{P}$ alone is insufficient, since a poorly performing model could trivially increase error under triggered inputs without actually mitigating the backdoor.
>
> For this reason, our evaluation does not use $\mathrm{MAE}\_\mathrm{P}$ alone as the criterion for defense success. Instead, following prior TSF backdoor settings [5, 6], we always report both clean forecasting error ($\mathrm{MAE}\_\mathrm{C}$) and poisoned forecasting error relative to the attacker’s target ($\mathrm{MAE}\_\mathrm{P}$). Here, $\mathrm{MAE}\_\mathrm{C}$ measures benign forecasting quality on clean inputs, while $\mathrm{MAE}\_\mathrm{P}$ reflects how closely triggered predictions follow the attacker’s target. A successful defense should therefore preserve low $\mathrm{MAE}\_\mathrm{C}$ while increasing $\mathrm{MAE}\_\mathrm{P}$; by contrast, a model that is simply degraded would also suffer in $\mathrm{MAE}\_\mathrm{C}$. We introduced $\mathrm{FDER}$ only as a compact summary to penalize such “false wins,” not as a replacement for the underlying metrics, and we will clarify this point more explicitly in the revision.
>
> In our setting, benign means good forecasting on clean inputs (low $\mathrm{MAE}\_\mathrm{C}$); malicious success means that triggered inputs are steered toward the attacker’s target (low $\mathrm{MAE}\_\mathrm{P}$); and simply wrong means the model performs poorly in general, which is also reflected by high error on clean inputs. We also do not assume that poisoned TSF windows must always have globally distinct trajectories from benign ones, since both the trigger and target patterns are attacker-defined. When these patterns mimic common clean motifs, poisoned and clean windows can indeed become ambiguous. Our point is therefore not that defense success should be judged by trajectory separability, but by whether a method preserves benign forecasting utility while disrupting malicious target alignment. We will make this distinction clearer in the revision.
>
> ---
> [1] Kim et al. *A comprehensive survey of deep learning for time series forecasting: Architectural diversity and open challenges.* Artificial Intelligence Review, 2025.
>
> [2] Xu et al. *CPiRi: Channel Permutation-Invariant Relational Interaction for Multivariate Time Series Forecasting.* ICLR 2026.
>
> [3] Lin et al. *BACKTIME: Backdoor Attacks on Multivariate Time Series Forecasting.* NeurIPS 2024.
>
> [4] Xiang et al. *BadTime: An Effective Backdoor Attack on Multivariate Long-Term Time Series Forecasting.* arXiv:2508.04189, 2025.
>
> [5] Gao et al. *Backdoor Defense via Adaptively Splitting Poisoned Dataset.* CVPR 2023.
>
> [6] Yu et al. *Backdoor Defense via Enhanced Splitting and Trap Isolation.* ICCV 2025.

---

### Decision · Program_Chairs · 2026-04-30

**Decision:**

Accept (regular)

**Comment:**

This paper studies backdoor defenses in multivariate time series forecasting (TSF). Analysis of 13 existing backdoor defenses against TSF backdoor attacks suggests 2 failure modes, which motivates the paper to propose Timeguard, a training-time defense that constructs a channel-wise reliable pool of training samples.

The reviewers are initially concerned with the paper's evaluation metrics, heuristic designs, limited evaluation scope (datasets and attacks), and practical efficiency. The rebuttal responses have clarified these issues. I also agree with the reviewers' assessments of this paper, that the paper is well motivated, and makes a good contribution toward the defensive aspect of TSF backdoors, with good and extensive empirical results. However, I also agree with one of the reviewers that a good definition of a valid backdoor attack in the Timeseries domain is necessary to judge the validity of new technical developments accurately, and I strongly suggest that the authors acknowledge the current assumption of valid backdoor attacks in the revision.